# Universal consensus 3D segmentation of cells from 2D segmented stacks

**Felix Y. Zhou** [1,2] ✉, **Zach Marin** [1,2,3], **Clarence Yapp** [4,5], **Qiongjing Zou** [1], **Benjamin A. Nanes** [2,6], **Stephan Daetwyler** [1,2], **Andrew R. Jamieson** [1], Md Torikul Islam[7], Edward Jenkins[8], Gabriel M. Gihana[1,2], Jinlong Lin[1,2], **Hazel M. Borges** [1,2], **Bo-Jui Chang** [1,2], Andrew Weems[1,2,12], **Sean J. Morrison** [7,9], **Peter K. Sorger** [4,5,10], **Reto Fiolka** [1,11], **Kevin M. Dean** [1,2] & **Gaudenz Danuser** [1,2,13] ✉

Cell segmentation is the foundation of a wide range of microscopy-based biological studies. Deep learning has revolutionized two-dimensional (2D) cell segmentation, enabling generalized solutions across cell types and imaging modalities. This has been driven by the ease of scaling up image acquisition, annotation and computation. However, three-dimensional (3D) cell segmentation, requiring dense annotation of 2D slices, still poses substantial challenges. Manual labeling of 3D cells to train broadly applicable segmentation models is prohibitive. Even in high-contrast images annotation is ambiguous and time-consuming. Here we develop a theory and toolbox, u-Segment3D, for 2D-to-3D segmentation, compatible with any 2D method generating pixel-based instance cell masks. u-Segment3D translates and enhances 2D instance segmentations to a 3D consensus instance segmentation without training data, as demonstrated on 11 real-life datasets, comprising >70,000 cells, spanning single cells, cell aggregates and tissue. Moreover, u-Segment3D is competitive with native 3D segmentation, even exceeding when cells are crowded and have complex morphologies.

Instance segmentation is the problem of assigning pixels in a 2D or voxels in a 3D image to unique objects. Near universally, it is the first step in quantitative image analysis. Only through segmentation are objects, such as nuclei[1,2], organelles[3], cells[4], bacteria[5], plants[6], organs[7,8] or vasculature[9], explicitly identified for further measurements. Segmenting individual cells is readily accomplished by intensity thresholding and connected component analysis when they are isolated, well-contrasted and uniformly illuminated[10]. However, this scenario is rare. In practice, cells of diverse morphologies often aggregate in clusters in vitro and in vivo, precluding separation by mere thresholding[11,12]. Furthermore, the imaging processes used to visualize cellular structures often result in weak, partial, sparse or unspecific foreground intensity signals[3,13].

Thanks to advancements in GPU architecture and publicly available labeled datasets, generalist or 'foundational' 2D cell segmentation models have emerged both for interactive segmentation using prompts

[1]Lyda Hill Department of Bioinformatics, University of Texas Southwestern Medical Center, Dallas, TX, USA. [2]Cecil H. & Ida Green Center for System Biology, University of Texas Southwestern Medical Center, Dallas, TX, USA. [3]Max Perutz Labs, Department of Structural and Computational Biology, University of Vienna, Vienna, Austria. [4]Laboratory of Systems Pharmacology, Department of Systems Biology, Harvard Medical School, Boston, MA, USA. [5]Ludwig Center at Harvard, Harvard Medical School, Boston, MA, USA. [6]Department of Dermatology, University of Texas Southwestern Medical Center, Dallas, TX, USA. [7]Children's Research Institute, University of Texas Southwestern Medical Center, Dallas, TX, USA. [8]Kennedy Institute of Rheumatology, University of Oxford, Oxford, UK. [9]Howard Hughes Medical Institute, University of Texas Southwestern Medical Center, Dallas, TX, USA. [10]Department of Systems Biology, Harvard Medical School, Boston, MA, USA. [11]Department of Cell Biology, University of Texas Southwestern Medical Center, Dallas, TX, USA. [12]Present address: Department of Molecular Biosciences, University of Texas at Austin, Austin, TX, USA. [13]Present address: Institute of Human Biology, Roche Pharma Research & Early Development of F. Hoffmann-La Roche Ltd, Basel, Switzerland. ✉e-mail: felix.zhou@utsouthwestern.edu; gaudenz.danuser@roche.com

such as μSAM[14], CellSAM[15] and for dense cell segmentation such as Cellpose[4] and transformer models[12]. These methods leverage 'big data' and harness diversity in the training set to achieve impressive 2D segmentation performance across imaging modalities and cell types[12].

Physiologically, cells mostly reside in complex 3D environments. The importance of studying cell biological processes in the relevant physiological 3D environments is well documented[16–18]. This yielded the development of countless approaches for 3D in situ tissue imaging with subcellular resolution[19]. Unlocking the potential of this imaging necessitates reliable, general and scalable 3D cell segmentation solutions. Simply replicating the training strategy of 2D foundation models is impossible: requiring large well-labeled and diverse 3D cell datasets, and more computational resources.

Even for 2D cell segmentation, the image datasets needed to train foundation models are minimal. The Cellpose training dataset comprises 540 training images (total ~70,000 cells, five modalities), and the most recent and largest multimodal segmentation challenge dataset[12] comprises 1,000 training images (total 168,491 cells, four modalities). Replicating a densely labeled 3D dataset with comparable levels of cell diversity and numbers, given more complex microenvironments, more variable image quality, and more diverse morphologies and cell packing would be a formidable undertaking[20,21]. Despite ongoing efforts to develop scalable 3D annotation tools[22,23] with AI assistance[24,25] and proofreading[26], the generation of meaningful training data requires an extraordinary level of manual intervention[21,27]. Even with expert annotators, labeling is affected by inter-operator and intra-operator variation[28–30] and is inherently biased toward easy cases. Synthetic[31,32], partial[33] or generative[34] model-derived datasets have been proposed to alleviate the need for fully labeled data but have only been demonstrated with largely star-convex morphologies. It is unclear how these methods generalize to encompass more complex morphologies, microenvironments, and future, novel 3D imaging modalities, as they still fundamentally use data-driven supervised learning.

To address the shortcomings of directly training 3D segmentation models, we revisit the idea of leveraging 2D cell segmentations from orthoviews without model retraining to generate a 'consensus' 3D segmentation. Almost universally, existing tools construct a 3D segmentation by matching and stitching 2D segmentations across x-y-slices[4,35]. Relying on a single view, these 3D segmentations are notoriously rasterized and erroneously join multiple touching cells as tubes[35–37]. CellStitch[38] and 3DCellComposer[36] propose matching across orthogonal x-y, x-z and y-z views to create a more accurate consensus 3D segmentation. However, discrete matching approaches are inherently difficult to scale up with cell number, not applicable to branched structures and cannot easily handle missing, undersegmented or oversegmented cells across slices. Alternatively, Cellpose[4] proposed to average predicted 2D flow vectors along the x-y, x-z and y-z directions to construct a 3D gradient map. By tracing the gradient map to the simulated heat origin, the 3D cell instances are found by grouping voxels flowing to the same sink. While conceptually elegant, its execution has been restricted to Cellpose-predicted gradients and demonstrates limited performance on anisotropic[38], noisy or morphologically non-ellipsoidal datasets[11]. Moreover, we and others[38]

have observed fragmentation of whole 3D cells into angular sectors, a behavior inconsistent with Cellpose's representation of whole cells as a 360° angular field, and worse than just stitching the 2D cell masks[38].

Here, we derive a formal framework for 2D-to-3D segmentation from first principles that unifies stitching and gradient tracing. We show that 2D-to-3D translation can be formulated generally as an optimization problem, whereby we reconstruct the 3D gradient vectors of the distance transform representation of each cell's 3D medial-axis skeleton. Then, 3D cells can be optimally reconstructed using gradient descent and spatial connected component analysis. Using these two principles, we developed u-Segment3D, a framework to perform consensus 3D segmentation from 2D segmentations in one, two or all three orthoview stacks (for example, in x-y, x-z and y-z). Our method is universal, independent of both the cell morphology and the 2D segmentation method used to generate pixel-based instance segmentation masks. We validated u-Segment3D in 11 real-life datasets, encompassing >70,000 cells from cell aggregates, embryos, tissue and entire vasculature networks. We compared 2D-to-3D u-Segment3D translation of trained 2D and counterpart native 3D segmentation models on the same datasets. u-Segment3D matched, and for crowded cells or complex, branched morphologies even exceeded native 3D segmentations. Thus, u-Segment3D's implementation guarantees that the better the 2D segmentation, the better the resultant 3D segmentation. Finally, we demonstrate the flexibility and capacity of u-Segment3D to segment unseen 3D volume datasets of anisotropic cell cultures, and unwrapped embryo surfaces[39]; high-resolution single cells and cell aggregates with intricate surface protrusions[40]; thin, sprouting vasculature in zebrafish, and tissue architectures imaged with spatial multiplexing[41] and extracellular labeling[42].

u-Segment3D is implemented in Python 3, and is freely installable from https://github.com/DanuserLab/u-segment3D/ and the Python Package Index, PyPI.

## Results

### A formalism for 2D-to-3D segmentation

Dense instance segmentation identifies every object in the image and assigns a unique object ID to all pixels and voxels. This is equivalent to two separate tasks: (i) binary labeling every image voxel as foreground (value 1) or background (value 0), and (ii) assigning to each designated foreground voxel, a unique ID denoting the instance (Fig. 1a). The critical challenge of 2D-to-3D segmentation is preserving the 2D cell boundaries in all orthoviews, and obtaining the correct number of 3D instances after translation, while keeping computation low.

Starting for the purpose of illustration with a 2D instance segmentation, erosion of each object border by one pixel spatially separates all cell areas. In this scenario, cell instances are faithfully represented by a binary foreground/background image and parsing of object IDs by connected component analysis (Fig. 1b). Unlike the discrete label representation, this binary image is a continuous scalar field that can be computed from its one-dimensional (1D) slices in either the x- or y-direction (Fig. 1c). This is because the 2D objects are spatially contiguous across adjacent 1D regions. Hence, a 1D instance segmenter that delineates individual cell boundaries with touching 1D cells can

**Fig. 1 | u-Segment3D is a toolbox for generating consensus instance 3D segmentation from 2D segmentations. a**, Computational representation of the 2D segmentation of densely packed cells as two images, a foreground binary mask and a labeled image where each cell is assigned a unique integer ID. Panel **a** adapted from ref. 4, Springer Nature America. **b**, Equivalent representation of the eroded segmentation where individual cells are spatially separated and can be equivalently represented using a foreground binary mask that can be parsed using connected component analysis to recover individual cell IDs. **c**, Schematic of decomposing 2D instance cell segmentations into 1D slice segmentations scanning the 2D image in orthogonal x- or y- directions. **d**, Schematic of the reconstruction of 2D instance cell segmentations using

1D segmentations from orthogonal x- and y-direction scans of the 2D image to generate a consensus foreground followed by 2D spatial proximity clustering. Segmentations are eroded vertically or horizontally for x- or y-direction scans, respectively. ≡ denotes equivalent representation. **e**, Schematic of the minimal set of algorithmic steps to operationalize the framework in **d** for a consensus 3D segmentation of densely packed cells. **f**, u-Segment3D is a toolbox to enable the application of the algorithmic steps in **e** to real datasets with additional preprocessing methods to adapt any pretrained 2D segmentation model or 2D method and postprocessing methods to improve and recover missing local features in the reconstructed 3D segmentation such as subcellular protrusions.

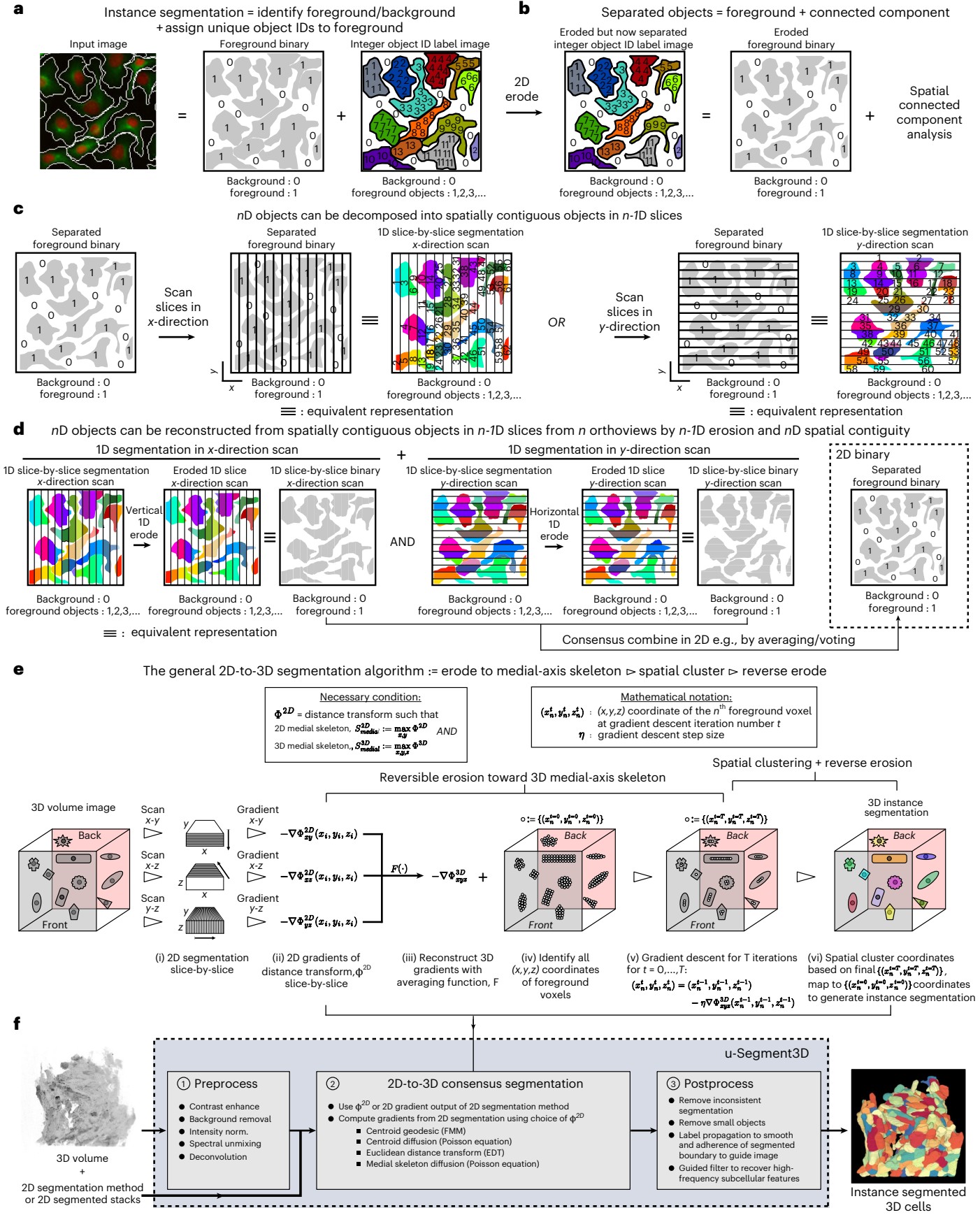

be applied in reverse to reconstruct the 2D instance segmentation (Fig. 1d). The 1D segmenter is run slice-by-slice in the $x$-direction. Each unique 1D 'cell' is eroded vertically within each $x$-slice and then restacked to a 2D binary image. The erosion ensures spatial separation in the $x$-direction of the 2D binary; however, cells may still touch in the orthogonal $y$-direction. The binary generated by an $x$-direction scan must be combined by a consensus operation, for example, computing a pixel-wise average (and rebinarize), or pixel-wise intersection, with the equivalent binary produced from an orthogonal $y$-direction scan to reconstruct a single foreground binary representing fully separated 2D cells. Spatial connected components can then identify and label all contiguous 2D regions as unique cells. The erosion in $x$- and $y$-slices is reversed for each 2D cell to recover touching cell boundaries. The same first-principle arguments apply to reconstructing an $n$D segmentation from minimally $n$ orthogonal $n$-1D segmentations, that is, for 3D cell segmentations from $x$-$y$, $x$-$z$ and $y$-$z$ 2D slices.

In practice, this conceptual framework is suboptimal if directly applied to data. Images rasterize the cell shape, with resolution dictated by the grid size; the lower-dimensional segmenter is imperfect with inevitable missing or inconsistent segmentations across slices whereby a cell splits into multiple or multiple becoming one; and uniform morphological dilation after $n$D reconstruction will not preserve eroded curvature features. Discrete computational processes such as label matching and morphological operations cannot overcome these issues across orthoviews. The former cannot manipulate and correct erroneous inputs, either segmentations are matched or are discarded. The latter does not retain the shape manipulation history. A central question is how to erode 2D cells to guarantee framework applicability to heterogeneous cell sizes and morphologies.

u-Segment3D proposes a continuous implementation to address these issues. First, each cell is represented as a foreground binary and associated dense gradient field. This enables segmentations to be flexibly manipulated with continuous computations: smoothing to impute across slices or to join an arbitrary number of neighboring cells; and averaging to obtain consensus, downvoting information unsupported by majority orthoviews. Second, we iteratively erode cell shapes to their medial-axis skeleton[43,44] or medial-axis transform (MAT). By construction, the MAT maximally separates foreground points belonging to separate neighboring cells during erosion and is computable for any cell shape. Crucial for the eventual 2D-to-3D segmentation, by definition, the MAT of 2D slices coincides with the MAT of the corresponding 3D object[44–46]. Resolution permitting, the 2D skeletal slices of each unique 3D object remain spatially proximal after 3D stacking, enabling identification by spatial proximity. Moreover, medial-axis skeletons are attractors of distance transforms[43,47], $\Phi$. The gradient field of a cell is, therefore, the distance transform gradient, and gradient descent specifies a reversible, continuous erosion process without shape information loss. A cell shape is eroded by advecting its foreground coordinates with step size $\eta$, in the direction of the local gradient, $\nabla\Phi$ (Methods). The history of coordinates is

the reverse erosion. The general 2D-to-3D segmentation algorithm (Fig. 1e) is thus:

1. Generate 2D pixel-based instance segmentations independently in orthogonal $x$-$y$, $x$-$z$ and $y$-$z$ views.
2. Convert cells in $x$-$y$, $x$-$z$ and $y$-$z$ views to their 2D gradient field representation by computing a 2D distance transform, $\Phi^{2D}$, whose attractor is a 2D medial-axis skeleton, and its 2D gradient, $-\nabla\Phi^{2D}_{xy}$, $-\nabla\Phi^{2D}_{xz}$ and $-\nabla\Phi^{2D}_{yz}$, respectively. Should the 2D segmenter already predict suitable 2D gradients, for example Cellpose[4], these can be directly used instead (Methods).
3. Reconstruct the 3D gradients of the distance transform $\nabla\Phi^{3D}_{xyz}$ from the 2D gradients, using a consensus averaging function, $F$ (Methods), where $\nabla\Phi^{3D}_{xyz} \approx F(\nabla\Phi^{2D}_{xy} + \nabla\Phi^{2D}_{xz} + \nabla\Phi^{2D}_{yz})$.

   Apply $F$ to the 3D foreground stacked from $x$-$y$, $x$-$z$ and $y$-$z$ views and binarize to reconstruct the consensus 3D foreground binary, $B$ (Methods).
4. Identify all 3D foreground voxels in $B$ as initial ($t = 0$) coordinates of gradient descent. The coordinate of the $n$th foreground voxel is $(x_n^{t=0}, y_n^{t=0}, z_n^{t=0})$.

   Foreground :=
   $$\{\{(x_1^{t=0}, y_1^{t=0}, z_1^{t=0}), ..., (x_n^{t=0}, y_n^{t=0}, z_n^{t=0})\} \,|\, B(x_n^{t=0}, y_n^{t=0}, z_n^{t=0}) = 1\}$$

5. Apply gradient descent in 3D to iteratively advect the $n$th foreground point for a fixed number of total iterations, $t = 1, 2, ..., T$, with step size $\eta$ to uncover its 3D medial-axis skeleton attractor

   $$(x_n^t, y_n^t, z_n^t) \leftarrow (x_n^{t-1}, y_n^{t-1}, z_n^{t-1}) - \eta\nabla\Phi^{3D}_{xyz}(x_n^{t-1}, y_n^{t-1}, z_n^{t-1})$$

   where the gradient field is normalized to unit-length vectors, $\nabla\Phi \leftarrow \frac{\nabla\Phi}{|\nabla\Phi|}$, that is, the parameter $\eta$ defines the true voxel step size.
6. Cluster all points in the eroded space, at final advected coordinates ($t = T$) by spatial proximity, labeling each cluster with a unique positive integer object ID, with all points part of the same cluster having the same ID. This produces a 3D label image, $L^{t=T}$.

   $$L^{t=T}(x_n^T, y_n^T, z_n^T) = id \in \mathbb{z}^+$$

   Finally, to obtain the 3D instance segmentation in the initial non-eroded space, generate the label image corresponding to the initial coordinates ($t = 0$), $L^{t=0}$.

   3D segmentation :=
   $$L^{t=0}(x_n^{t=0}, y_n^{t=0}, z_n^{t=0}) = L^{t=T}(x_n^T, y_n^T, z_n^T) = id \in \mathbb{z}^+$$

**u-Segment3D optimally reconstructs 3D shapes from 2D.** The u-Segment3D pipeline implements each step of the outlined 2D-to-3D segmentation algorithm (Fig. 1f and Supplementary Movie 1).

**Fig. 2 | u-Segment3D reconstructs 3D objects from their 2D slices in orthogonal $x$-$y$, $x$-$z$ and $y$-$z$ views. a,b,** Characterization of test dataset. Illustration of the eight computed geometrical and topological features to describe the shape complexity of test data (**a**). UMAP embedding of individual cells from 11 datasets covering the full spectrum of morphological complexity from convex-spherical to branching to networks (**b**). Left, colormap of individual dataset and total number of uniquely labeled cells in each dataset. Middle, UMAP, each point is a cell, color coded by their dataset. Right, median UMAP coordinate of each dataset (top left) and heat map of three features representing the extent of branching (total number of skeleton nodes, top right), the extent of elongation (stretch factor = 1 − minor length/major length) and their image size (total number of voxels). **c,** Illustration of the experimental workflow to compute 2D slice-by-slice distance transforms in orthogonal directions using a 3D reference shape. These 2D transforms are then integrated by u-Segment3D to reconstruct

a 3D cell shape. Comparison of reference and reconstruction is used to assess performance of the 2D-to-3D reconstruction. $(x_n^t, y_n^t, z_n^t)$ denotes the coordinate of the $n$th foreground voxel at iteration number $t$, $\eta$ the step size, $\delta, \mu$ the weighting of current and previous gradients where $\delta \geq \mu$. We set $\delta = 1$ and $\mu$ is the momentum. **d,** Reconstruction performance measured by the AP curve (Methods) for the Ovules dataset using three different 2D distance transforms. From top to bottom: AP versus IoU curve; 3D rendering of reference versus reconstructions based on point-based diffusion distance transform and skeleton-based EDT. Shown are the reconstructed 3D objects and their respective mid-slices in the three orthogonal views. **e,f,** Same as **d** for the LRP dataset containing instances of branching morphology and DeepVesselNet representing complex, thin network morphologies. Individual cells are uniquely colored but are not color matched with the reference object. Diffusion fragmented the cell with the longest branch into two 'cells' (white arrowheads).

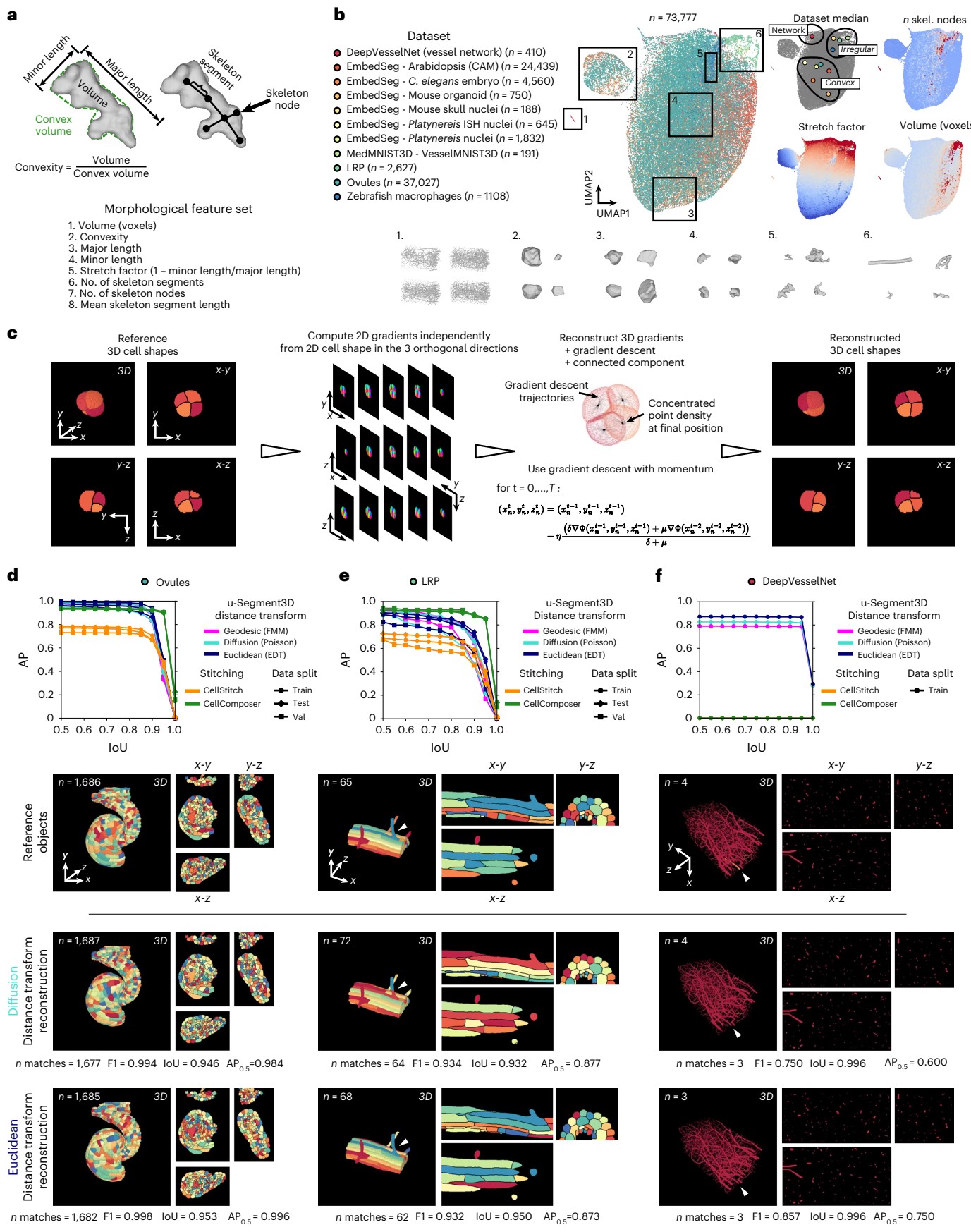

The rationale and robustness of the algorithmic implementations are documented in Supplementary Note 1. Briefly, (i) since an object's medial-axis skeleton is not unique[43,47], u-Segment3D implements a choice of distance transforms to translate 2D segmentation masks to gradients with respect to computation speed, accuracy and compatibility with 2D model outputs (Supplementary Fig. 1 and Movie 2); u-Segment3D implements (ii) a content-based consensus averaging function, $F$, to fuse 2D image stacks while accounting for inconsistencies between orthoviews, assessed by the local spatial variance in a pixel neighborhood of $N \times N \times N$ (Supplementary Fig. 2); (iii) a numerically accurate gradient descent in 2D and 3D to propagate foreground points together to their attractors; and (iv) spatial proximity clustering to identify individual attractors as the final segmentation. Advected points are counted at each voxel, and smoothed by Gaussian filtering (width = $\sigma$) to generate a sparse local point density heat map, $\rho$. Then, connected component analysis of the binarized $\rho$, with a global threshold, mean($\rho$) + $k \cdot$ std($\rho$) (default $k = 1$, being tunable) identifies all spatially separated attractors. With heterogeneous cell shapes, points do not converge to their attractors at equal speed. These steps were crucial to identify any-shaped attractors and prevent severe over-segmentation or fractured shapes using alternative density-based clustering methods (Supplementary Figs. 3 and 4 and Supplementary Movie 3). Importantly, our implementation produces the expected evolution behavior: at iteration 0, the recovered cell shapes are equivalent to connected component analysis of the binary cell segmentation, but converge with increasing gradient descent iterations to the exact cell shapes and numbers (Supplementary Fig. 4). The use of $\rho$ adds a probabilistic perspective to cluster identification. By increasing $\sigma$ u-Segment3D can 'fuzzy' link erroneously split clusters, equivalent to merging segmentations in the final 3D segmentation. Finally, by carefully investigating 1D-to-2D (Supplementary Figs. 5 and 6) and 2D-to-3D (Supplementary Fig. 7) shape reconstruction on exemplars selected to reflect the diversity and complexity of real-life data, we found smoothing of the reconstructed 3D gradients, and a suppressed gradient descent, with a decaying step size, $\eta = \frac{1}{1+\tau \cdot t}$, for higher iterations was essential to realize optimal 2D-to-3D segmentation across all morphotypes (Supplementary Note 2 and Supplementary Movie 4). Here, $t$ denotes the iteration and $\tau$ an adjustable decay rate[5].

To test the accuracy of u-Segment3D's reconstruction of 3D objects from 2D slices, we assembled 11 published 3D datasets with dense segmentation labels (Supplementary Table 2). The total number of cells across all datasets was 73,777. Visualizing the cells in 2D by applying uniform manifold approximation and projection (UMAP)[48] to eight computed morphological features (Fig. 2a and Methods) chosen to assess cell size (total number of voxels), elongation (stretch factor) and topological complexity (number of skeleton nodes), the assembled datasets capture the spectrum of 3D morphotypes encountered in tissue including thin, complex vessel-like

networks (region 1), pseudo-spherical (regions 2–4), irregular (region 5) and tubular or branched (region 6; Fig. 2b). Using the per-dataset median UMAP coordinate, we categorized the 11 datasets into three super-morphotypes: convex (*Caenorhabditis elegans* embryo[11]/mouse organoid[11]/mouse skull nuclei[11]/*Platynereis* nuclei[11]/Arabidopsis (Cambridge University, (CAM))[11]/Ovules[6]); irregular/branched (zebrafish macrophages[49]/*Platynereis* ISH nuclei[11]/VesselMNIST3D[50]/lateral root primordia (LRP)[6]); and complex networks (DeepVesselNet[9]).

We used the provided segmentation labels of each dataset as reference 3D objects, and tested the reconstruction of each 3D object from their 2D slice projections (Fig. 2c). All 3D objects were scanned in a slice-by-slice manner in *x-y*, *x-z* and *y-z* views, treating each contiguous region within a 2D slice as a unique 'cell'. For each 2D 'cell', its 2D gradients were computed using a distance transform to assemble the 3D gradients and reconstruct the 3D object by 3D gradient descent and connected component analysis. This setup allowed us to assess the accuracy of 3D shape reconstruction from 2D gradients. Three different 2D distance transforms were tested: Poisson diffusion centroid as an example of an explicit transform and also used in Cellpose[4]; Euclidean distance transform (EDT) as an example of an implicit transform and used within models like Omnipose[5] and StarDist[1]; and geodesic centroid as a second example of an implicit transform, but computed by solving the Eikonal equation with fast marching (Methods). For all datasets, the number of iterations in the total gradient descent was fixed at 250, and reference 3D objects were resized to isotropic voxels with nearest-neighbor interpolation (Supplementary Table. 2). Temporal decay $\tau$ was the only parameter adjusted for each transform and dataset (Supplementary Table 3). Postprocessing was applied to remove cells of less than 15 voxels. Reconstructed 3D objects were evaluated using the average precision (AP) and F1 curve. AP and F1 are two aggregate scores of precision and recall, with AP equally weighting true positives, false positives and false negatives (Methods). Their curves compute the score between reference and reconstructed shapes as the overlap cutoff (intersection over union (IoU)) for a valid match increases from 0.5 to 1.0 (perfect overlap; Methods). We use $AP_{0.5}$ and $F1_{0.5}$ to denote AP and F1 with IoU cutoff = 0.5 henceforth. For perfect reconstruction, AP = 1 and F1 = 1 at all IoU cutoffs. In practice, due to limited numerical accuracy, AP and F1 always drop to 0 above an IoU cutoff.

For each of the super-morphotypes, we first analyzed the dataset with the highest number of cells: Ovules for convex (Fig. 2d), LRP for irregular (Fig. 2e) and DeepVesselNet for networks (Fig. 2f and Supplementary Movie 5). We find near-perfect reconstruction across all distance transforms, morphotypes and data splits, qualitatively and quantitatively: Ovules, $AP_{0.5} \approx 1.0$, LRP, $AP_{0.5} \geq 0.8$ and DeepVesselNet, $AP_{0.5} \geq 0.8$. As expected, increased $\tau$ was necessary for thinner, branching cells: using EDT, $\tau = 0.5$ for Ovules and LRP, $\tau = 2.0$ for DeepVesselNet. These results were reflected in the other eight datasets (Extended Data Fig. 1), with $AP_{0.5} \geq 0.75$. F1 curves support the same conclusion as $AP_{0.5}$ across all datasets (Extended Data Fig. 2)

**Fig. 3 | Consensus 3D segmentation from 2D stacks is competitive with native 3D segmentation and superior for densely packed cells and cells with complex morphologies. a**, Schematic of the steps to construct (i) classical unsupervised distance transform watershed (dtWS) 3D segmentation baseline (left), and the training and application for (ii) native 3D segmentation (middle) and (iii) consensus 3D segmentation from 2D models with u-Segment3D (right). **b,c**, AP and F1 curves on test split and 3D render of reference cell masks of an example image (top), and corresponding output with each 3D segmentation method (bottom) for Ovules (**b**) and LRP (**c**) datasets. Each trained model is colored uniquely with a solid line, using a darker hue for native 3D segmentation and dashed line, and a lighter hue for consensus segmentation using their trained 2D counterpart model. **d**, $AP_{0.5}$ of trained 2D model in 2D segmentation (*x* axis) plotted against matching $AP_{0.5}$ of 3D performance after u-Segment3D. Each point is uniquely colored by 2D model (left) and dataset (right). In **b–d**, Cellpose 2D segmentations were generated from predicted outputs using u-Segment3D's

method, indicated by the (u) annotation. **e**, $AP_{0.5}$ of training native 3D model in 2D segmentation (*x* axis) plotted against $AP_{0.5}$ of training its 2D model counterpart and consensus segmentation with u-Segment3D. Each point is uniquely colored by model (left) and dataset (right). In **d** and **e**, the dashed black line represents identical performance. **f**, Schematic of training an EmbedSeg3D model using u-Segment3D generated consensus segmentation of EmbedSeg2D segmented stacks (left). Render of the reference 3D ground-truth masks and AP curve of native 3D training with reference 3D masks (with reference), with consensus segmentation derived from 2D (with Consensus), and the consensus generated by u-Segment3D from 2D (Consensus; top row, right). 3D renders of the segmentation for the same image as the reference, with each method (bottom row, right). **g**, 3D render of the consensus segmentation from using only one (*y-z*), two (*x-y* and *y-z*) and all three (*x-y*, *x-z* and *y-z*) EmbedSeg2D segmented stacks of an example image (left). Corresponding AP and F1 curve evaluated across all test images using one, two and all three orthoviews.

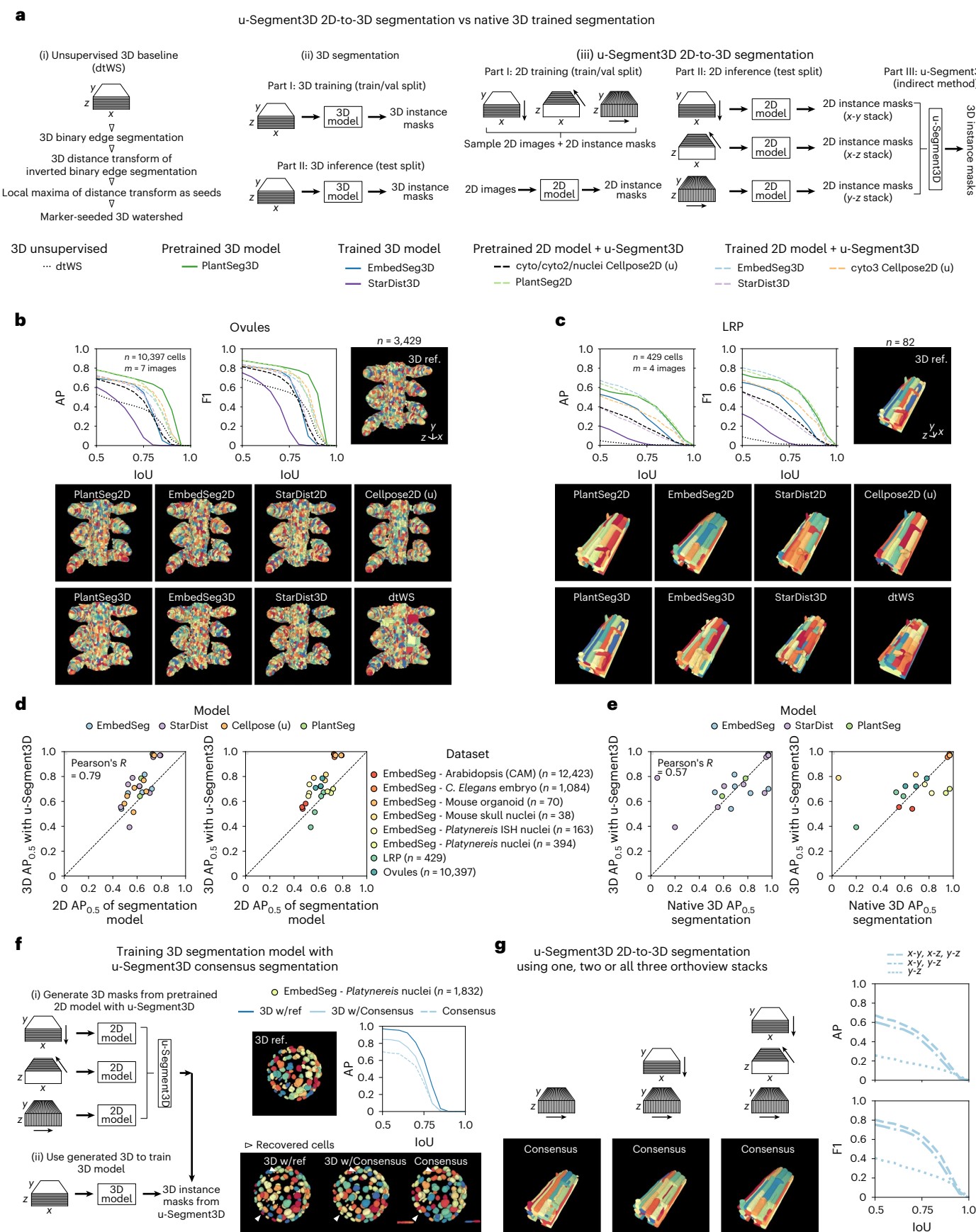

but with higher scores, Ovules, $F1_{0.5} \approx 1.0$, LRP, $F1_{0.5} \geq 0.9$, and DeepVesselNet, $F1_{0.5} \geq 0.85$. Moreover, IoU was high, with both performance metrics decaying prominently only for $IoU \geq 0.85$ and for many, $IoU \geq 0.95$. Notably, $IoU > 0.8$ masks are nearly indistinguishable from the reference by eye[5,51]. We also compared reconstruction by stitching using CellComposer[36] and CellStitch[38]. Whereas u-Segment3D demonstrates universal applicability with high performance across all datasets and distance transforms, CellStitch and CellComposer were inconsistent, with varying performance within the convex morphotype, ranging from $AP_{0.5} \approx 0.60$ (mouse organoid) to 1.0 (Ovules). Moreover, both are inapplicable to branched morphologies ($AP_{0.5} \approx 0.4$ (VesselMNIST3D) and $AP_{0.5} \approx 0.0$ (DeepVesselNet)).

As noted by Omnipose[5], there are discrepancies between different distance transforms for different morphotypes. The two explicit transforms with point-source attractors, Poisson and geodesic, differed minimally. Both outperformed EDT on convex morphologies, most evidently in the LRP validation data (Fig. 2e and Extended Data Fig. 2b), mouse skull nuclei test (Extended Data Figs. 1d and 2g), *Platynereis* ISH nuclei test (Extended Data Figs. 1e and 2h) and zebrafish macrophages (Extended Data Figs. 1h and 2k) datasets. This is primarily due to the increased stability of explicit transforms. However, EDT was superior for thin and complex vasculature networks (Fig. 2f and Extended Data Fig. 2c; DeepVesselNet). We visualized both the diffusion and EDT reconstruction on exemplars from Ovules, LRP and DeepVesselNet. Despite similar F1 and IoU, only the EDT fully reconstructed all branching cells. Diffusion fragmented the cell with the longest branch into two 'cells' (Fig. 2e). Importantly, the fragments are standalone and not erroneously part of neighboring cells.

In summary, u-Segment3D is universally applicable across all morphotypes and empirically achieves near-perfect, consistent 3D shape reconstruction from 2D slice projections in orthogonal views. In the best case, it is perfect. In the worst case, a subset of branching cells might be decomposed into a few standalone segments that could be reassembled. Our results indicate that, generally, explicit distance transforms perform more optimally; however, these are more computationally expensive than implicit transforms.

## 2D-to-3D u-Segment3D segmentation is comparable to native 3D

We next tested whether u-Segment3D could use 2D pretrained foundation or generalist segmentation models for 3D segmentation. These models either (i) already predict a suitable distance transform or 2D gradients (Extended Data Fig. 2 and Methods), for example Cellpose[4], or (ii) predict 2D instance segmentation masks. u-Segment3D accounts for both cases (Fig. 3a): for the former, using the predicted 2D gradients directly (the direct method), and for the latter, users choosing a 2D distance transform to translate the 2D segmentation masks to 2D gradients (the indirect method). Unlike the ideal 2D projections derived from given 3D shapes, now the imperfect 2D segmentations mean the morphology of the reconstructed 3D foreground binary is an additional determinant of overall performance. If the foreground

does not provide a contiguous path for gradient descent, the resulting segmentation will be fragmented, even with correct gradients. We used pretrained generalist Cellpose models with automated tuning of two key parameters, diameter and foreground threshold (Supplementary Figs. 8 and 9 and Supplementary Note 3) to compare the optimal segmentations of both methods on the validation or test splits of 9/11 datasets when available (see Supplementary Table. 4 for parameter details).

For Ovules, LRP and DeepVesselNet, we assessed the direct result relative to Cellpose 3D and native 3D segmentation baselines: an unsupervised 3D watershed as the lower bound, and a dataset-specific native 3D segmentation as the upper bound. We compared the indirect result to the stitching-based methods, CellComposer and CellStitch (see Supplementary Note 4 for detailed analysis; Methods). Without training, u-Segment3D delivered strong performance across all three datasets, greatly surpassing watershed. u-Segment3D recovered all salient vasculature in DeepVesselNet, and matched specialist 3D baselines in Ovules using a generalist model (Extended Data Fig. 3), and in LRP using a pretrained specialist model[5] (Extended Data Fig. 4 and Methods). In comparison, Cellpose 3D's performance deteriorated rapidly from slightly worse in Ovules, to watershed levels in LRP, to $AP_{0.5} = 0$ in DeepVesselNet with increasing over-segmentation. Importantly, as expected of true consensus translation of 2D segmentations, u-Segment3D exhibited similar performance using either the direct or indirect method, and preserved or exceeded the quality of 2D segmentation inputs from *x-y*, *x-z* and *y-z* orthoviews in its consensus 3D segmentation without view bias. In LRP, u-Segment3D exploited complementary information from all views. With marginally worse performance in *y-z*, both *x-y* and *x-z* AP curves were improved for both tested pretrained models (Extended Data Fig. 3c(iv)). In contrast, CellStitch and CellComposer now using less-than-perfect segmentation inputs were at best only as good as watershed and exhibited clear preference for *x-y* irrespective of dataset. In LRP, despite the *y-z* view being the most convex-shaped with the best-predicted 2D segmentations, *y-z* had the worst AP curve, with a $\approx 0.5$ drop in $AP_{0.5}$ after CellStitch and CellComposer stitching. Finally, except for Arabidopsis (CAM; best $AP_{0.5} = 0.4$), which had low image quality and densely packed cells, u-Segment3D segmented the six EmbedSeg datasets using the direct method with at least one pretrained model outperforming watershed and $AP_{0.5} \geq 0.6$ (Extended Data Fig. 5). Moreover, mouse organoids and skull nuclei matched a specialist-trained native 3D EmbedSeg3D[11] model (Methods). For mouse organoids, u-Segment3D and cyto2 ($AP_{0.5} = 0.93$) even nearly matched ideal 2D inputs ($AP_{0.5} = 1.0$).

Altogether, our experiments with pretrained Cellpose 2D models suggest that u-Segment3D can translate 2D orthoview segmentations with performance rivaling natively trained 3D models. We next formally assessed this to investigate the pros and cons of 2D-to-3D segmentation across all eight datasets (minus DeepVesselNet) amenable to being trained with 3D instance cell segmentation models. For each dataset, we constructed training, validation and test splits and

---

**Fig. 4 | u-Segment3D postprocessing recovers missing high-frequency, high-curvature subcellular structures. a**, Workflow with postprocessing to segment individual cells and recover subcellular features of each cell. 3D rendering (top) and *x-y* mid-slice (bottom) of the output at each step. Image in panel **a** adapted from fig. 2 in ref. 60. **b**, Binary segmentation and recovery of lamellipodial features on a dendritic cell using u-Segment3D postprocessing. 3D rendering of the (i) deconvolved input, (ii) initial 3D consensus segmentation integrating Cellpose 2D cell probability maps (after step ii of a)), (iii) final postprocessed 3D segmentation (after step (iv) of **a**)) and comparison with the segmentation from binary Otsu thresholding on the 3D image intensity. *g* denotes the genus of extracted surface mesh. **c,d**, Binary segmentation and recovery of filopodial and ruffle features on a human bronchial epithelial cell (HBEC) and COR-L23 cell using u-Segment3D postprocessing as in **b**). **e**, Segmentation of T cells

with u-Segment3D postprocessing. 3D rendering of deconvolved image volume (top), initial 3D consensus segmentation integrating Cellpose 2D outputs (middle) and final 3D segmentation with recovered subcellular protrusions (bottom). **f**, Segmentation of zebrafish macrophages using Cellpose 2D outputs with u-Segment3D postprocessing. 3D rendering of deconvolved image volume (top), initial 3D segmentation from aggregated Cellpose 2D outputs (middle) and final 3D segmentation with recovered subcellular protrusions (bottom). **g**, Binary 3D segmentation of developing zebrafish vasculature using Cellpose 2D outputs with u-Segment3D postprocessing. 3D rendering of raw image volume (top), initial 3D segmentation from aggregated Cellpose 2D cell probability maps (middle) and final 3D segmentation with recovered sprouting vessels (bottom). Images in panels **b** and **c** adapted from ref. 40. Images in panel **f** adapted from ref. 49.

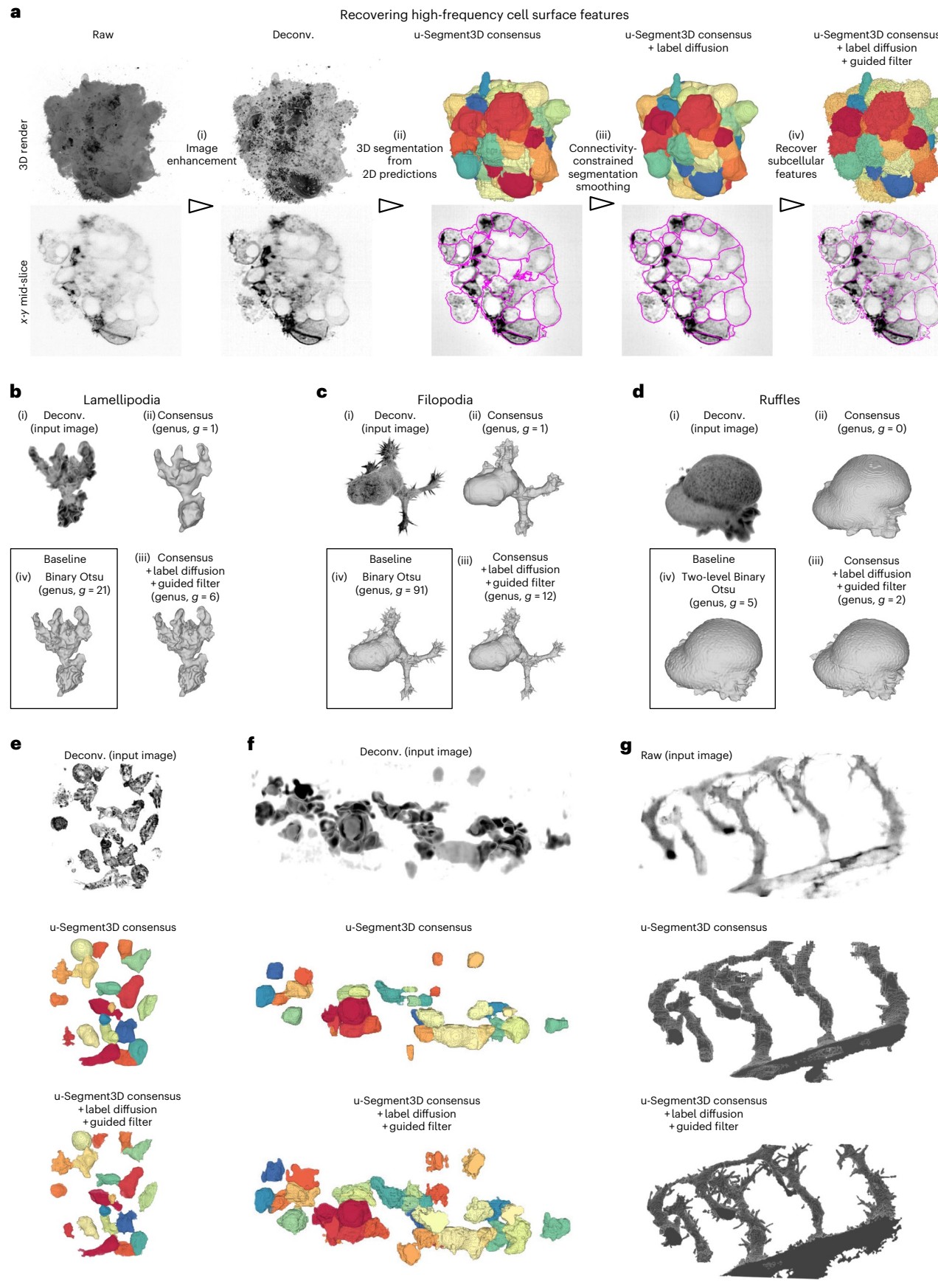

**a** | Recovering high-frequency cell surface features

Raw → (i) Image enhancement → Deconv. → (ii) 3D segmentation from 2D predictions → u-Segment3D consensus → (iii) Connectivity-constrained segmentation smoothing → u-Segment3D consensus + label diffusion → (iv) Recover subcellular features → u-Segment3D consensus + label diffusion + guided filter

3D render / x-y mid-slice

**b** Lamellipodia
(i) Deconv. (input image)
(ii) Consensus (genus, g = 1)
(iv) Baseline Binary Otsu (genus, g = 21)
(iii) Consensus + label diffusion + guided filter (genus, g = 6)

**c** Filopodia
(i) Deconv. (input image)
(ii) Consensus (genus, g = 1)
(iv) Baseline Binary Otsu (genus, g = 91)
(iii) Consensus + label diffusion + guided filter (genus, g = 12)

**d** Ruffles
(i) Deconv. (input image)
(ii) Consensus (genus, g = 0)
(iv) Baseline Two-level Binary Otsu (genus, g = 5)
(iii) Consensus + label diffusion + guided filter (genus, g = 2)

**e** Deconv. (input image) / u-Segment3D consensus / u-Segment3D consensus + label diffusion + guided filter

**f** Deconv. (input image) / u-Segment3D consensus / u-Segment3D consensus + label diffusion + guided filter

**g** Raw (input image) / u-Segment3D consensus / u-Segment3D consensus + label diffusion + guided filter

performed (i) distance transform watershed as an unsupervised classical baseline, and (ii) trained 3D models and (iii) counterpart 2D models from 2D slices pooled across orthoviews (Fig. 3a and Methods). We evaluated EmbedSeg3D and EmbedSeg2D[11] as representative of the state-of-the-art instance embedding approach; StarDist3D[52] and StarDist2D[1] as representative of the shape-prior approach; PlantSeg3D[6] and PlantSeg2D as the original and state-of-the-art method for segmenting Ovules and LRP datasets; and trained the latest state-of-the-art 2D Cellpose3 (ref. 53) models with masks generated by u-Segment3D (when suffixed with (u)). The 2D segmentations were then translated to 3D by indirect method u-Segment3D (Methods). In Ovules, u-Segment3D consensus AP and F1 curves matched native 3D models (Fig. 3b). StarDist2D ($AP_{0.5} = 0.7$, $F1_{0.5} = 0.85$) was even better than StarDist3D ($AP_{0.5} = 0.6$, $F1_{0.5} = 0.75$). This highlights 2D-to-3D segmentation as a superior approach for complex 3D shapes, whose 2D cross-sectional shape may be overall simpler, that is, more convex and less elongated. All 2D models outperformed their 3D counterparts for LRP by as much as 0.2 in $AP_{0.5}$, $F1_{0.5}$ for StarDist (Fig. 3c). Most dramatically, for mouse skull nuclei (Extended Data Fig. 6d), u-Segment3D achieved $AP_{0.5} = 0.8$, $F1_{0.5} = 0.9$ with StarDist2D, whereas StarDist3D failed ($AP_{0.5} = 0.05$, $F1_{0.5} = 0.1$). EmbedSeg3D was the best method only for Arabidopsis and nuclei datasets (Extended Data Fig. 6d). The advantage of native 3D models appears primarily in capturing thin and small 3D cells (relative to the image size), that are not well captured in 2D orthoslices. Plotting 2D $AP_{0.5}$ versus 3D $AP_{0.5}$ of its u-Segment3D consensus across datasets and models, u-Segment3D faithfully translates 2D performance (Pearson's $R = 0.79$) and in most cases, with higher 3D $AP_{0.5}$ (Fig. 3d). The 2D AP and F1 curves verify that u-Segment3D parses Cellpose outputs with additional flexibility without affecting performance (Extended Data Fig. 7). Across datasets and models, 3D $AP_{0.5}$ of native 3D segmentations correlates with the corresponding u-Segment3D consensus segmentation (Pearson's $R = 0.57$). Individually, except for *Platynereis* nuclei, we found at least one 2D model where u-Segment3D $AP_{0.5}$ matches or exceeds the native 3D segmentation (Fig. 3e). For *Platynereis* nuclei, the discrepancy was not due to a failure of u-Segment3D to translate 2D performance but insufficient capture of smaller cells in 2D slices. Training an EmbedSeg3D model guided by EmbedSeg2D, u-Segment3D consensus segmentations (Methods) restored smaller cells and boosted $AP_{0.5}$ to 0.85, much closer to native EmbedSeg3D ($AP_{0.5} = 0.95$) (Fig. 3f).

Lastly, u-Segment3D can integrate fewer than three orthoviews in any combination. Depending on the cell packing and shape symmetry, this can reduce computation with results comparable to using all three orthoviews, as demonstrated for LRP (Fig. 3g and Supplementary Movie 6). Due to the microscope or culture conditions, 3D images cannot always be acquired isotropically or be interpolated to be near-isotropic with similar image quality in *x-y*, *x-z* and *y-z* orthoviews. In these cases, application of 2D models, trained solely on the equivalent of in-focus '*x-y*' slices, to *x-z* and *y-z* views may yield worse 'consensus' 3D segmentations. For these cases, Supplementary Note 5,

Extended Data Fig. 8 and Supplementary Movies 7 and 8 demonstrate the application of u-Segment3D to integrate *x-y*-slices only. u-Segment3D then operates similarly to slice-by-slice stitching but retains the advantage of adjusting parameters such as smoothing to interpolate missing segmentations (Supplementary Table 1).

## Recovering 3D protrusions not captured by 2D segmentation

In 3D, cells exhibit a spectrum of protrusive, subcellular surface morphologies. Existing 2D cell segmentation training datasets represent fine-grained morphological motifs such as blebs, lamellipodia, filopodia or villi incompletely, compounded by the spectral bias of neural networks[54], that is, low-frequency modes are learned faster and more robustly than high-frequency modes. Rectifying the bias requires revising the model architecture and additional training on fine-grained higher-quality labels[55]. To circumvent this formidable and often impossible task, u-Segment3D provides a two-stage solution, to be applied in postprocessing after filtering out implausible cells by size and gradient consistency (Fig. 4a(i–iii)). The first stage (Fig. 4a(iv)) is label diffusion, a semi-supervised learning[56] technique to improve adherence to the cell boundaries within a guide image, while enforcing spatial connectivity. Each cell in the input segmentation defines unique 'sources' that simultaneously diffuse to neighbor voxels for $T$ iterations (typically $T < 50$) using an affinity graph combining local intensity differences in the guide image with spatial proximity (Methods). The final segmentation is generated by assigning each voxel to the source with the highest contribution (Methods). The guide image can be the intensity-normalized raw image or any image, enhancing the desired features to be segmented. The second stage uses a guided filter[57] to transfer detailed features from a guide image to the segmentation, constrained to the local neighborhood around the cell border (Methods). Conceptually, this filter is analogous to blending the binary cell mask and the intensities in the corresponding spatial region of the guide image. The neighborhood size may be fixed for all cells or set proportionate to cell diameter. For guided filtering, a good guide image should amplify the subcellular protrusions. We use $I_{guide} = \alpha \cdot I_{norm} + (1 - \alpha)I_{ridge}$, that is, a weighted sum of the normalized input image, $I_{norm}$ and its ridge filter-enhanced counterpart, $I_{ridge}$, (Supplementary Table 1).

Applying this workflow, we recovered missing surface protrusions for cells in tight aggregates while retaining the benefits of the shape prior from Cellpose (Fig. 4a and Supplementary Movie 9). We expected that this would allow us to segment cells imaged with high-resolution light-sheet microscopy even when the membrane staining is inhomogeneous or sparse, situations that challenge thresholding-based techniques[40]. We tested this on single cells with different morphological motifs (Fig. 4b–d(i)). The result captures the global cortical shapes, but cell protrusions only approximately (Fig. 4b–d(ii)). After the postprocessing, all protrusions are recovered (Fig. 4b–d(iii)), with comparable fidelity to binary thresholding (Fig. 4b–d(iv)). However, this segmentation is better suited for surface analysis, as evidenced by a lower genus, $g$, of the extracted surface mesh. We could further recover

**Fig. 5 | u-Segment3D implements parallel computing for tissue-scale segmentation. a**, Schematic of the parallelized gradient descent in overlapped subvolume tiles used by u-Segment3D to facilitate consensus single-cell 3D segmentation in large tissue volumes. **b**, *x-y*, *x-z* and *y-z* mid-slice cross-sections of the fused nuclear (red) and membrane (green) signal channels from multiple biomarkers (Methods) for a CyCIF multiplexed human biopsy sample of metastatic melanoma with white boundaries to delineate the individual cells of a u-Segment3D consensus segmentation using the Cellpose cyto model in each view (left). Zoom-ins of three subregions in *x-y* (black box, regions 1–3), *x-z* (cyan box, regions 4–6) and *y-z* (magenta box, 7–9) cross-sections (right). Image in panel **b** adapted from ref. 41. **c**, u-Segment3D consensus segmentation of nuclei channel (TO-PRO-3, green) using the Cellpose nuclei model and cancer cells (luciferase-GFP, magenta) using the Cellpose cyto2 model in a cleared lung tissue

of a mouse xenografted with YUMM 1.7 melanoma cells (Methods). Left-to-right: merged input volume image, individual segmented nuclei from nuclei channel, cancer luciferase-GFP channel only image showing weak, nonspecific staining (white arrow) compared to the specific positive label of bona fide metastatic colonies (green arrow) and final u-Segment3D representation of metastatic cells post-filtered by mean cell luciferase-GFP intensity. **d**, 3D rendering of the input coCATs volume (left) and u-Segment3D consensus segmentation of salient tissue architecture using the Cellpose cyto model (with model inverted; right). **e**, Mid-slice cross-sections in *x-y*, *x-z* and *y-z* with individual segmentation boundaries in green and overlaid to the input image. **f**, Mid ± 10 *z*-slice *x-y* cross-section with individual segmentation boundaries in green overlaid on the input image. Images in panels **d**–**f** adapted from the neuropil of an organotypic hippocampal brain slice data in ref. 42.

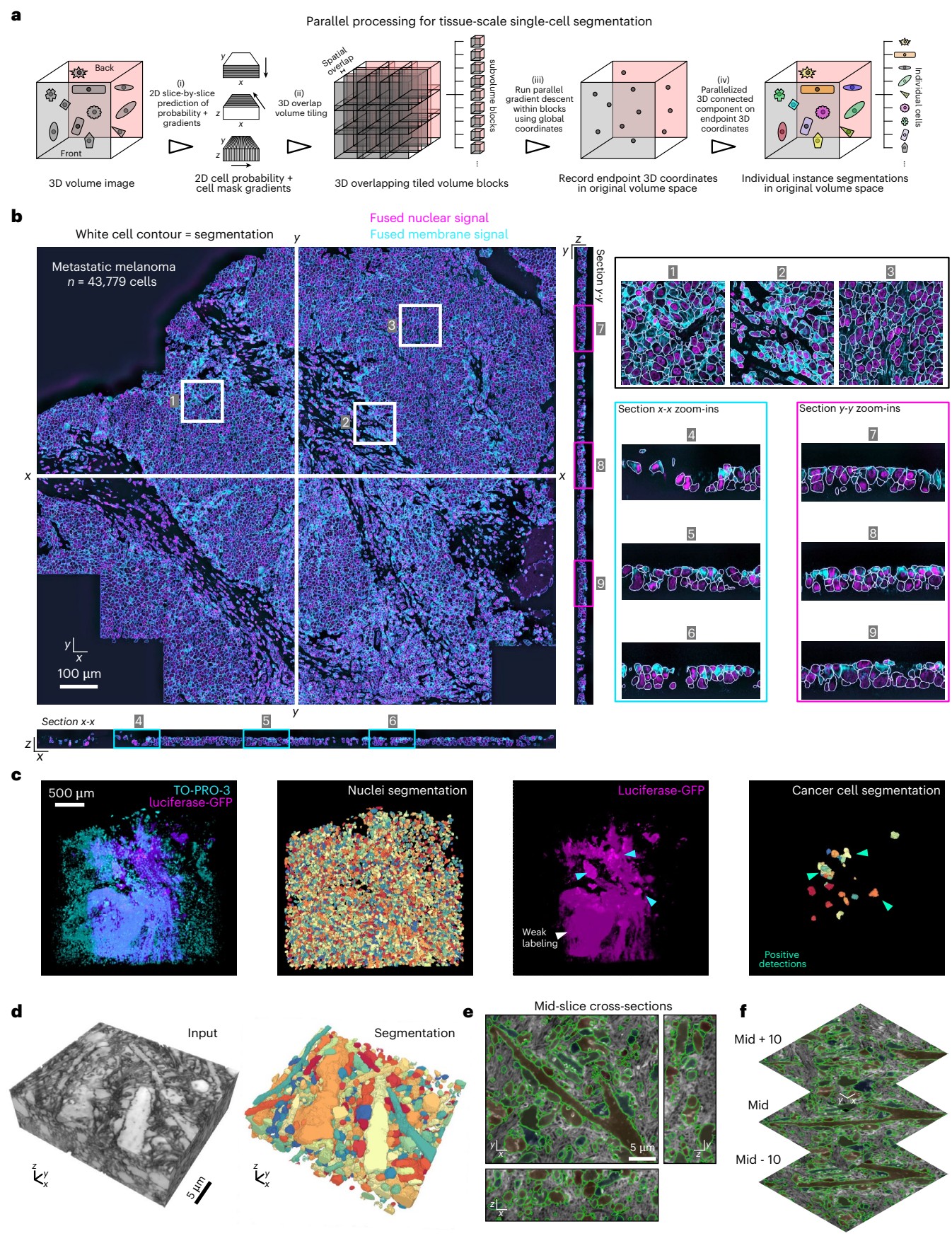

protrusive features on touching cells in a field of view as shown for T cells (Fig. 4e) and zebrafish macrophages (Fig. 4f). Lastly, we tested the segmentation of zebrafish vasculature undergoing angiogenesis (Fig. 4g). The combination of a pretrained Cellpose 2D as shape prior and guided filtering recovered complete, thin sprouting vessels, despite the noisy background and inhomogeneous staining (Supplementary Table 5 and Movie 10).

## 2D-to-3D cell segmentation of tissue by parallel computing

Recent advances in high-resolution thick-section tissue imaging generate image volumes easily containing 10,000's of cells[41]. The time for gradient descent increases with iteration number and the number of foreground pixels (related to image size). To scale up segmentations, we implemented a multiprocessing variant of the 2D-to-3D segmentation pipeline taking advantage of CPU-based cluster computing (Methods). Figure 5a illustrates the key steps: (i) Running pretrained 2D models by GPU inference[4,53] on stacks of orthogonal views of the full volume; (ii, iii) applying gradient descent in parallel in local spatially overlapped subvolumes while generating globally indexed foreground coordinates. Spatial overlap ensures border cells across subvolumes retain the same global attractor, avoiding additional stitching in postprocessing (Methods); and (iv) using an efficient parallelized connected component analysis[58] to generate the full image 3D instance segmentations from advected coordinates. Any further postprocessing is applied in parallel to individual cells. The segmentation of a metastatic melanoma CyCIF multiplexed tissue sample using fused nuclear and membrane signal composites (Methods), imaged with an equivalent isotropic voxel size of 280-nm resolution and dimensions of 194 × 5,440 × 4,792 pixels took ≈2 h for preprocessing and running Cellpose in a slice-by-slice manner in x-y, x-z and y-z; ≈2 h to generate the initial 3D segmentation from 250 gradient descent iterations and using subvolumes of 128 × 256 × 256 pixels with 25% spatial overlap; ≈1 h for size filtering and gradient consistency checking; and ≈2 h for label diffusion refinement. Thus, with a total of 7-h processing time, we extracted the volumes of 43,779 cells on a CPU cluster with 32 physical cores, 72 threads, 1.5 TB RAM and a single A100 GPU (40 GB) for initial 2D model inference. Notably, run on the full volume, the gradient descent alone would be >20× slower. Moreover, the obtained segmentations are free of stitching artifacts (Supplementary Movie 11). Inspecting the zoom-ins of the mid-slices from each of the three orthoviews, our segmentation agrees well with the fused cell nuclei and membrane markers (Fig. 5b and Supplementary Movie 11). Functionally, these segmentations enabled us to improve the accuracy of 3D cell phenotyping and to show, for example, how mature and precursor T cells in metastatic melanoma engage in an unexpectedly diverse array of juxtracrine and membrane–membrane interactions[41].

Axially swept light-sheet microscopy[59] can image cleared-tissue volumes at subcellular resolution over sections as thick as 2 mm, enabling visualization of single cells within micrometastases in lung tissue. u-Segment3D could readily extract all nuclei in the invaded lung tissue stained by TO-PRO-3 and segment luciferase-GFP cancer cells, despite the weak GFP fluorescence (Fig. 5c and Supplementary Movie 12). To generate this segmentation both channels were conjointly input to Cellpose. With the cancer channel alone, individual cells could not be resolved due to insufficient stain contrast between cell boundaries. The nuclei channel helped provide complementary information to localize individual cancer cells. Even so, 2D slice-by-slice segmentations were noisy. u-Segment3D's consensus segmentation contained many spurious segmented cells. However, cells were not catastrophically fragmented, and invalid cells could be readily filtered out by mean luciferase-GFP intensity to identify only the bona fide cancer cells (Extended Data Fig. 9).

Finally, we tested u-Segment3D on an image stack of brain tissue acquired by CATS[42], a recent technique that labels the extracellular space of tissue. Tissue structure based on this label is difficult to discern (Fig. 5d). We hoped Cellpose and u-Segment3D could provide an exploratory tool that 'scans' the volume to generate consensus representations of larger pockets in extracellular space. These spaces are heterogeneous, different in size and morphotype, challenging Cellpose's implicit approach of producing homogeneously sized segmentations. A faithful translation of Cellpose models would fragment the thick, dominant, branching dendrites. By adjusting parameters of the u-Segment3D 2D-to-3D segmentation process, we can alleviate limitations of the 2D model to successfully preserve the multi-scale tissue architecture, both in 3D and in 2D cross-sections (Fig. 5d–f, Supplementary Movie 13 and Tables 1 and 5). Specifically, we increased Gaussian smoothing ($\sigma = 2$) of the reconstructed 3D gradients, and ran suppressed gradient descent with a larger decay ($\tau = 0.25$) to overwrite the splitting of branched structures indicated by Cellpose 2D segmentations. We also set $\sigma = 1.2$ in computing the point density after gradient descent. The slightly larger $\sigma (= 1$ in the rest of the paper) merges attractors separated by a small distance that are part of the same 3D cell, without affecting attractors that genuinely represent distinct cells.

## Discussion

Focused on cell shape reconstruction in biological tissues, we developed a formalism to generate optimal, consensus 3D segmentations from stacks of 2D segmented slices. Our method unifies existing proposals of 2D-to-3D segmentation and shows near-perfect 3D consensus segmentations are achievable across single cells, dense tissue contexts and morphotypes. We refer to the method as 'universal' because it (i) is applicable to diverse morphologies with minimal adjustment of parameters, (ii) exhibits no bias to any particular orthoview, and (iii) encapsulates and generalizes existing stitching and gradient tracing approaches for 2D-to-3D segmentation.

Our work reformulates the ad hoc procedure of stitching discrete label segmentations into a continuous domain problem with well-behaved mathematical properties. This not only enabled true integration of complementary information from orthoviews without bias (Extended Data Fig. 3) but also empowers users to perform well-rationalized fine-tuning of the pipeline rather than empirical trial and error (Supplementary Table 1), which is prohibitive with hard-to-visualize 3D datasets.

These insights led us to develop a general toolbox, u-Segment3D, which accommodates any 2D segmentation method to generate pixel-wise instance segmentation masks in 3D. Extensive validation on public datasets demonstrates u-Segment3D faithfully translates the performance of the 2D segmentation to 3D without further data training. The results are competitive with native 3D segmentation models (Fig. 3). u-Segment3D opens up the development of creative 3D solutions beyond the scope of this paper, including training of individual 2D models for each orthoview, and extension to more than three views. Moreover, u-Segment3D provides fine-tuning and postprocessing methods for further improving 3D segmentations without training, notably recovering fine-grained, cell surface protrusions. We also implemented multiprocessing to enable scalable 2D-to-3D segmentation on CPU clusters. Further speed improvements could be made such as implementing a multi-scale scheme to run u-Segment3D, which we leave for future work.

With the successes of foundation models such as ChatGPT in natural language processing and Segment Anything for object segmentation in image analysis, there is a prevalent notion to learn everything from data, that more data is better and models should be 'turnkey', working directly out-of-the-box or if not, be 'fine-tuned' on more data. In the quest for generality, we must not neglect the value of grounded formalism and robust design. Our analyses provide multiple cautionary tales. First, optimally parsing the outputs of neural network models is just as important as training. Replacing the adaptive density spatial proximity clustering used by Cellpose with connected component analysis notably reduced over-segmentation and boosted performance

on noisy and out-of-distribution datasets (Supplementary Note 1). Second, considering extremal morphotypes and the simpler 1D-to-2D segmentation problem revealed the critical importance of suppressed gradient descent in making 2D-to-3D segmentation applicable to branched and network morphotypes (Supplementary Note 2). Third, by recognizing the spectral bias of neural networks and annotation bias, we developed simple label diffusion and guided filter postprocessing to recover intricate surface morphologies of 3D cells. This enabled us to extend any pretrained neural network to segment high-resolution single cells comparable to state-of-the-art classical methods but with 3D surfaces more suited for mesh-based applications. Our experiments directly question the proposition of training native 3D segmentation models when a viable 2D segmentation model is available. We demonstrate the capacity to 3D segment a most diverse set of cells using a single pretrained 2D model—in this case, Cellpose. With widespread availability of generalist and specialized 2D segmentation models, u-Segment3D paves a way toward accessible 3D segmentation for all, translating time-consuming annotation and training toward more impactful time spent on analyzing the acquired 3D datasets to provide biological insights.

## Online content

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

## Methods

### Datasets

**Validation datasets.** Ten independent public datasets with reference 3D segmentation labels and one dataset collected in-house with 3D segmentations generated with the aid of u-Segment3D were used to evaluate the ability of u-Segment3D to reconstruct 3D objects from ideal and predicted orthogonal *x-y*, *x-z* and *y-z* 2D slices (Fig. 2). Nine of the public datasets with both images and reference segmentations were used to assess the performance of u-Segment3D with pretrained Cellpose models (Extended Data Fig. 3). Details of all datasets are given in Supplementary Table 2.

**Demonstration datasets.** The following datasets were collected largely in-house and segmented using either the pretrained Cellpose cyto, cyto2 or nuclei 2D segmentation models, whichever was observed qualitatively to perform best after using u-Segment3D (if not stated otherwise). Parameter details are provided in Supplementary Table 5.

*Cleared-tissue septin imaging (Fig. 1f).* Two million MV3 melanoma cells expressing SEPT6-GFP[18] were subcutaneously injected into the flank of a NOD/SCID mouse, as previously described[61]. The xenograft tumor was grown until palpable (approximately 3 weeks), at which point the mouse was euthanized and the tumor was excised along with nearby surrounding tissue within ≈5 mm. The tumor was cut into quadrants, cleared using the CUBIC method[62], and melanoma cells within the invasive margin were imaged via ctASLM[59].

*3D epidermal organoid culture (Extended Data Fig. 8a–i). Cell culture.* Human keratinocyte Ker-CT cells (American Type Culture Collection, CRL-4048) were a kind gift from J. Shay (University of Texas (UT) Southwestern Medical Center). Ker-CT cells stably expressing mNeonGreen-tagged keratin 5 to label intermediate filaments were created as previously described[63].

*Epidermal organoid culture.* We adapted an epidermal organoid culture model from existing protocols[63–65]. Polycarbonate filters with 0.4-µm pore size (0.47 cm² area; Nunc, 140620) were placed in larger tissue culture dishes using a sterile forceps. On day 0, a cell suspension of $5 \times 10^5$ keratinocytes expressing keratin 5-mNeonGreen in 400 µl of keratinocyte serum-free media (K-SFM) was added to each filter well, and additional K-SFM was added to the culture dish to reach the level of the filter. On day 10, culture medium was aspirated from above the filter to place the cultures at an air–liquid interface. At the same time, medium in the culture dish was changed from K-SFM to differentiation medium[63]. On day 13, an additional 0.5 mM calcium chloride was added to the differentiation medium in the culture dish. Mature epidermal organoids were processed for imaging on day 20, after 10 days of differentiation at the air–liquid interface. Throughout the procedure, culture media were refreshed every 2 to 3 days.

*Epidermal organoid imaging.* Mature epidermal organoids were transferred to a clean dish, washed three times with PBS, then fixed in 4% paraformaldehyde (Electron Microscopy Sciences, 15713) for 1 h at room temperature. Filters with the organoids were cut out of the plastic housing using an 8-mm punch biopsy tool and inverted onto glass-bottom plates. Throughout imaging, PBS was added one drop at a time as needed to keep each organoid damp without flooding the dish. Organoids were imaged using a Zeiss LSM 880 inverted laser scanning confocal microscope equipped with a tunable near-infrared laser for multiphoton excitation and a non-descanned detector optimized for deep tissue imaging. Images were acquired using an Achroplan ×40/0.8-NA water-immersion objective resulting in an effective planar pixel size of 0.21 µm, and *z*-stack volumes with a 1-µm step size.

*Single-cell tracking challenge datasets[66,67] (Extended Data Fig. 8j–m). MDA231 human breast carcinoma cells (Fluo-C3DL-MDA231)* (Extended Data Fig. 8j). This dataset imaged cells infected with a pMSCV

vector including the GFP sequence, embedded in a collagen matrix, with an Olympus FluoView F1000 microscope with Plan ×20/7 objective lens, sampling rate of 80 min and voxel size of 1.242 × 1.242 × 6.0 µm. We used '01' from the train dataset containing 11 time points. Consensus 3D segmentations were intersected with a foreground mask obtained by Otsu binary thresholding.

*Drosophila melanogaster embryo (Fluo-N3DL-DRO)* (Extended Data Fig. 8k–m). This dataset imaged developing embryos imaged on a SIMView light-sheet microscope[68] with a sampling rate of 30 s, a ×16/0.8 (water) objective lens and a voxel size 0.406 × 0.406 × 2.03 µm. We used 'Cell01' from the test dataset containing 50 time points. We used the pretrained Cellpose 'cyto' model and u-Segment3D to segment the surface for each time point (Supplementary Table 4). Using the binary segmentation, we unwrapped a proximal surface depth using u-Unwrap3D[39].

*HBEC aggregate (Fig. 4a).* The aggregate of transformed HBECs expressing eGFP-Kras^V12 was generated and imaged using meSPIM in Welf et al.[60].

*Single dendritic cells with lamellipodia (Fig. 4b).* The dendritic cells expressing Lifeact-GFP were cultured and imaged in Driscoll et al.[40] and are publicly available from Mazloom-Farsibaf et al.[69].

*Single HBECs with filopodia (Fig. 4c).* The HBEC cells immortalized with Cdk4 and hTERT expression and transformed with p53 knockdown, Kras^V12 and cMyc expression were cultured and imaged in Driscoll et al.[40] and are publicly available from Mazloom-Farsibaf et al.[69].

*Single COR-L23 cells with ruffles (Fig. 4d). Culture.* COR-L23 cells (human large cell lung carcinoma) were resuspended in 2 mg ml⁻¹ bovine collagen (Advanced BioMatrix, 5005) and incubated for 48 h in RPMI 1640 medium (Gibco, 11875093) supplemented with 10% fetal bovine serum (PEAK SERUM, PS-FB2) and 1% antibiotic–antimycotic (Gibco, 14240062). Cells were incubated in a humidified incubator at 37 °C and 5% carbon dioxide.

*Imaging.* Images were acquired with our home-built light-sheet fluorescence microscope system that generates equivalents to dithered lattice light sheets through field synthesis[70]. Briefly, the system uses a ×25 1.1-NA water-immersion objective (Nikon, CFI75 Apo, MRD77220) for detection, and a ×28.6 0.7-NA water-immersion objective (Special Optics, 54-10-7) for illumination. With a 500-mm tube lens, the voxel size of the raw data is 0.104 µm × 0.104 µm × 0.300 µm. The microscope was controlled by the software developed by Coleman technologies. It uses a 64-bit version of LabView 2016 equipped with the LabView Run-Time Engine, Vision Development Module and Vision Run-Time Module (National Instruments). We used a Gaussian light sheet and optimized the light-sheet properties so the confocal length was enough to cover the cell size without sacrificing too much the axial resolution[71]. Typically, the light sheets are about 20 µm long and 1 µm thick. In each acquisition, we optimized the laser power and the exposure time to achieve fast acquisition without introducing too much photo-bleaching. Usually, the time interval for our volumetric acquisition is chosen to be either 5 s or 10 s. The segmented cell is the first time point.

*T cell co-culture (Fig. 4e).* T cells were obtained from blood leukocyte cones purchased from NHS Blood and Transplant, John Radcliffe Hospital, Oxford, United Kingdom. Blood cones were used under the ethical guidelines of the NHS Blood and Transplant. The Non-Clinical Issue division of the National Health Service approved the use of blood leukocyte cones at the University of Oxford (REC 11/H0711/11). Blasted human CD8⁺ T cells, were produced by activating naive T cells isolated from peripheral blood mononuclear cells using anti-CD3/CD28 Dynabeads for 2 days, then rested for 5 days after removing the beads.

Cells were frozen on day 5 of resting and thawed 48 h before use. T cells were grown in complete RPMI 1640 (10% FCS, 1% penicillin–streptomycin, 1% glutamine, 1% HEPES) + 50 U ml⁻¹ of IL-2. For migration-based imaging of multiple T cells, eight-well glass-bottom IBIDI chambers were coated with 1 µg ml⁻¹ of hICAM-1-6xHis linker and hCXCL11 (PeproTech) for 1 h at room temperature, washed, then coated with 1% BSA. Blasted human CD8⁺ T cells (0.5 × 10⁶) were labeled with CellMask DeepRed diluted to a 1× working solution in imaging buffer (that is, colorless RPMI with 1% added penicillin–streptomycin, 1% glutamine, 1% HEPES) for 30 min at 37 °C. Cells and the glass slides were washed and resuspended in pre-warmed (37 °C) imaging buffer. Cells (0.1–0.2 × 10⁶) were gently added to the coated glass slides and left to settle for 30 min before imaging. Cells were imaged using the lattice light-sheet microscope 7 (LLSM7) and control software from Zeiss using the 641-nm laser at 4% power with 4 ms of exposure. A large field of view was used for imaging multiple cells at once, with a complete volume taken every second. Deconvolution was performed using the Zeiss software.

*Zebrafish macrophages (Fig. 4f).* This dataset imaged zebrafish larvae with fluorescent macrophages, labeled with Tg(mpeg1:EGFP), and is publicly available from Daetwyler et al.[49] (Supplementary Table 2).

*Zebrafish vasculature (Fig. 4g).* Zebrafish (*Danio rerio*) embryos, larvae and adults were kept at 28.5 °C and were handled according to established protocols[72,73]. All zebrafish experiments were performed at the larval stage; therefore, the sex of the organism was not yet determined. To visualize the growing vasculature at around 34 h after fertilization, zebrafish larvae expressing the vascular marker *Tg(kdrl:Hsa. HRAS-mCherry)*[74] in a casper background[75] were used. To immobilize the zebrafish larvae for imaging, they were anesthetized with 200 mg l⁻¹ Tricaine (Sigma-Aldrich, E10521)[76] and mounted in 0.1% low melting agarose (Sigma-Aldrich, A9414) inside fluorinated ethylene propylene (FEP) tubes (Pro Liquid, 2001048_E; inner diameter, 0.8 mm; outer diameter, 1.2 mm), coated with 3% methyl cellulose (Sigma-Aldrich, M0387)[77]. The mounted zebrafish larvae were imaged on a custom multi-scale light-sheet microscope with axially swept light-sheet microscopy and controlled by custom Python software (https://github.com/DaetwylerStephan/self_driving_multiscale_control/)[49]. All zebrafish husbandry and experiments have been approved and conducted under the oversight of the Institutional Animal Care and Use Committee (IACUC) at UT Southwestern under protocol number 101805.

*Multiplexed CyCIF tissue (Fig. 5b).* This dataset is described in detail, imaged and published in Yapp et al.[41]. Briefly, the image is of a primary melanoma sample from the archives of the Department of Pathology at Brigham and Women's Hospital. The protocol was adapted from Nirmal et al.[78]. Briefly, a fresh 35-µm-thick FFPE tissue section was obtained from the block and de-paraffinized using a Leica Bond. The region in Fig. 5b was selected and annotated from a serial H&E section by board-certified pathologists as a vertical growth phase. The 35-µm-thick section underwent 18 rounds of cyclical immunofluorescence (CyCIF)[79] over a region spanning 1.4 mm by 1.4 mm and sampled at 140 nm laterally and 280 nm axially. Image acquisition was conducted on a Zeiss LSM 980 Airyscan 2 with a ×40/1.3-NA oil-immersion lens yielding a 53-plex 3D dataset[41]. A custom MATLAB script was used to register subsequent cycles to the first cycle, which was stitched in ZEN 3.9 (Zeiss). The quality of image registration was assessed with Hoechst across multiple cycles in Imaris (Bitplane). For segmentation, multiple channel markers were combined to create fused nuclei and cytoplasmic channels. Hoechst and lamin B1 were combined for nuclei. MHC-II, CD31 and CD3E were combined as a cytoplasm marker to cover all cells including tumor, blood vessels and T cells.

*Cleared-tissue lung micrometastases (Fig. 5c). Cancer growth.* Lung tissue containing a metastatic tumor was harvested from mice injected with YUMM 1.7 GFP-luciferase melanoma cells[80] and grown as previously described[81].

*Lung tissue staining and clearing.* Lung tissue was fixed in 4% paraformaldehyde at 4 °C for less than 24 h and then washed three times with PBS with 0.02% sodium azide for 2 h per wash. The tissue was sliced into 2-mm-thick sections. Tissue slices (~2 mm) were permeabilized and blocked in buffer (0.5% NP40, 10% dimethylsulfoxide, 0.5% Triton X-100, 5% donkey serum, 1× PBS) overnight at room temperature. Tissues were incubated in anti-GFP (1:100 dilution) for 72 h at room temperature in a tube revolver rotator. After incubation, samples were washed with wash buffer (0.5% NP40, 10% dimethylsulfoxide, 1× PBS) three times for 2 h each and then left rotating in wash buffer overnight. Tissues were immersed in the AF488-conjugated secondary antibody solution (1:250 dilution) for 72 h at room temperature. Then, the secondary antibody was removed with wash buffer for at least 2 days changing the solution: the first day three times every 2 h and on the second day refreshed once. Finally, tissues were stained for nuclei with TO-PRO-3 647 (1:500 dilution) in PBS for 24 h at room temperature. Nuclear dye was washed out with wash buffer three times for 10 min each. Lung tissue was cleared using the benzyl alcohol and benzyl benzoate (BABB) clearing protocol (also known as Murray's clear). Lungs were dehydrated in a methanol gradient (25%/50%/75%/100% for 20 min each). The final clearing was achieved with fresh BABB (1:2 dilution). Before starting dehydration, 5 g of aluminum oxide was added to 45 ml BABB and rotated at room temperature for at least an hour to remove peroxides. BABB was kept protected from light and air. The samples were quickly washed with BABB three times, then left standing in fresh BABB for 15 min. Sample BABB was refreshed and left overnight. The sample BABB was refreshed again shortly before imaging.

*Lung tissue imaging.* Lung tissue slices were imaged on a ctASLMv2 (ref. 59) microscope chamber controlled by navigate[82] (https://github.com/TheDeanLab/navigate/). Nuclei were imaged using the TO-PRO-3 647 via illumination with a LuxX 642-nm laser, using 140 mW at 100% laser power and a Semrock BLP01-647R-25 filter in the detection path. Cancer cells were imaged via illumination with a LuxX 488-150, using 150 mW at 100% laser power and a Semrock FF01-515/30-32 bandpass filter in the detection path. Images were acquired with a Hamamatsu ORCA-Flash 4.0 v3 with 200-ms integration time in light-sheet readout mode.

*Ethics.* All mouse experiments complied with all relevant ethical regulations and were performed according to protocols approved by the IACUC at the UT Southwestern Medical Center (protocol no. 2016-101360).

*coCATS labeled volume (Fig. 5d).* We used a coCATS[42] imaging volume recorded with z-STED at near-isotropic resolution in the neuropil of an organotypic hippocampal brain slice published in Michalska et al.[42] (Fig. 3). This volume was downloaded already denoised with Noise2Void.

**UMAP to map morphological diversity of different cell datasets.**
*Morphological features.* Eight features were extracted for each cell based on their 3D reference segmentations.

1. *Volume* - the total number of voxels occupied by the segmented volume.
2. *Convexity* - the ratio of total volume to total volume occupied by the convex hull. The convex hull was computed with 'scipy.spatial.ConvexHull' in Python, using the 3D coordinates of the binary volume.
3. *Major length* - Length of the longest axis of an ellipse fitted to the cell. Computed by 'skimage.measure.regionprops' in Python Scikit-Image as the largest eigenvalue of the inertia matrix.

4. *Minor length* - Length of the shortest axis of an ellipse fitted to the cell. Computed by 'skimage.measure.regionprops' in Python Scikit-Image as the smallest eigenvalue of the inertia matrix.

5. *1 − minor length/major length* - Measure of the extent of elongation with value 0–1. When spherical, minor length = major length and the measure is 0. When highly elongated, minor length < major length and the measure is 1.

6. *Number of skeleton segments* - Number of line segments composing the skeleton of the 3D binary volume.

7. *Number of skeleton nodes* - Number of branch point nodes, where a node is defined as at least three line segments meeting at a junction.

8. *Mean skeleton segment length* - Mean number of voxels in each segment of the 3D binary skeleton.

The 3D binary skeleton was computed using skimage.morphology.skeletonize in Python Scikit-Image. The decomposition of the skeleton into nodes and segments was performed using the Python sknw library (https://github.com/Image-Py/sknw/). Non-dimensionless measurements such as volume were not converted to metric units as only the number of raw voxels is relevant for segmentation.

*UMAP parameters.* The eight morphological features were power transformed to be more Gaussian-like using the Yeo–Johnson method[83] (Python Scikit-learn, sklearn.preprocessing.power_transform). Then, $z$-score normalization was applied to create normalized features. UMAP (using the Python umap-learn library) was used to project the eight features after normalization to two dimensions for visualization (n_neighbors = 15, random_state = 0, spread = 1, metric = 'Euclidean'). The median UMAP coordinate for each dataset was computed by taking the median of the 2D UMAP coordinates of individual cells within the respective dataset. The heat map coloring of the UMAP uses the normalized feature value and the 'coolwarm' color scheme, clipping values to the range [−2,2].

**u-Segment3D.** u-Segment3D is a toolbox that aims to provide methods that require no further training to aggregate 2D slice-by-slice segmentations into consensus 3D segmentations. It is provided as a Python library (https://github.com/DanuserLab/u-Segment3D/). The methods within can be broadly categorized into modules based on their purpose; module 1: image preprocessing; module 2: general 2D-to-3D aggregation using suppressed gradient descent with choice of different 3D distance transforms; and module 3: postprocessing to improve the concordance of segmentation to that of a guide image. Postprocessing helps achieve a tighter segmentation and recover missing local high-frequency surface protrusions.

**Module 1: preprocessing.** Described below are the image preprocessing functions included in u-Segment3D to address the primary problems of intensity normalization, image feature enhancement and uneven illumination that can greatly affect pretrained segmentation models, like Cellpose. Generally, the order of operation or the inclusion/exclusion of a step is dependent on the input data. We found the basic workflow of (i) rescaling to isotropic voxels and resizing for the desired segmentation scale, (ii) uneven illumination correction, adaptive histogram equalization or gamma correction, (iii) deconvolution, and (iv) intensity normalization applied to the 3D raw image, works well for Cellpose models. For Omnipose[5] models, we only used intensity normalization. Any other preprocessing led to worse performance. When both nuclei and cytoplasm channels are both available, we find Cellpose cell segmentations can be much better if both channels are used together as an RGB image (red channel, nuclei; green channel, cytoplasm) instead of nuclei or cytoplasm as a single grayscale image.

**Rescaling to isotropic voxels and resizing to the desired segmentation scale.** Pretrained segmentation models work best when input images contain object types and object sizes reflective of the original training dataset. If images are upscaled to be bigger, segmentation models may be biased toward segmenting physically smaller objects. Correspondingly, if images are downscaled to be smaller, larger objects become enhanced and easier to segment as smaller objects become oversmoothed. Cellpose models are trained at a fixed diameter of 30 pixels and with isotropic '$x$-$y$' images. We find empirically, the u-Segment3D tuning performs best for each orthoview if the input image volume is first rescaled to isotropic voxels and resized using linear interpolation so the desired feature to segment such as cell/vessel results in a peak around 30 pixels (Supplementary Fig. 8). The rescaling and resizing is implemented as one function using the 'scipy.ndimage.zoom' function in Python SciPy, with a Python Dask tile-based accelerated variant for large volumes.

**Contrast-enhancing intensity normalization.** Image intensities are normalized such that 0 is set to the $p_{lower}$ percentile and 1 is the $p_{upper}$ percentile of the image intensity. By default, $p_{lower} = 2$ and $p_{upper} = 99.8$. This contrast enhances the image by clipping out sporadic high intensities caused by saturated probe aggregates and zeroing small, but nonzero background intensities common to fluorescence microscopy.

**Image deconvolution.** For 2D fluorescence microscopy images or anisotropic 3D images, we use blind deconvolution with the unsupervised Wiener–Hunt approach[84] (2D slice-by-slice for 3D) where the hyperparameters are automatically estimated using a Gibbs sampler (implemented using Python Scikit-image, skimage.restoration.unsupervised_wiener). The initial point-spread function is specified as a $15 \times 15$-pixel sum normalized Gaussian ($\sigma = 1$) squared kernel. For 3D light-sheet imaging, we use Wiener–Hunt deconvolution, and our previously published experimental point-spread function[40] as a 'synthetic' point-spread function.

**Model-free uneven illumination correction.** The raw image intensity of 2D or 3D images, $I_{raw}^{ch}$, is corrected for uneven illumination ratiometrically, $I_{correct}^{ch} = \overline{I_{raw}^{ch}} \dfrac{I_{raw}^{ch}}{I_{bg}^{ch}}$ where $\overline{I_{raw}^{ch}}$ denotes the mean image intensity of the input image and $I_{bg}^{ch}$ is an estimate of the uneven background illumination. $I_{bg}^{ch}$ is estimated by downsampling the image by a factor of $ds$, isotropic Gaussian smoothing of $\sigma$ then resizing back to the dimensions of the input image. For 2D images, the downsampling factor does not need to be used as Gaussian smoothing is fast and $\sigma$ is specified as a fraction of the actual image dimension, typically 1/4 or 1/8 is a good starting point. For 3D images a default $\sigma = 5$ is used, with a $ds = 8$ or 16. If segmentation is worse at the higher $ds$, we decrease $ds$ by factor of 2. If $ds = 1$, Gaussian smoothing is applied at the original image resolution. The resultant enhanced image should have even illumination with minimal artifactual enhancement of border background intensities. A more sophisticated background correction is the N4 bias correction available in SimpleITK, originally developed for magnetic resonance imaging and has been successfully applied to 3D cleared-tissue imaging[20].

**Adaptive histogram equalization.** Contrast limited adaptive histogram equalization (Python Scikit-image, skimage.exposure.equalize_adapthist) can also be used as an alternative to our model-free uneven correction. The image is divided into nonoverlapping tiles and the pixel intensity is histogram equalized within each tile. While this obtains good results, we find that the method is computationally more memory intensive and slower for large 3D volumes if the size of individual tiles is required to be small, thus increasing the overall number of tiles

to be processed. However, there is less artifact for originally low-valued intensities compared to our faster ratiometric method.

**Gamma correction.** Transforms the input image, $I_{in}$, pixel-wise, raising the intensity to a power $\gamma$ (float between 0 and 1) so that the output image $I_{out} = I_{in}^{\gamma}$ after scaling the image pixel intensity linearly in the range of 0 to 1. Used to nonlinearly amplify low-intensity pixels to create a more uniform illumination for segmentation that is computationally inexpensive.

**Ridge, vessel-like feature enhancement.** Neurites, tubes, vessels and edges of cell surface protrusions all represent ridge-like structures that are both thin and long or exhibit high-curvature and tortuous morphologies that are often only weakly stained and visualized from raw image intensities. Ridge image filters use the eigenvalues of the Hessian matrix of image intensities to enhance these ridge-like structures assuming the intensity changes perpendicular to but not along the structure. Many ridge filters have been developed. u-Segment3D uses the Meijering[85] filter (Python Scikit-image, skimage.filters.meijering), which enhances ridge image features by pooling the maximum filter responses from multiple Gaussian $\sigma$. We observed empirically good performance for a diverse range of objects including vessels and cells, without requiring additional hyperparameters and hyperparameter tuning unlike Frangi filtering[86].

**Semiautomated diameter tuning for pretrained Cellpose models.** The tuning process is illustrated in Supplementary Fig. 9a. Given a 2D image, the Cellpose outputs—the non-normalized cell probability, $p$ and predicted 2D gradients in x- ($\nabla_x \Phi$) and y- ($\nabla_y \Phi$) directions—are computed. $p$ is clipped to a range of [−88.72, 88.72] to avoid overflow for IEEE float32 and normalized to a value in the range [0,1], $p \leftarrow \frac{1}{1+e^{-p}}$. These ouputs define the pixel-wise contrast score, $w \cdot \{\sigma_{\mathcal{N}}(\nabla_x \Phi) + \sigma_{\mathcal{N}}(\nabla_y \Phi)\}$, where $w$ is a pixel-wise weight. We set $w = p$ but observed little difference if $p = 1$ for Cellpose models. $\sigma(\cdot)$ is the local standard deviation at each pixel, computed over the local pixel neighborhood of width $P \times P$ pixels. The mean score over all pixels, $\frac{1}{N}\sum w \cdot \{\sigma_{\mathcal{N}}(\nabla_x \Phi) + \sigma_{\mathcal{N}}(\nabla_y \Phi)\}$, is computed over a range of equisampled diameters, for example, 15 to 120 at increments of 2.5. A centered moving-window average (default window size of 5) using symmetric padding at edges is then applied to smoothen the diameter versus score plot. Prominent peaks in this plot highlight potential segmentations at different size scales. The more sizes, the more peaks. Users may use this plot to inform the setting of a preferred diameter. For automatic operation, the diameter with highest contrast score is used. The neighborhood size acts like an attention mechanism (Supplementary Fig. 9c). The larger the neighborhood size, the more the segmentation result corresponds to larger objects being favored. If there is no larger salient segmentation, the optimal diameter will be the same as that found with a smaller neighborhood size.

**Semiautomated cell probability thresholding for pretrained Cellpose models.** For out-of-distribution images and noisy input images, we observed that pretrained Cellpose 2D models can perform well using an appropriate threshold for cell probability combined with u-Segment3D's gradient descent and spatial connected component analysis (Supplementary Fig. 4f versus Extended Data Fig. 8j). The choice of threshold is particularly important. If the threshold is too high, there is no continuous path for the gradient descent, resulting in over-segmentation. Therefore, it is better to veer on the side of caution and use a lower threshold to get a more connected foreground binary. However, if the threshold is too low, the foreground binary will be larger than the region with predicted gradients. The additional foreground voxels have zero gradients and may segment as erroneous, extraneous cells. To automate the threshold, u-Segment3D applies multi-class

Otsu thresholding to the normalized cell probability ($p \in [0,1]$) output of Cellpose, $p \leftarrow \frac{1}{1+e^{-p}}$. u-Segment3D further performs morphological closing to fill small holes. If only one object is known to be present, further operations such as extracting the largest connected component and binary infilling can be conducted. The default Otsu thresholding used in this paper is 2-class. If the segmentation partially captures the cells, we use 3-class Otsu and the lower of the two thresholds. Vice versa, if too much area is segmented, we use 3-class Otsu and the higher of the two thresholds. Optionally, we cast the threshold to the nearest decimal point, rounding down (threshold $\leftarrow \lfloor$ threshold $\times 10\rfloor/10$ where $\lfloor\cdot\rfloor$ is the floor operator).

**Module 2: gradient descent and distance transforms to assemble 2D slice-by-slice segmentation stacks into a 3D consensus segmentation.** Methods in this module are used to implement the core 2D-to-3D segmentation algorithm outlined in Fig. 1e. If 2D segmentations are not provided as a normalized cell probability (0–1) and 2D gradients in the manner of Cellpose[4], then a 2D distance transform is specified to generate the necessary 2D gradients for consensus 3D segmentation.

**2D distance transforms.** u-Segment3D categorizes the distance transforms according to whether the limit or attractor of propagating points using gradient descent over an infinite number of steps is implicitly or explicitly defined (Supplementary Fig. 1). Explicitly defined transforms are further categorized by the type of attractor: a single fixed point source or a source that comprises a set of points.

u-Segment3D implements distance transforms, $\Phi$, that are solutions within the cell interior, of the Eikonal equation ($\|\nabla\Phi\|^2 = 1$, which gives the shortest geodesic solution) or Poisson's equation ($\nabla^2\Phi = -1$, which gives a smooth harmonic solution). The Eikonal equation finds the shortest time of propagation for a point. Poisson's equation can also be viewed as solving the shortest time of propagation but with the additional constraint of minimizing curvature, yielding smoother solutions.

**Implicit attractor distance transforms.** With only the boundary condition $\Phi = 0$, the Eikonal and Poisson equations conceptually propagate a wave inward symmetrically from the cell boundaries. The limit solution is the definition of the medial-axis skeleton, the locus of the centers of all inscribed maximal spheres of the object where these spheres touch the boundary at more than one point[43,44,87].

*EDT*. Solves the Eikonal equation using fast image morphological operations. u-Segment3D uses the memory and speed optimized implementation in the Python 'edt' package released by the Seung Lab (https://github.com/seung-lab/euclidean-distance-transform-3d/).

*Poisson distance transform (or diffusion)*. Solves the Poisson equation for each cell shape using LU decomposition (Python SciPy, scipy.sparse.linalg.spsolve). We parallelize the solving for all cells in an image using the Python Dask library.

**Explicit attractor distance transforms.** The implicit attractor solves the equations everywhere in the cell interior. The explicit attractor variants modify the equations to have different source terms (right-hand side of equation) in different parts of the cell interior. For the Eikonal equation, $\Phi = 0$ at the cell boundary and outside, non-source points obey $\|\nabla\Phi\|^2 = 1$, while source points act as obstacles with vanishing speed, so that $\|\nabla\Phi\|^2 = 0$. For the Poisson equation, $\Phi = 0$ at the cell boundary and outside, non-source points obey the Laplace equation, $\nabla^2\Phi = 0$, while source points obey $\nabla^2\Phi = -1$.

(i) **Point sources**. A single interior point is designated as a point source. u-Segment3D finds the interior point with EDT value greater than the percentile threshold (default: 10th percentile) nearest the median coordinate of all points.

*Eikonal equation solution (geodesic centroid distance)*. At the interior point, $\|\nabla\Phi\|^2 = 0$. The modified equations are solved using the fast marching method[88], with the constraint enforced by the Python scikit-fmm library using masked arrays. Central first-order differences are used to compute the unit-normalized 2D gradient.

*Poisson equation solution (Poisson or diffusion centroid distance)*. Only at the interior point, $\nabla^2\Phi = -1$. The modified equations are solved using LU decomposition as before. To apply power transformation with exponent $p > 0$, the minimum is first subtracted from $\Phi$ to ensure positivity, $\Phi^p := (\Phi - \Phi_{min})^p$. Central first-order differences are used to compute the respective unit-normalized 2D gradient.

(ii) **Point set sources**. Any number of interior points are designated as point sources. u-Segment3D computes the 2D medial-axis skeleton as the point set attractor. The binary skeleton of a binary cell image is computed by iteratively removing border pixels over multiple image passes[89] (Python Scikit-image, skimage.morphology.skeletonize). This raw result often produces skeletons with extraneous branches that are too close to a neighboring cell. To improve the skeleton quality, the binary image is first Gaussian filtered with $\sigma = 3$ pixels (default), rebinarized by mean value thresholding and then skeletonized.

*Eikonal equation solution (geodesic centroid distance)*. For all points part of the skeleton, $\|\nabla\Phi\|^2 = 0$. The modified equations are then solved using the fast marching method[88] as above with central first-order differences for computing the unit-normalized 2D gradient. The gradients for all points part of the skeleton are set to zero to enforce the limiting behavior under gradient descent.

*Poisson equation solution (Poisson or diffusion centroid distance)*. For all points part of the skeleton, $\nabla^2\Phi = -1$. The modified equations are solved using LU decomposition as above with central first-order differences for computing the unit-normalized 2D gradient. The gradients for all points part of the skeleton are set to zero to enforce the limiting behavior under gradient descent.

**Content-based averaging function, $F$.** u-Segment3D fuses 3D volume images, $I^i$ from $i = 1, \dots, N$, multiple views using a content-based average function, $F$, with pixel-wise weighting of the contribution of each view $i$ given by the inverse local variance, $\sigma_{\mathcal{N}}^i$, evaluated over an isotropic neighborhood, $\mathcal{N}$, of width $P$ pixels,

$$I_{\text{fuse}} = \frac{\sum_{i=1}^{N} \frac{1}{\sigma_{\mathcal{N}}^i + \alpha} I^i}{\sum_{i=1}^{N} \frac{1}{\sigma_{\mathcal{N}}^i + \alpha} + \varepsilon}$$

with $\alpha$ acting as a pseudo count. If $\alpha$ is small, $\sigma_{\mathcal{N}}^i$ dominates. If $\alpha$ is large, $\sigma_{\mathcal{N}}^i$ has little effect and all views are equally weighted. $\varepsilon$ is a small value ($10^{-20}$) to prevent infinity. For a neighborhood of width $P = 1$ pixels, $F$ is equivalent to the simple mean used by Cellpose[4] (Supplementary Fig. 2a). Compared to potentially more accurate approaches such as solving the multi-view reconstruction problem[90], entropy-based averaging[91] or using Gaussian filters[92], the proposed $F$ can be implemented more efficiently with uniform filters.

**Fusing normalized 2D cell probabilities (0–1) from orthoviews and binary thresholding.** Stacked normalized 2D cell probabilities (0–1) are fused using the content-based averaging function, $F$, above with neighborhood $P = 1$ (default) pixels, the same as the fusion of 2D gradients below. For Cellpose models, the raw cell probability outputs, $p$, are first clipped to the range [−88.72, 88.72] to prevent underflow/overflow in IEEE float32 and transformed, $p \leftarrow \frac{1}{1+e^{-p}}$, before fusing. The methods for yielding only 2D segmentations are: (i) fuse

using the binary then apply appropriate Gaussian filtering to smooth; (ii) use the intermediate cell probability image, which is always available for deep learning methods; or (iii) generate a proxy cell probability image, for example, using a normalized EDT with values of 0–1.

**Fusing 2D gradients from orthoviews.** Stacked 2D gradients from x-y, x-z and y-z are pre-filtered with an isotropic Gaussian of $\sigma_{\text{pre}} = 1$. The fused 3D gradients combine three separate fusions: the fusing of the x-component from x-y and x-z views, the y-component from x-y and y-z views and the z-component from x-z and y-z views. The constructed 3D gradients are post-filtered with $\sigma_{\text{post}} = 1$ (default) and unit-length normalized. The greater $\sigma_{\text{post}}$ is, the greater the regularization effect, reducing the number of attractors and preventing over-segmentation. This is helpful when using pretrained Cellpose models to segment cells that are larger and more branched than the majority of cells in an image. However, a large $\sigma_{\text{post}}$ can also merge smaller cells. For fusion, we use $\alpha = 0.5$ and in general $P = 1$ for $F$ to maximize segmentation recall and use postprocessing to remove any erroneous segmentations. Larger $P$ improves segmentation precision but may miss cells with lower contrast. These settings are generally not modified from the default. Preventing over-segmentation can be more controllably carried out by adjusting the temporal decay parameter in the gradient descent (see below and Supplementary Table 1.).

**Gradient descent.** Given the reconstructed 3D gradients, $\nabla\Phi$, gradient descent is applied to the set of all foreground image coordinates, $\{(x_n, y_n, z_n)\}$. The iterative update for gradient descent with momentum in 3D for iteration number, $t = 0, \dots, T$, with $T = 250$ defining the total number of iterations implemented by u-Segment3D, is

$$(x_n^t, y_n^t, z_n^t) \leftarrow (x_n^{t-1}, y_n^{t-1}, z_n^{t-1})$$
$$-\eta \frac{(\delta \cdot \nabla\Phi(x_n^{t-1}, y_n^{t-1}, z_n^{t-1}) + \mu \cdot \nabla\Phi(x_n^{t-2}, y_n^{t-2}, z_n^{t-2}))}{\delta + \mu}$$

where $\mu$ is the momentum parameter governing the extent of influence of the previous gradient, ranging from 0 to 1 (default $\mu = 0.95$), and $\delta > \mu$ is the weighting of the current gradient and the step size. $\delta$ is usually fixed with $\delta = 1$ and only $\mu$ is adjusted. $\mu = 0$ recovers the standard gradient descent iteration. Nearest-neighbor interpolation is used for computational efficiency. As such, $(x_n^t, y_n^t, z_n^t)$ is always integer valued. $\eta$ defines the step size and varies as a function of the iteration number,

$$\eta = \frac{\delta}{1 + t \cdot \tau}$$

where $\tau \in \mathbb{R}^+$ is a floating point number that controls the step-size decay[5]. The greater $\tau$ is, the less the points are propagated. When $\tau = 0$, the step size is constant $\eta = \delta$.

**Parallelized gradient descent implementation using subvolumes.** For the CyCIF segmentation (Fig. 5a,b), the image volume and associated data: the foreground binary and reconstructed 3D gradient map were tiled with subvolumes of (256, 512, 512), with 25% spatial overlap between adjacent subvolumes. Within each subvolume, 3D gradient descent with momentum ($\mu = 0.98$) and gradient decay ($\tau = 0.0$) were run for 250 iterations, and a step size $\delta = 1$ was used to propagate foreground coordinates toward their attractor. At each iteration, coordinates were clipped to lie within the bounds of the subvolume. The final coordinates were converted to global image coordinates by adding the offset position of the subvolume.

$$(x_n^{\text{global}}, y_n^{\text{global}}, z_n^{\text{global}}) = (x_n^{\text{subvolume}}, y_n^{\text{subvolume}}, z_n^{\text{subvolume}})$$
$$+ (x_{min}^{\text{subvolume}}, y_{min}^{\text{subvolume}}, z_{min}^{\text{subvolume}})$$

where $(x_{min}^{subvolume}, y_{min}^{subvolume}, z_{min}^{subvolume})$ is the corner corresponding to the (0, 0, 0) origin of the subvolume. Binary thresholding and optimized connected component analysis[58] is applied to the global coordinates, pooled from all subvolumes, to generate a single globally consistent 3D segmentation. We find this procedure requires no further stitching across subvolumes provided the subvolume size covers a few cells and the spatial overlap enables the attractor basin to be represented in adjacent subvolume tiles.

**Image-based connected component analysis for identifying the unique number of cell centers for instance segmentation.** The method is depicted in Supplementary Fig. 4 for a 2D image and described here for a 3D image. Step (i), the final ($t = T$) gradient descent advected foreground coordinate positions $\{(x_n^{t=T}, y_n^{t=T}, z_n^{t=T})\}$, is rasterized onto the image grid by flooring, that is, $\{(\lfloor x_n^{t=T}\rfloor, \lfloor y_n^{t=T}\rfloor, \lfloor z_n^{t=T}\rfloor)\}$, and clipping coordinate values to lie within the bounds of the $L \times M \times N$ image volume that is $0 \le \lfloor x_n^{t=T}\rfloor \le L - 1$, $0 \le \lfloor y_n^{t=T}\rfloor \le M - 1$, $0 \le \lfloor z_n^{t=T}\rfloor \le N - 1$ in Python. Step (ii), the number of points at each voxel position is tabulated, each point contributing +1 count. Step (iii), the counts image is Gaussian filtered with $\sigma = 1$ as a fast approximation to the Gaussian kernel density to produce a point density heat map, $\rho(x,y)$ for 2D and $p(x,y,z)$ for 3D. This step accounts for gradient errors and spatially connects points into a cluster in a soft manner. The larger the width of the Gaussian filter $\sigma$ the more nearby points will be grouped into the same cluster. This is helpful when segmenting branching structures as highlighted in the coCATs labeled example in Fig. 5d–f. (iv) The density heat map is sparse, allowing all unique clusters to be identified using a mean threshold with an optional tunable offset specified as a constant multiplicative factor, $k$, of the standard deviation (std) of $\rho$, threshold = mean($\rho$) + $k \cdot$ std($\rho$). For examples in this paper, we set $k = 0$. Image-based connected component analysis is then applied to the binary segmentation of $\rho$ to create the distinct spatial cluster segmentation image at $t = T$; $L^{t=T}(x,y)$ for 2D and $L^{t=T}(x,y,z)$ for 3D. Each foreground coordinate, $\{(x_n^{t=T}, y_n^{t=T}, z_n^{t=T})\}$, is then assigned to a cluster ID by image indexing and the final cell segmentation is computed by mapping the labels of points to their initial positions, $\{(x_n^{t=0}, y_n^{t=0}, z_n^{t=0})\}$. For connected component analysis, u-Segment3D uses the optimized, parallel implementation developed by the Seung Lab (https://github.com/seung-lab/connected-components-3d/)[58].

**Module 3: postprocessing the 3D consensus segmentation.** Described below are the implemented postprocessing methods that can be applied to the initial consensus 3D segmentation (module 2). The recommended sequential u-Segment3D workflow is: (i) the removal of implausible predicted cells involving (i(a)) the removal of predicted cells below a user-specified size limit (in voxels), (i(b)) the removal of segmented cells with recomputed 3D gradients inconsistent with that of the 2D-to-3D reconstructed gradients and (i(c)) the removal of cells that are statistically too large (volume > mean(volumes) + $k \cdot$ std(volumes) where $k$ is a multiplicative factor, default $k = 5$); (ii) label diffusion to smooth, enforce the spatial connectivity constraint of segmentation and propagate the initial consensus segmentation to better adhere to the desired features within the given guide image; (iii) guided filter to refine and transfer missing local, high-frequency subcellular structures from a guide image to the segmentation.

The guide image used in label diffusion and guided filtering does not need to be the same as the raw image. Generally, it is a version of the raw image where desired cellular features are enhanced.

(i)     **Removal of implausible predicted cells**.
    (i(a))  *Removal of predicted cells that are too small.* Volumes of individual cells are computed as the number of voxels. The IDs of all cells with a volume less than the

user-specified threshold (default, 200) are removed by setting their voxels to 0. Additionally, each cell is checked whether they comprise multiple spatially disconnected components. If so, only the largest component is retained as each segmented cell should be spatially contiguous.

(i(b))  *Removal of predicted cells is inconsistent with the reconstructed 3D gradients.* The reconstructed 3D gradients, $\nabla\Phi_{3Dsegmentation}$ are computed from $x$-$y$, $x$-$z$ and $y$-$z$ views of the assembled consensus 3D segmentation. The mean absolute error with the predicted 3D gradients, $\nabla\Phi_{3D}$, used as input in gradient descent is computed per cell, $MAE_{cell} = $ mean $(|\nabla\Phi_{3Dsegmentation} - \nabla\Phi_{3D}|)_{cell}$. If $MAE_{cell}$ > user-defined threshold (default 0.85 for $\sigma_{post} = 1$). If the post-Gaussian filter $\sigma_{post}$ used when fusing gradients from orthoviews is >1, the threshold may need to be relaxed, that is, threshold > 0.85.

(i(c))  *Removal of predicted cells that are statistically too large.* Ratiometric uneven illumination correction may unduly amplify background at the borders of the image, potentially resulting in the erroneous segmentation of very large background regions. Also in dense tissue, when staining is inhomogeneous and weak, multiple closely packed cells may be segmented as one in the initial 2D segmentation. Assuming cell volumes are approximately normally distributed, we filter out improbably large cells by using the mean and standard deviation of all segmented cell volumes to set a cutoff. Only cells with a volume smaller than mean(volume) + $k \cdot$ std(cell volumes) are then retained, with $k = 5$ as default.

(ii)    **Label diffusion to smooth and propagate cell segmentations with spatial connectivity constraint**. Labelspreading[56] is a semi-supervised learning method developed to infer the label of objects in a dataset given the labels of a partial subset of the objects. It works by diffusing labels after one-hot encoding on an affinity graph between objects. u-Segment3D adapts this algorithm for cell segmentation. To be computationally scalable for large cell numbers, for each cell mask, $M_i$, a subvolume, $V_i$, is cropped with the size of its bounding box isotropically padded by a default of 25 voxels. Every label in $V_i$ is one-hot encoded to form a label vector $L \in \mathbb{R}^{N \times p}$ where $N$ is the total number of voxels and $p$ the number of unique cell IDs, including background. We then construct an affinity matrix, $A$, between voxels as a weighted sum (weight, $\alpha$) of an affinity matrix constructed from the intensity differences in the guide image, $I$, between eight connected voxel neighbors, $A_{intensity}$, and another based solely on the voxel spatial connectivity, $A_{Laplacian}$:

$$A = \alpha A_{intensiy} + (1 - \alpha)A_{Laplacian}$$

and

$$A_{intensity}(i,j) = \begin{cases} e^{-D(i,j)_{intensity}^2 / (2\mu(D(i,j)_{intensity})^2)} & i \ne j \\ 1 & i = j \end{cases}$$

$$A_{Laplacian}(i,j) = \begin{cases} e^{-D(i,j)_{Laplacian}^2 / (2\mu(D(i,j)_{Laplacian})^2)} & i \ne j \\ 1 & i = j \end{cases}$$

$D(i,j)_{intensity}$ is the pairwise absolute difference in intensity values between two neighboring voxels $i$ and $j$. $D(i,j)_{Laplacian}$ is the graph Laplacian with a value of 1 if a voxel $i$ is a neighbor of voxel $j$, and 0 otherwise. $\mu(D)$ denotes the mean value of the entries of the matrix $D$. The iterative label diffusion is then

$$z \in \mathbb{R}^{N \times p}$$

$$z^{t=0} = \mathbf{0}$$

$$z^{t+1} \leftarrow (1 - \gamma) A z^t + (\gamma) L$$

where $t$ is the iteration number, $\mathbf{0}$, the empty label vector and $\gamma$ is a 'clamping' factor controlling the extent by which the original labeling is preserved. The final $z$ is normalized using the softmax operation, and argmax is used to obtain the final cell IDs. The refined cell mask, $M_i^{refine}$ for cell ID $i$ includes all voxels where the final $z$ is assigned to the same cell ID $i$. Multiprocessing is used to refine all individual cells in parallel. It is recommended to set the parameters per dataset, depending on the extent of correction required. We typically start with a conservative $\alpha = 0.5$, $\gamma = 0.75$, and run the propagation for 25 iterations, and adjust accordingly. The guide image, $I$, is usually the normalized input image (after any preprocessing) to the 2D segmentation but can be any processed image that enhances the desired cell features. For additional speed, particularly for tissue, we would typically treat each cell mask, $M_i$, independently as binary without considering the multi-label setting of jointly refining neighboring cell masks in the cropped volumes.

(iii) **Guided image filtering to recover missing high-frequency features and subcellular protrusions**. The guided filter[57] is a filter that can be implemented in linear time, to efficiently transfer features in a guidance image, $I$, to the input image to be filtered, $P$. Setting $I$ to the ridge-filtered input image that enhances high-frequency cellular protrusion and vessel features, and $P$ to be the binary mask of cell $i$, the resulting filtered output $Q$ is a 'feathered' binary, where transferred image features appear as an alpha matte locally around the mask boundaries. The radius of the boundary that is refined is controlled by a radius parameter, $r = 35$ voxels (by default), and the extent of transfer by a regularization parameter, $\epsilon = 1 \times 10^{-4}$. We find the binary mask can encapsulate the cell relatively coarsely. The stronger the features are enhanced in $I$ the more prominent the transferred structure. $Q$ is then rebinarized using multilevel Otsu thresholding. Typically, we use the two-class binary Otsu. As for label diffusion, guided filtering is applied to cropped subvolumes, $V_i$ of individual cells, with the size of individual bounding boxes isotropically padded by a default 25 voxels. For computational efficiency, for touching cells, we perform the guided filter segmentation independently for each cell and mask out spatial regions occupied by surrounding cell IDs. More accurately, we could obtain the guided filter response for all cell IDs in the subvolume and use argmax to define the maximum filter response. Multiprocessing is used to perform the guided filter refinement to all cells in parallel. The radius $r$ sets the maximum protrusion length that can be recovered. If the cell density is high, it may not be possible to adjust $r$ to recover long protrusions without erroneously incorporating features of neighboring cells. Nevertheless, the guided filter result may assist the application of matching algorithms or serve as an improved seed image for watershed algorithms in further specialized downstream processing. Alternatively, it can be applied repeatedly to the result of the previous, to capture extended protrusions.

## Semiautomatic tuning of diameter parameter in Cellpose models

The process is illustrated in Supplementary Fig. 9a. for 3D and described below.

**Determining the optimal diameter for 2D images.** Given a pixel neighborhood size with isotropic width, $P$ pixels, we conduct a parameter scan of diameter $= [d_{low}, d_{high}]$ (typically $d_{low} = 10$, $d_{high} = 120$) at equal increments of 2.5 or 5. For each diameter, a contrast score is computed measuring the 'sharpness' of the Cellpose model predicted 2D $x$- and $y$-gradients ($\nabla_x \Phi$ and $\nabla_y \Phi$, respectively) and optionally the normalized cell probability map, $p$ (0–1).

$$\text{Contrast score}(d) = \frac{1}{N} \sum w \cdot \{\sigma_{\mathcal{N}}(\nabla_x \Phi) + \sigma_{\mathcal{N}}(\nabla_y \Phi)\}$$

where $N$ is the total number of image pixels, $w$ is a pixel-wise weight set to be $p$ and $\sigma_{\mathcal{N}}(I)$ is the pixel-wise local standard deviation of the image $I$ evaluated over the isotropic local neighborhood of width $P$ pixels. $p$ is computed from the unnormalized raw cell probabilities after clipping to range (−88.72, 88.72; to prevent overflow or underflow in IEEE float32) by applying the transformation, $p \leftarrow \frac{1}{1+e^{-p}}$. The result is a contrast score function of $d$. A centralized moving average of 5 (if diameter increment is 2.5) or 3 (if diameter increment is 5) is applied to smooth the contrast score function. The diameter $d$ that maximizes the contrast score is used as the optimal diameter, $d_{opt}$, in the Cellpose model. We generally observe no difference in $d_{opt}$ between $w = 1$ and $w = p$ for Cellpose models.

**Determining the optimal diameter for 3D volume.** If cells exhibit large size variations in a slice-by-slice manner, the optimal diameter determination for 2D should be applied slice by slice (Extended Data Fig. 8). For large numbers of slices this is slow. As a compromise, we find good performance for many datasets, if we set an optimal diameter using a single representative 2D slice and apply it to the 3D volume for each orthoview. This representative 2D slice is set automatically using (i) the most in-focus slice as determined by the highest mean sobel magnitude, (ii) the slice with highest mean intensity, (iii) the mid-slice, or (iv) a user-defined threshold.

## Compared 2D-to-3D stitching methods

**3DCellComposer[36].** We used 3DCellComposer version 1.2.1. We modified the 'process_segmentation_masks' function in https://github.com/murphygroup/3DCellComposer/blob/main/run_3DCellComposer.py/ to allow for stitching from only $x$-$y$, $x$-$z$ and $y$-$z$ cell mask inputs. We did not use the unsupervised segmentation metric to set the optimal Jaccard index overlap for stitching. Instead, we performed a search, stitching for Jaccard index at 0, 0.1, 0.2, 0.3 and 0.4, the same as hardcoded in the code, and report the AP curve with highest mean value.

**CellStitch[38].** We used CellStitch version 1.0.0. We used the 'full_stitch' function from the official code repository to stitch input $x$-$y$, $x$-$z$ and $y$-$z$ cell segmentation masks in https://github.com/imyiningliu/cellstitch/ as illustrated in the provided example notebooks (https://github.com/imyiningliu/cellstitch/tree/main/notebooks/). This function is parameterless.

## Other tested segmentation methods

**Cellpose 2D models.** Cellpose3 was released during revision of the paper. All experiments except for Fig. 3, and its associated Extended Data Figs. 6 and 7 used Cellpose version 2.3.dev7+g03e02bc and pretrained 'cyto' and 'cyto2' models. Figure 3 and Extended Data Figs. 6 and 7 used Cellpose3 (version 3.0.8) and trained 'cyto3' Cellpose models.

**Cellpose 3D mode with pretrained models.** These experiments used Cellpose version 2.3.dev7+g03e02bc. We ran pretrained 2D Cellpose models in 3D mode to generate 3D segmentations by setting do_3D = True. As we find this mode prone to over-segment and Cellpose 3D only allows one diameter for all orthoviews, we used the largest

diameter inferred by our contrast score function. Models were run twice. The first time was to obtain the raw, unnormalized cell probability image, which was then used to determine the binarization threshold. We then ran a second time using the determined threshold to generate the 3D segmentation. We then additionally removed all cells with volume < 2,500 voxels to get a segmentation that maximizes the measured average AP.

**Omnipose 3D.** We used Omnipose version 0.3.5.dev10+ge22262b. We ran the pretrained 'plant_omni' model, modifying the script in the documentation (https://omnipose.readthedocs.io/examples/mono_channel_3D.html). This model operates on the raw image downsampled by a factor of 1/3 in all dimensions and does not reinterpolate the raw image to isotropic resolution. No other preprocessing was used. We found the raw output to predict many small objects leading to an artificially low AP when compared with qualitative assessment. Therefore, we removed all objects with volume <2,500 voxels to get a segmentation with maximum average AP. Specifying a higher size cutoff led to lower mean AP, as it detrimentally affected lateral primordial images comprising largely smaller cells in the dataset.

**Cellpose 3D mode with Omnipose trained 'plant_cp' model.** We used Omnipose version 0.3.5.dev10+ge22262b. We ran the 2D pretrained Cellpose 'plant_cp' model using the same function call as the example in the Omnipose documentation for plant_omni but with omni = False and do_3D = True. As in the case of the Omnipose 3D plant_omni model, we found many small objects were predicted and additionally postprocessed the output segmentation, removing all objects with volume < 2,500 voxels.

**PlantSeg2D and 3D.** We used PlantSeg version 1.6.0. We applied PlantSeg[6] models pretrained on 2D slices and 3D volumes of the LRP and Ovules dataset as is, without any custom preprocessing of downloaded images. The 'light-sheet_2D_unet_root_ds1x' and 'light-sheet_3D_unet_root_ds1x' models were used for 2D and 3D segmentation of LRP, respectively. The 'confocal_2D_unet_ovules_ds2x' and 'confocal_3D_unet_ovules_ds2x' models were used for 2D and 3D segmentation of Ovules, respectively. We used a tile size of (1, 96, 96) for 2D and (128, 160, 160) for 3D with 'MultiCut' and watershed in 2D or 3D, respectively, with postprocessing 'on' to produce segmentations. Multiple nonzero labels corresponded to image background in 2D predicted segmentations. To not unfairly penalize PlantSeg, for 2D evaluation and consensus segmentation with u-Segment3D, we reassigned all 2D predicted labels with more than 0.7 spatial overlap with the reference background as 'background'. This problem was not present for PlantSeg3D where we could assign the largest contiguous label as 'background'. PlantSeg2D and 3D were evaluated on the test splits of Ovules and LRP. 2D test datasets were constructed from provided 3D splits as described below for the training of segmentation models.

### Training of 3D segmentation models
**Dataset construction.** Datasets were resized to be isotropic voxels with the same dimensions as used for the pretrained Cellpose experiments in Extended Data Fig. 3. We used the existing train/test/validation split for each dataset when provided. For datasets without a test split, and those without a validation split, all of which come from EmbedSeg[11], we followed the splitting procedure used in the official EmbedSeg code repository for each dataset (https://github.com/juglab/EmbedSeg/tree/main/examples/3d/). This procedure first splits 10% of the train data as test, if required, and the 15% of the remainder train data as validation. All 3D models were trained on the constructed train/validation datasets and evaluation reported on the test dataset.

**EmbedSeg3D.** We used EmbedSeg version 0.2.5. We trained EmbedSeg3D models following the provided notebooks for each dataset (https://github.com/juglab/EmbedSeg/tree/main/examples/3d/), using the model with best IoU after 200 epochs. For Ovules and LRP, we followed the example for the Arabidopsis-CAM dataset.

**StarDist3D.** We used StarDist version 0.9.1. We trained StarDist3D[52] models for 400 epochs and 96 rays following the example code (https://github.com/stardist/stardist/tree/main/examples/3D/). For datasets with high shape anisotropy—mouse skull nuclei and LRP or large cell sizes—Arabidopsis and Ovules, we isotropically downsampled the volume and, if required, additionally doubled the subsampled grid to ensure the neural network field of view was greater than median object size.

### Training of 3D segmentation models guided by u-Segment3D consensus segmentations
A new EmbedSeg3D model was trained for *Platynereis* nuclei using the same train/validation data splits but using the u-Segment3D consensus segmentation of the trained Embedseg2D model as the reference 3D instance masks. We do not want to train to convergence. Instead, we stopped training as the IoU on the validation split begins to plateau (25 epochs). The best IoU model was then evaluated on the test data split using the actual reference segmentation.

### Training of 2D segmentation models
**Dataset construction.** The 2D train/test/validation splits were derived based on the 3D train/test/validation volumes constructed for 3D segmentation. For each 3D volume, we equisampled 15% of the 2D slices between the slice containing the first cell and that containing the last cell. For example, if the reference segmentation *x-y*-slices have a cell starting from $z = 15$ and have a cell up to $z = 225$, we would sample every 15th from 15. This was done for *x-y*, *x-z* and *y-z* views independently for each volume. The 2D slices sampled from train/test/validation 3D volumes formed the respective 2D train/test/validation datasets that all 2D models were trained on.

**EmbedSeg2D.** We used EmbedSeg version 0.2.5. We trained EmbedSeg2D models for each dataset following the example codes for dsb-2018 (https://github.com/juglab/EmbedSeg/tree/main/examples/2d/dsb-2018/). In 2D, which yields thousands of cells (Extended Data Fig. 7), we found that the instance embedding-based training approach of EmbedSeg2D was time-consuming and observed marginal performance gains after just a few epochs. Therefore, we only ran the full 200 epochs on the smaller datasets with fewer cells: mouse organoids, mouse skull nuclei, *Platynereis* ISH nuclei and *Platynereis* nuclei. For the remainder, we ran sufficient epochs (minimum 35 epochs) to observe plateauing and slowdown of the validation IoU. We also rederived the train/validation at 5% equisampling, filtering out 2D slices that did not contain a sufficient number of foreground pixels. Performance was still evaluated on the original constructed 2D test dataset equisampled at 15% for all datasets.

**StarDist2D.** We used StarDist version 0.9.1. We trained StarDist2D models following the tutorial notebook (https://github.com/stardist/stardist/blob/main/examples/2D/2_training.ipynb/) with 32 rays. A unique feature of StarDist models is the ability to enforce a star-convex shape prior. Therefore, we trained StarDist2D for all datasets first with the parameter 'train_shape_completion = True'. However, we found this was detrimental for mouse skull nuclei, LRP and Ovules whose cells in 2D slices were presumably too oblong and/or concave. 'Trained' models only predicted cell centroids with a circle of the same radius for each cell. For these datasets, we set train_shape_completion = True, and further filtered out all 2D images in train/validation that did not contain a sufficient number of foreground pixels. Performance was still evaluated on the original constructed 2D test dataset.

**Cellpose2D.** We used Cellpose version 3.0.8. We trained the latest Cellpose3 (ref. 53) 'cyto3' model and current default model using the 'train_seg' function in single-channel mode (channels = [0,0]), following the official API documentation (https://cellpose.readthedocs.io/en/latest/train.html) for the default 100 epochs. For Ovules, we trained for 250 epochs. To generate 2D instance segmentation masks (Extended Data Fig. 7), we used the default parameters for Cellpose. The equivalent u-Segment3D generation (Cellpose(u)) uses the same parameters but now with u-Segment3D's gradient descent and connected component clustering implementation with a parameter setting of 50 gradient descent iterations, gradient decay of 0.1, and automatically inferring the binary cell foreground threshold as the higher of a minimum threshold 0.25 and the higher threshold from applying 3-class Otsu thresholding to the normalized predicted cell probability. The minimum threshold was used to suppress spurious segmentations when the input 2D image was of only background without cells.

## u-Segment3D with trained 2D segmentation models
For a dataset, we used the same parameter settings for each trained 2D model and PlantSeg2D. The main u-Segment3D parameters to consider when using the indirect method with 2D segmentation masks are (i) choice of distance transform, (ii) gradient decay, (iii) gradient descent iterations and (iv) minimum cell size. For these we used (i) the explicit medial point diffusion distance transform, (ii) a gradient decay of 0.01, (iii) 50 gradient descent iterations and (iv) a minimum cell size cutoff of 50. These base parameters were rationally adapted to generate the consensus 3D cell segmentation in individual datasets as follows. In case of sporadic segmentation across consecutive 2D slices, we used a neighborhood size of $3 \times 3 \times 3$ pixels for content-based averaging of foreground (Supplementary Fig. 2) for all datasets and not varied.

**Ovules.** Cells are densely packed. We used an increased 250 gradient descent iterations.

**Lateral root primordia.** Cells are densely packed. There are both cells that are small and convex-like and are highly elongated with long branches. We used for the distance transform, the explicit Poisson diffusion distance with medial-axis skeletons as the point set source attractor (Supplementary Fig. 1), gradient decay, $\tau = 0.1$, and an increased 100 gradient descent iterations.

**Arabidopsis-CAM.** Cells are densely packed and have small volume in voxels. We used an increased 100 gradient descent iterations.

**Mouse organoids.** Cells are not densely packed, and not branching. Base parameters were used without modification.

**C. elegans.** Cells are more densely packed than mouse organoids. We used an increased 100 gradient descent iterations.

**Mouse skull nuclei.** Nuclei are not densely packed, and not branching. Base parameters were used without modification.

**Platynereis ISH nuclei.** Nuclei are small, but not densely packed, and not branching. Base parameters were used without modification. We used a decreased minimum cell size cutoff of 25.

**Platynereis nuclei.** Nuclei are small, some touching larger nuclei, and not branching. We used an increased 100 gradient descent iterations, and a decreased minimum cell size cutoff of 25.

## Evaluation of segmentation quality
**Segmentation quality in single images.** For single 2D and 3D images, we find the optimal matching between predicted and reference cell segmentations. Given a total number of $M$ predicted cells, and $N$ reference cells, we iterate and find for each predicted cell $i$, its $k$-nearest reference cells according to the distance between their centroids. For each of the $k$-nearest reference cells, we compute the IoU metric (0–1; see below). This produces an IoU$(i,j) \in \mathbb{R}^{M \times K}$ matrix. We convert this to a distance cost matrix, dist$(i,j) = 1 - $IoU$(i,j) \in \mathbb{R}^{M \times K}$. The optimal matching between predicted and reference cells is then found by solving the linear sum assignment using a modified Jonker–Volgenant algorithm[93] implemented by Python SciPy's scipy.optimize.linear_sum_assignment function and retaining only the pairings that overlap spatially (IoU$(i,j) > 0$). The segmentation quality for an image was then assessed by (i) the mean IoU, to measure the spatial overlap of matched predicted and reference cells and (ii) the F1 score (see below), that is, the harmonic mean of precision and recall to measure how accurately the reconstructed segmentations detected only reference cells.

**IoU.** Also called the Jaccard index, IoU is defined as the total number of pixels in the intersection divided by the total number of pixels in the union of two binary segmentation masks $A$ and $B$, IoU$(A,B) = \frac{|A \cap B|}{|A \cup B|}$.

**F1 score.** Predicted cells that are validly matched to a reference cell (IoU > 0) were defined as true positives (TPs). Predicted cells that are not matched to a reference cell are false positives (FPs) and reference cells that are not matched are false negatives (FNs). The precision is the number of matched cells divided by the total number of predicted cells, precision $= \frac{TP}{TP+FP}$. The recall is the number of matched cells divided by the total number of reference cells, recall $= \frac{TP}{TP+FN}$. F1 score is the harmonic mean of precision and recall, F1 $= 2\frac{precision \times recall}{precision + recall} = \frac{2TP}{2TP+FP+FN}$.

## AP curve
We evaluate the quality of cell segmentation using AP, consistent with popular segmentation models such as StarDist[1], Cellpose[4] and Omnipose[5]. Each predicted cell label mask is matched to the reference cell label mask that is most similar, as defined by IoU. The predictions for an image are evaluated at various levels of IoU. At a lower IoU, a predicted cell can overlap a reference cell with fewer pixels to determine a valid match. For a given IoU threshold, the valid matches define the TPs, the predicted cells with no valid matches are FPs, and the reference cells with no valid matches are FNs. Using these definitions, the AP metric for a segmented image is

$$AP = \frac{TP}{TP + FP + FN}$$

The AP curve is reported for a dataset by averaging over the AP metric for each image in the dataset for a range of IoU thresholds. Optimal matching of predicted and reference cells is too computationally demanding in 3D even when restricting the search to nearest neighbors. Consequently, we use the same approximate matching as in Cellpose derived from the fast matching functions in StarDist. We find this fast matching is not invariant to cell ID permutation. To compute the correct AP, we first relabel all cells sequentially after performing an indirect stable sort based on their $(x,y,z)$ centroids for both reference and predicted cell segmentation independently. In line with Cellpose, the AP curve is reported for 11 IoU thresholds equisampling the range $[0.5, 1.0]$. Many datasets, for example, Ovules, do not rigorously label every cell in the image but only the cells of the primary, single connected component object in the field of view. In contrast, pretrained Cellpose models predict all cells in the field of view. For fair evaluation, for these datasets (all except for Embedseg skull nuclei, Platynereis nuclei and Platynereis ISH nuclei), we use the reference segmentation to define the foreground connected components to evaluate AP and include all predicted cells part of binary foreground spatial connected components that share at least 25% overlap with a reference connected component. For DeepVesselNet, we use at least 1% overlap due to the thinness of vasculature.

## F1 curve

We use the F1 curve as an additional segmentation performance measure. We compute the F1 curve for a dataset similar to the AP curve, by averaging the F1 score metric for each image in the dataset for a range of IoU thresholds for a range of IoU thresholds. We use the same matching procedure and IoU thresholds as for the AP curve. The F1 is correlated to AP that assigns greater weight to TPs.

$$F1 = 2\frac{precision \times recall}{precision + recall} = \frac{2TP}{2TP + FP + FN}$$

## Visualization

The Fiji ImageJ[94] 3D viewer plugin was used to render 3D intensity and segmentation image volumes. To visualize the intensity in Fig. 4, we acquired a snapshot of the rendering, then applied an inverse lookup table to the snapshot. Surface meshes in Fig. 4 were extracted using u-Unwrap3D[39] and visualized using MeshLab[95]. Rotating surface mesh movies were created using ChimeraX[96].

## Statistics and reproducibility

Our experiments, demonstrating consensus 3D shape reconstruction from 2D segmentations, are not hypothesis-driven. Randomization and blinding were not applicable in this study and were not performed. We performed benchmarking in Figs. 2 and 3 and Extended Data Figs. 1–7 using the provided data splits of publicly available datasets. We selected demonstration datasets of $n = 1$ image in Figs. 4 and 5 and Extended Data Figs. 8 and 9 to showcase application to a diverse range of biological imaging, and to highlight the additional features of translating only $x$-$y$ 2D segmentations, recovering 3D cellular surface protrusions, and parallel computing for large tissue volumes.

## Reporting summary

Further information on research design is available in the Nature Portfolio Reporting Summary linked to this article.

## Data availability

All data used in this study are publicly available. The already published segmentation datasets used for benchmarking and example demonstration are available from their sources as documented in Supplementary Table 2 and under 'Datasets' in the Methods. The original microscopy data generated for this paper are available on Zenodo (https://doi.org/10.5281/zenodo.15692302)[97].

## Code availability

u-Segment3D is freely available both through the Python Package Index, PyPI, and at https://github.com/DanuserLab/u-Segment3D. The GitHub repository includes installation instructions; our code to automate parameter tuning of Cellpose models; example scripts to help users get started; a download link to example data and code for a MATLAB-based GUI for running u-Segment3D.

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

## Acknowledgements

Funding for this work in the Danuser lab was provided by National Institutes of Health grants R35GM136428 (NIGMS) and U54CA268072 (NCI) to G.D., and a HHMI Hanna H. Gray Fellow (GT16003) to G.M.G. Software dissemination by the Danuser lab was funded by RM1-GM145399 to G.D. E.J. was funded by the Wellcome Trust (grant 224040/Z/21/Z to E.J.). Funding for the 3D data acquisition in the Sorger Lab was provided by the Ludwig Cancer Research, PCA (U2C-233262) and HTA (U2C-233280) to P.K.S. The Fiolka lab is supported by the grants R35GM133522 (NIGMS) and R01EB035538 (NIBIB) to R.F. T cell culture imaging was performed using the Oxford-Zeiss Centre of Excellence in Biomedical Imaging and the Kennedy Trust for Rheumatology Research (grants 202117 and 202103, respectively). For epidermal organoid imaging, we acknowledge the Quantitative Light Microscopy Core, a shared resource of the Harold C. Simmons Cancer Center, supported in part by an NCI Cancer Center support grant 1P30 CA142543-01. This research was supported in part by the computational resources provided by the BioHPC computing facility located in the Lyda Hill Department of Bioinformatics, UT Southwestern Medical Center (https://portal.biohpc.swmed.edu/). The research within this work complies with all relevant ethical regulations as reviewed and approved by the UT Southwestern Medical Center. Zebrafish husbandry and experiments described here have been approved and conducted under the oversight of the IACUC at UT Southwestern (protocol no. 101805). All mouse experiments complied with all relevant ethical regulations and were performed according to protocols approved by the IACUC at the UT Southwestern Medical Center (protocol no. 2016-101360). T cells were obtained from blood leukocyte cones purchased from NHS Blood and Transplant, John Radcliffe Hospital, Oxford, United Kingdom. Blood cones were used under the ethical guidelines of the NHS Blood and Transplant. The Non-Clinical Issue division of the National Health Service approved the use of blood leukocyte cones at the University of Oxford (REC 11/H0711/11).

## Author contributions

Conception: A.J., F.Y.Z. (using pretrained 2D for 3D segmentation) and F.Y.Z. (u-Segment3D). Investigation and analysis: F.Y.Z., C.Y. and Z.M. Graphical user interface (GUI) development: Q.Z. Data generation: C.Y. (CyCIF tissue), S.D. (zebrafish macrophages), Z.M., M.T.I., J.L., H.M.B., K.M.D., S.J.M. (lung micrometases), B.N. (epidermal organoid), E.J. (T cell co-culture), G.M.G., B.-J.C. (COR-L23 single cell with ruffles), Z.M., J.L. (ctALSM microscope development), H.M.B. (tissue clearing), A.W., H.M.B., K.D. (septin-cleared tissue) and R.F. (HBEC aggregate). Supervision: G.D. Funding acquisition: G.D., K.M.D., R.F. and P.K.S. Writing—original draft: F.Y.Z. and G.D. Writing—review and editing: all authors.

## Competing interests

K.M.D. and R.F. have a patent covering ASLM (US10989661) and consultancy agreements with 3i. K.M.D. has an ownership interest in Discovery Imaging Systems. P.K.S. is a cofounder and member of the Board of Directors (BOD) of Glencoe Software, a member of the BOD for Applied Biomath and a member of the Scientific Advisory Board for RareCyte, NanoString, Reverb Therapeutics and Montai Health; P.K.S. holds equity in Glencoe, Applied Biomath and RareCyte. P.K.S. consults for Merck and the Sorger lab has received research funding from Novartis and Merck in the past 5 years. S.J.M. is an advisor for Frequency Therapeutics and Protein Fluidics, as well as a stockholder in G1 Therapeutics and Mereo Biopharma. The other authors declare no competing interests.

## Additional information

**Extended data** is available for this paper at https://doi.org/10.1038/s41592-025-02887-w.

**Correspondence and requests for materials** should be addressed to Felix Y. Zhou or Gaudenz Danuser.

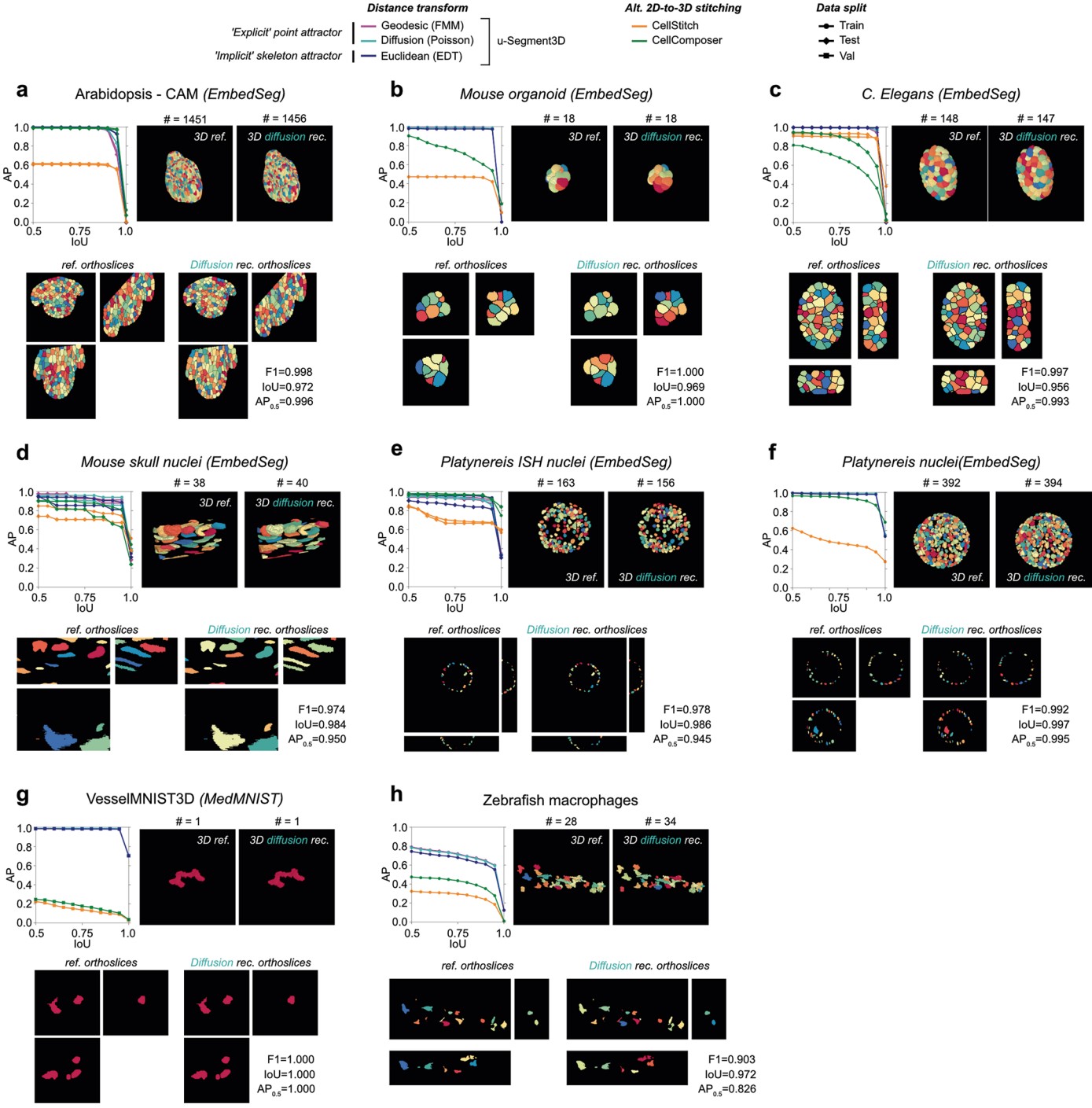

**Extended Data Fig. 1 | Reconstruction average precision performance of 3D cell shapes from ideal 2D slices sourced from real datasets.** Reconstruction performance measured by the mean average precision curve (Methods) using three different 2D distance transforms with u-Segment3D for all datasets not included in Fig. 2d–f in comparison to using CellComposer (green lines) or CellStitch (orange lines) which directly try to stitch 2D segmentations. **a**) Arabidopsis-CAM, **b**) mouse organoid, **c**) *C.Elegans*, **d**) mouse skull nuclei, **e**) *Platynereis* ISH nuclei, **f**) *Platynereis* nuclei, **g**) vesselMNIST3D and **h**) zebrafish macrophages. For each dataset, top row, left-to-right: average precision vs intersection over union (IoU) curve; 3D rendering of reference cells, and cells reconstructed using the point-based centroid diffusion distance transform. Bottom row, left-to-right, the respective midplane orthoslices in the three orthogonal views. All available data splits were used for each dataset except VesselMNIST3D for which we used only the validation split. See Supplementary Table 2 for the number of objects and images in each split.

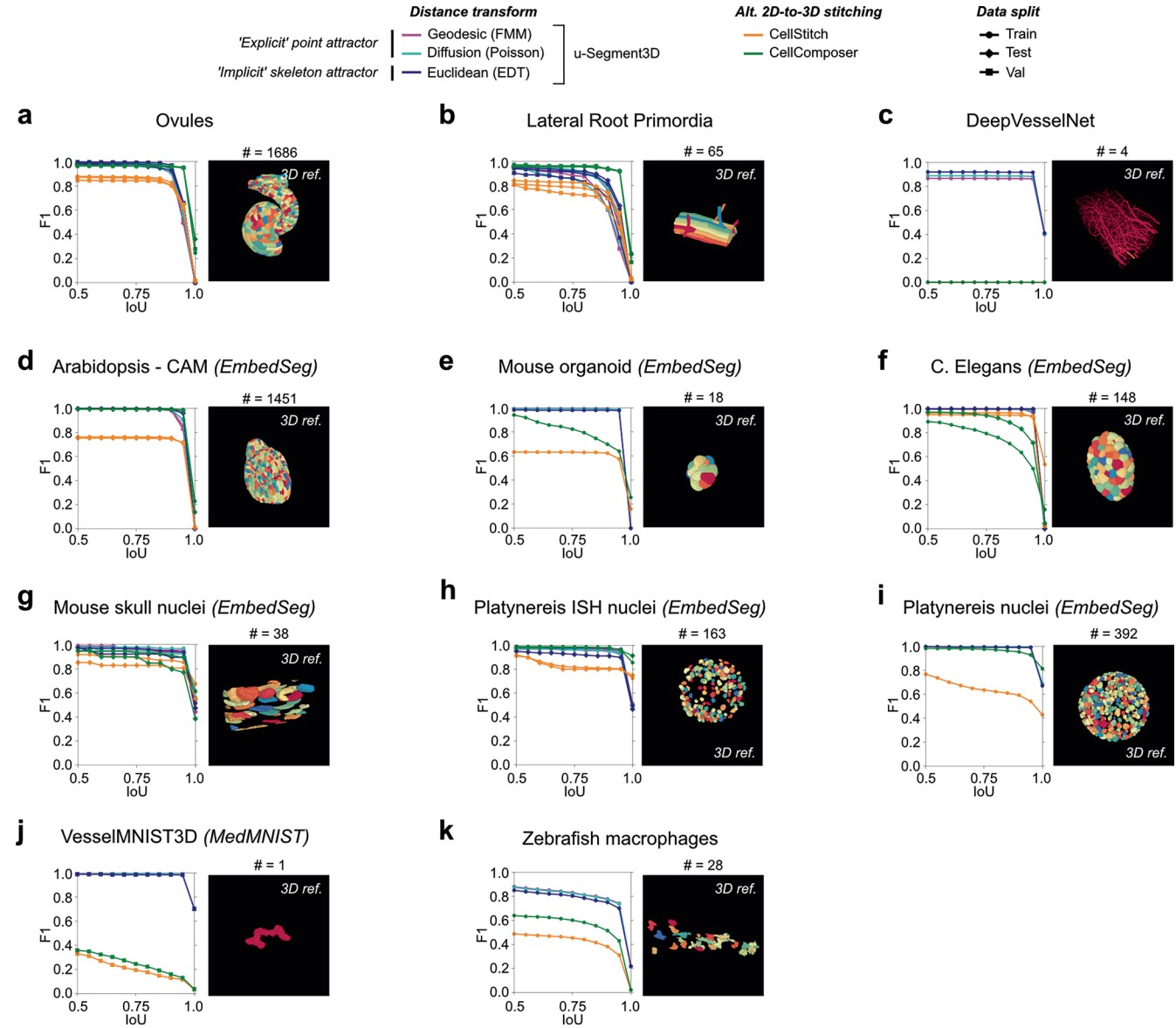

**Extended Data Fig. 2 | Reconstruction F1 performance of 3D cell shapes from ideal 2D slices sourced from real datasets.** Performance of the same shape reconstruction and same methods in Fig. 2 and Extended Data Fig. 1 measured alternatively by F1 score, the harmonic mean of precision and recall (Methods) for the same IoU thresholds as for the average precision curve.

**a**) Ovules, **b**) Lateral Root Primordia, **c**) DeepVesselNet, **d**) Arabidopsis-CAM, **e**) mouse organoid, **f**) *C.Elegans*, **g**) mouse skull nuclei, **h**) *Platynereis* ISH nuclei, **i**) *Platynereis* nuclei, **j**) vesselMNIST3D and **k**) zebrafish macrophages. For each dataset, left: F1 vs intersection over union (IoU) curve, right: 3D rendering of reference cells.

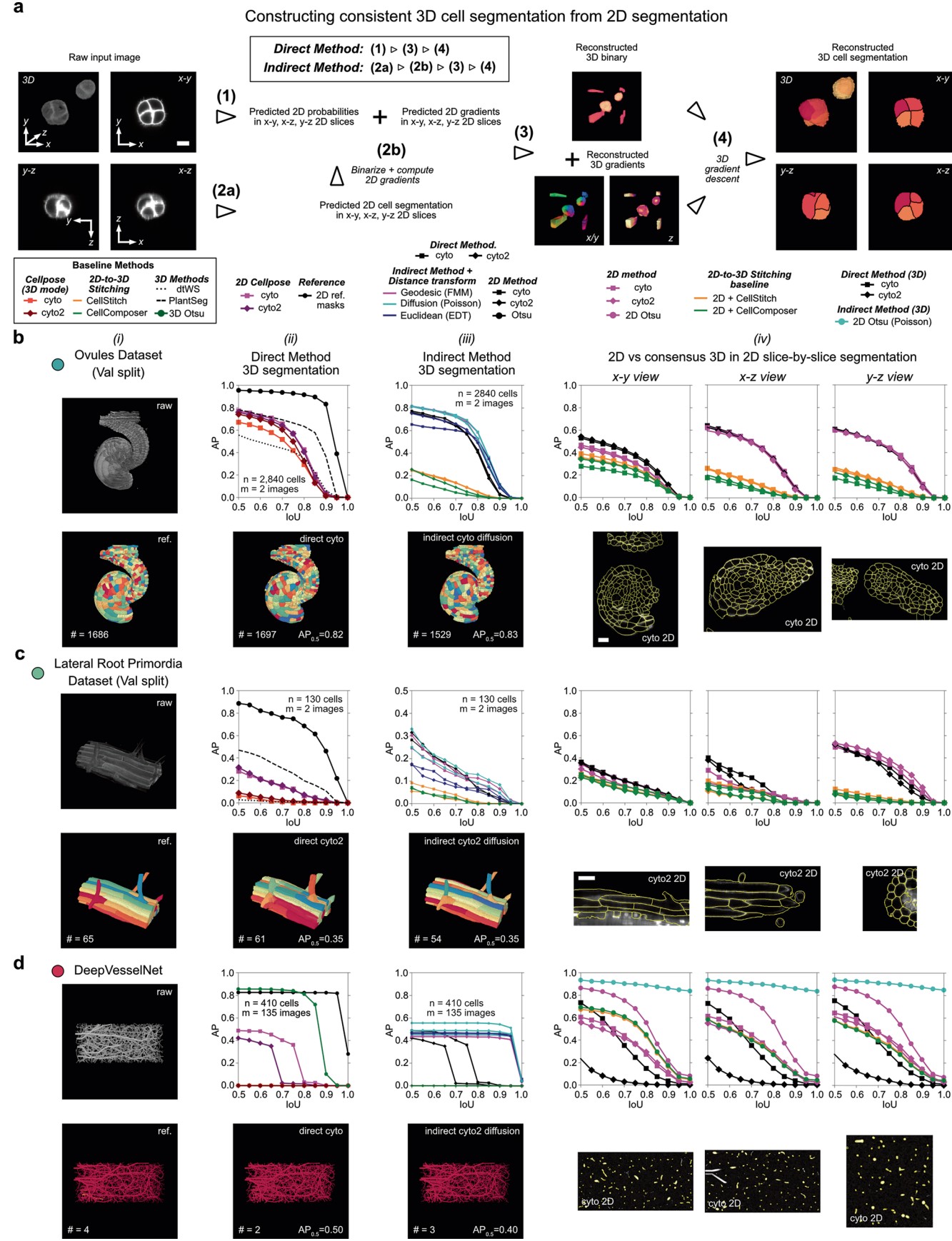

**Extended Data Fig. 3 | See next page for caption.**

**Extended Data Fig. 3 | Consensus segmentation of real 3D datasets using pretrained Cellpose 2D models applied to orthogonal x-y, x-z, y-z views.**
**a**) Schematic of the direct method (steps 1,3,4) utilizing predicted spatial gradients and cell probability maps output, and the indirect method (steps 2a, 2b, 3, 4) utilizing predicted instance segmentation masks to construct the 3D consensus segmentation. The latter converts masks using the specified 2D distance transform (see Fig. 2c). Step 3 image panels depict the Cellpose outputs which also predicts extraneous cells at the border. These are removed during postprocessing and therefore not present in step 4. **b**) u-Segment3D performance on the Ovules dataset (validation (val) split, n=2840 cells, m=2 volumes). (i) 3D rendering of raw image (top) and reference 3D labels (bottom). (ii) Average precision (AP) curve of the direct method using either Cellpose 2D models, and Cellpose 3D, compared to the best reconstruction from ideal 2D slices from Fig. 2d (top). 3D rendering of the best Cellpose model with u-Segment3D (bottom). (iii) AP curve of the indirect method using the predicted masks of the pretrained Cellpose models with u-Segment3D using different 2D distance transforms, compared to using direct method u-Segment3D, and to

stitching by CellStitch and CellComposer (top). 3D rendering of the best indirect method u-Segment3D segmentation (bottom). (iv) AP curve of the 2D slices from the direct method 3D consensus (that is 3D-to-2D segmentation, black lines), and 2D-to-3D stitched CellStitch (orange) and CellComposer (green) segmentations, compared to Cellpose 2D segmentation masks of each 2D slice (magenta lines) in each orthoview, *x-y, x-z, y-z* from left-to-right. Representative 2D segmentations from the highest performing Cellpose model are shown below the AP curves. **c), d**) Same as b) for the Lateral Root Primordia (validation (val) split, n=130 cells, m=2 volumes) and DeepVesselNet (n=410 network components, m=135 images) dataset with representative visualizations from the best performing combination of Cellpose model and distance transform. For **d**) we also included binary Otsu thresholding as a native 3D baseline, and as a 2D baseline with the indirect u-Segment3D method (explicit diffusion transform, circle marker) (turquoise), CellStitch (orange) and CellComposer (green). All scalebars: 20μm. Images in panel **a** adapted from the mouse organoid dataset in ref. [11]. Images in panels **b**, **c** adapted from the Ovules and LRP dataset in ref. [6]. Images in panel **d** adapted from the synthetic dataset in ref. [9].

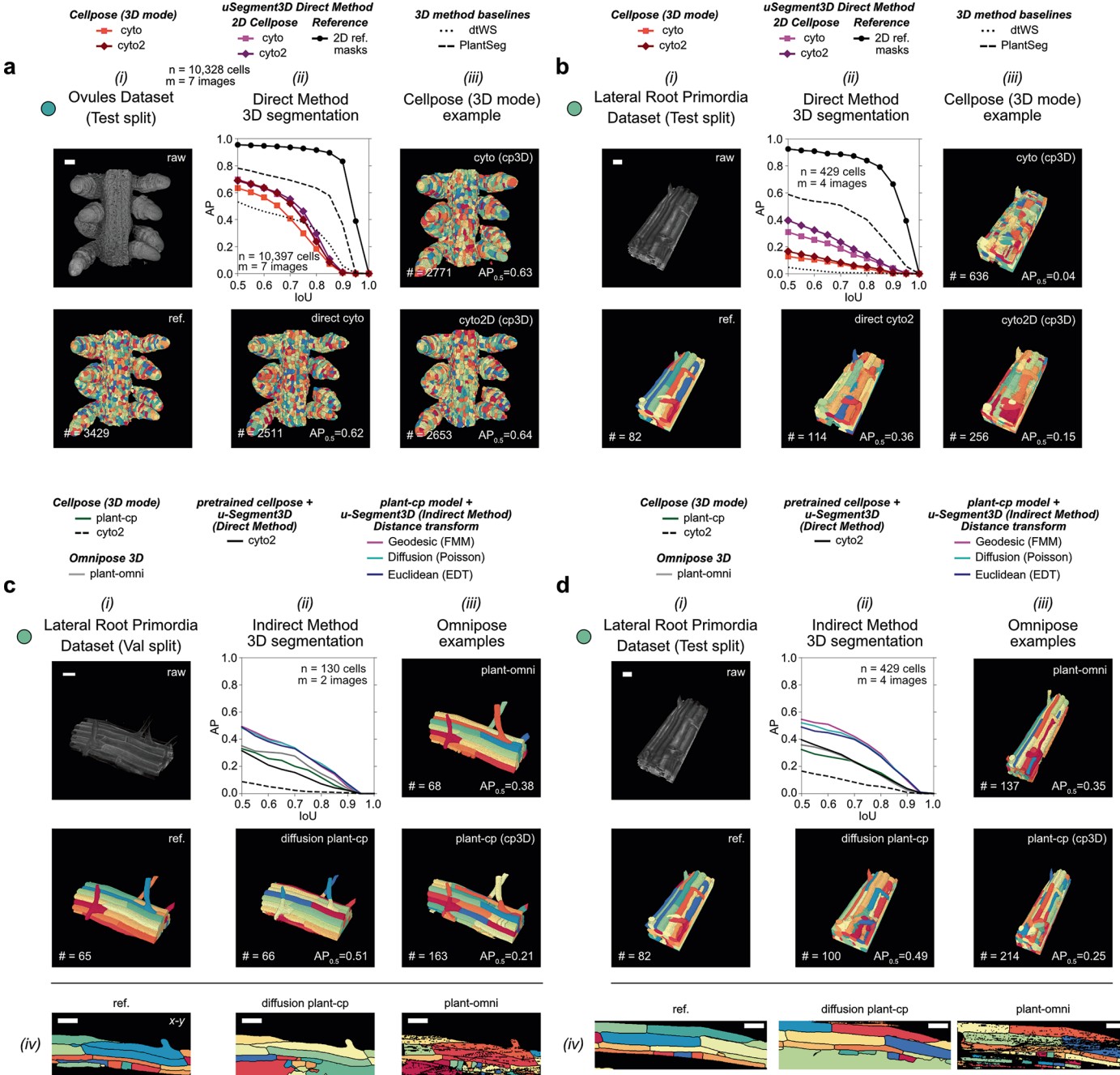

**Extended Data Fig. 4 | Performance of u-Segment3D using pretrained vs specialized plant Cellpose segmentation models. a)** Performance of consensus u-Segment3D and Cellpose 3D segmentations using the same pretrained Cellpose 2D models on the test split of Ovules relative to two baselines: classical unsupervised distance transform watershed and PlantSeg, a specialized 3D UNet model trained directly with the 3D masks. (i) example volume and corresponding reference segmentation; (ii) AP curves of all models, with best 3D reconstruction from ideal 2D slices (black line) (top) and segmentation of best pretrained model with u-Segment3D (bottom); (iii) Cellpose 3D segmentations using cyto (top) or cyto2 (bottom) models. Image in panel **a** adapted from the Ovules dataset in ref. 6. **b)** Performance of consensus u-Segment3D and Cellpose 3D segmentations using the same pretrained Cellpose 2D models on the test split of the Lateral

Root Primordia (LRP) dataset relative to two baselines: classical unsupervised distance transform watershed and PlantSeg, a specialized 3D UNet model trained directly with the 3D masks. (i)-(iii) similar to **a**. **c)** Performance on LRP val split using u-Segment3D or Cellpose 3D and plant-cp: a specialized Cellpose 2D model trained on LRP, with plant-omni: a specialized Omnipose 3D model trained on LRP natively in 3D. (i)-(ii) similar to **a**. (iii) Native 3D segmentation using plant-omni (top) or Cellpose 3D segmentation with plant-cp (bottom). (iv) mid $x$-$y$ slice of the reference, u-Segment3D diffusion centroid transform plant-cp consensus and 3D plant-omni segmentation (left-to-right). **d)** Same as **c)** for the test split of LRP. All scalebars: 20 μm. Images in panels **b-d** adapted from the LRP dataset in ref. 6.

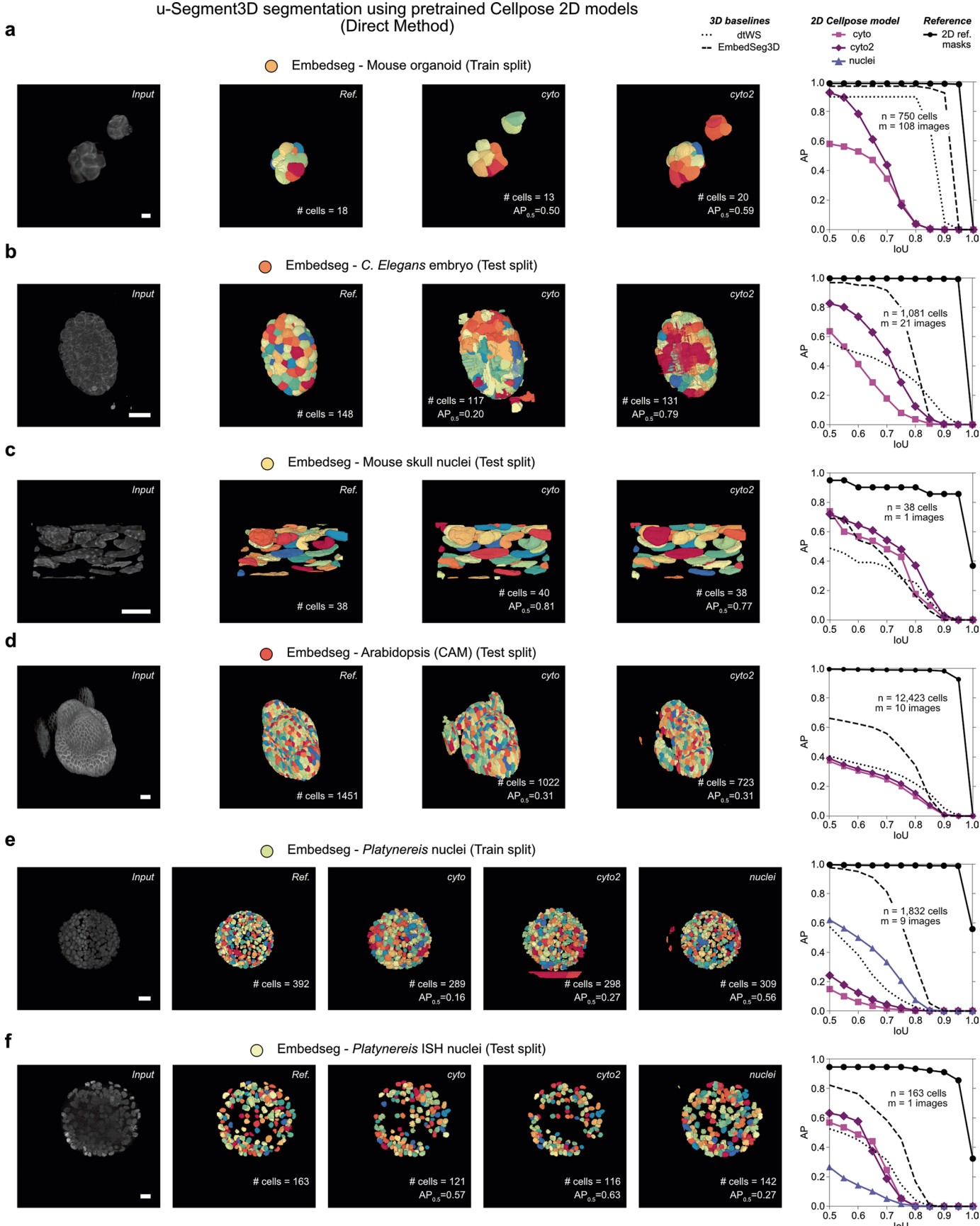

**Extended Data Fig. 5 | See next page for caption.**

**Extended Data Fig. 5 | Performance of 2D-to-3D segmentation for real datasets using u-Segment3D with pretrained cellpose2D model outputs. a**) 3D cell segmentation performance of the mouse organoid (from Embedseg) for the train data split, n=740 cells, m=108 volumes) using pretrained Cellpose 2D models with u-Segment3D and the direct method illustrated in Extended Data Fig. 3a relative to baseline native 3D segmentation: classical unsupervised distance transform watershed (dtWS) (dotted black line) and training an EmbedSeg3D model (dashed black line) (Fig. 3, Methods). Left-to-right: 3D rendering of the raw volume, reference 3D segmentation, generated 3D segmentations for each Cellpose 2D model and the average precision (AP) curve of each model coplotted

with the AP curve of the best 3D reconstruction with ideal 2D slices from the three orthoviews (black line with circles). The same as a) for **b**) *C. Elegans* embryo (test data split, n =1,081 cells, m = 21 images), **c**) mouse skull nuclei (test data split, n=38 cells, m=1 image), **d**) Arabidopsis (CAM) (test data split, n=12,424 cells, m=10 images), **e**) *Platynereis* nuclei (train data split, n=1,832 cells, m=9 images), **f**) *Platynereis* ISH nuclei (test data split, n=163 cells, m=1 image). For the *Platynereis* nuclei which are approximately spherical, we additionally evaluated the performance of the Cellpose 'nuclei' 2D model (light purple line with triangles). All scalebars: 20 μm. Images in panels are adapted from the same-named dataset in the GitHub repository of ref. 11.

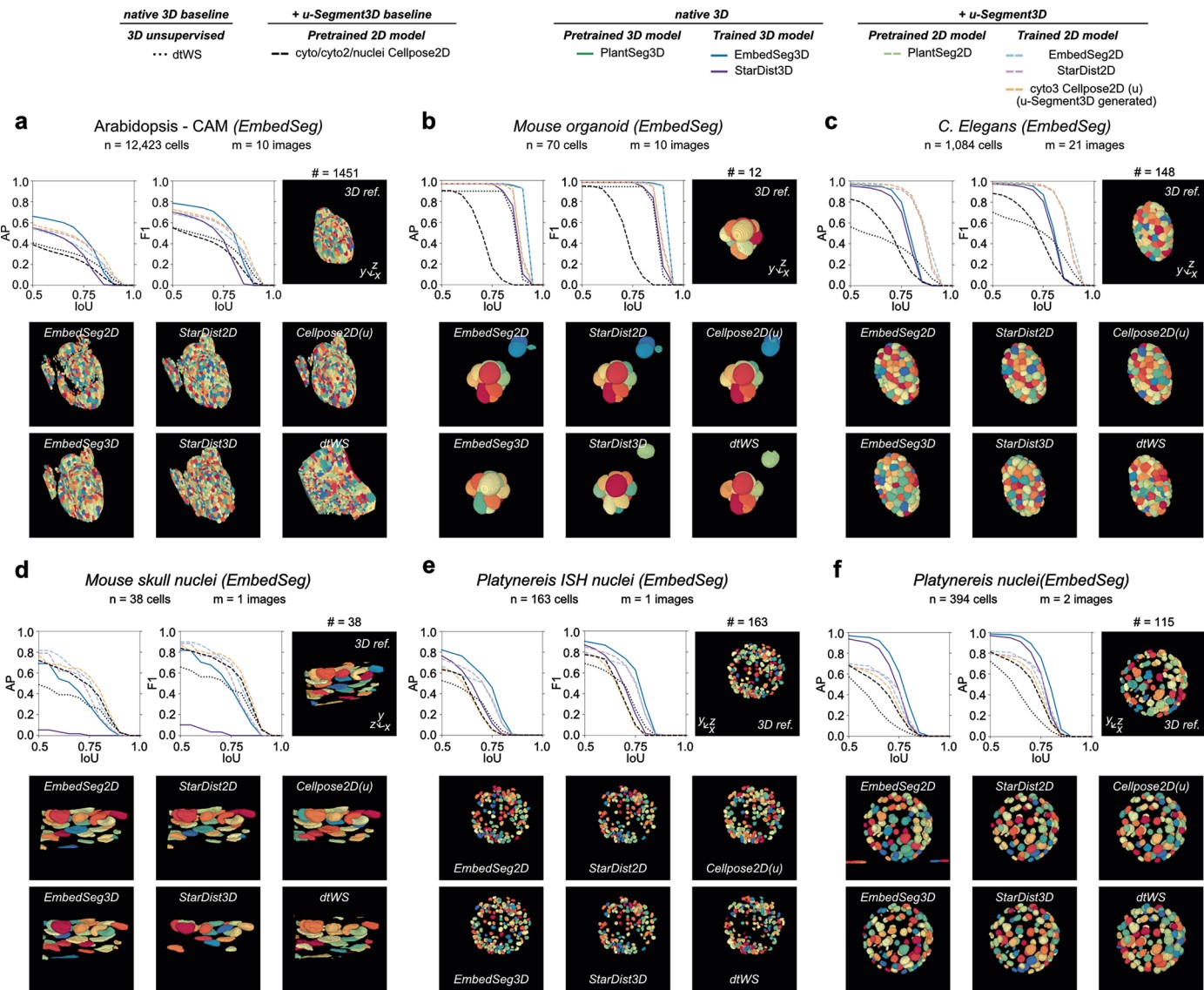

**Extended Data Fig. 6 | Performance of specialist native 3D trained segmentation models and consensus u-Segment3D segmentation with specialist trained 2D segmentation.** 3D cell segmentation performance of the Arabidopsis (CAM) dataset from Embedseg based on the test data split, n=12,423 cells, m=10 volumes) for native 3D trained (solid line, darker hue) and u-Segment3D consensus segmentation (dashed line, lighter hue) relative to the baseline of classical unsupervised distance transform watershed (dtWS) (dotted black line, Methods). Top row left-to-right: AP and F1 curve evaluation, and 3D rendering of the reference 3D segmentation. Bottom panels: 3D segmentations from each model with name of 2D model denoting the result from consensus u-Segment3D segmentation. The same as **a**) for test splits (Methods) of **b**) Mouse organoid (n=70 cells, m=10 images), **c**) *C. Elegans* embryo (n =1,081 cells, m = 21 images), **d**) mouse skull nuclei (n=38 cells, m=1 image), **e**) *Platynereis* ISH nuclei (n=163 cells, m=1 image), **f**) *Platynereis* nuclei (n=394 cells, m=2 images). 2D Cellpose segmentations were generated from network outputs using u-Segment3D's method (methods), as denoted by the (u) annotation.

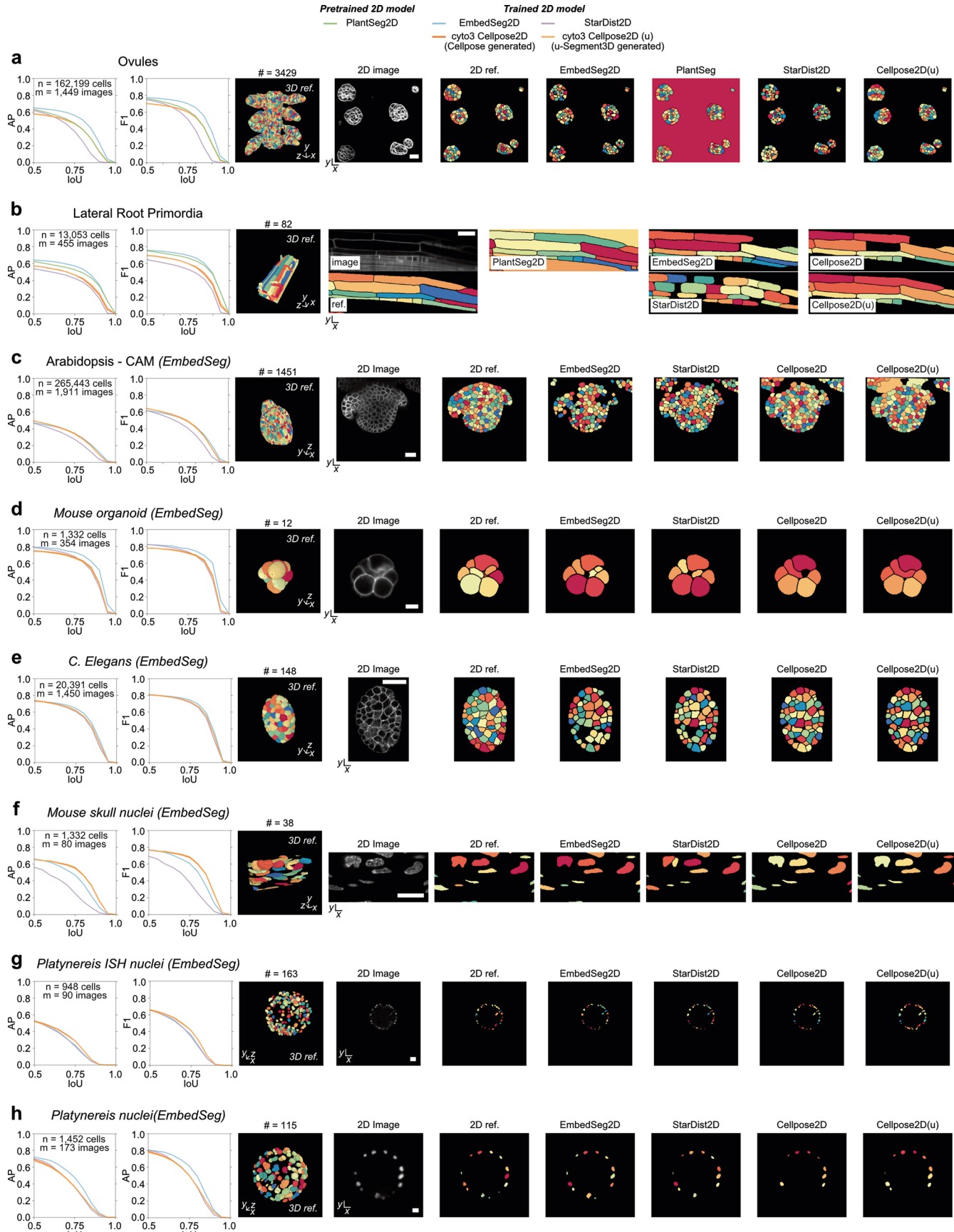

**Extended Data Fig. 7 | See next page for caption.**

**Extended Data Fig. 7 | Performance of training specialist 2D segmentation models on 2D *x-y*, *x-z* and *y-z* slices of 3D volumes. a**) 2D cell segmentation performance of each trained 2D model on sampled 2D *x-y*, *x-z* and *y-z* slice images from the test 3D volumes of the Ovules dataset (n=162,199 cells, m=1,449 images). Left-to-right: AP and F1 curve, 3D rendering of the reference 3D segmentation, raw image of the mid *x-y* slice and its reference 2D segmentation, and predicted 2D segmentation of the mid *x-y* slice with trained models. The same as **a**) of 2D images sampled from the test volumes (Methods) of **b**) Lateral Root Primordia (n=13,053 cells, m=455 images), **c**) Arabidopsis - CAM (n=265,443 cells, m=1,911 images), **d**) Mouse organoid (n=1,332 cells, m=354 images), **e**) *C. Elegans* embryo (n=20,391 cells, m=1,450 images), **f**) Mouse skull nuclei (n=1,332 cells, m=80 images), **g**) *Platynereis* ISH nuclei (n=948 cells, m=90 images), **h**) *Platynereis* nuclei (n=1,452 cells, m=173 images). Cellpose 2D segmentations were generated from Cellpose outputs using either Cellpose or using u-Segment3D, denoted by the absence or presence of an additional (u) annotation respectively. All scalebars: 20 µm. Images in panels **a**, **b** adapted from the Ovules and LRP datasets in ref. 6. Images in panels **c-h** adapted from the same-named dataset in the GitHub repository of ref. 11.

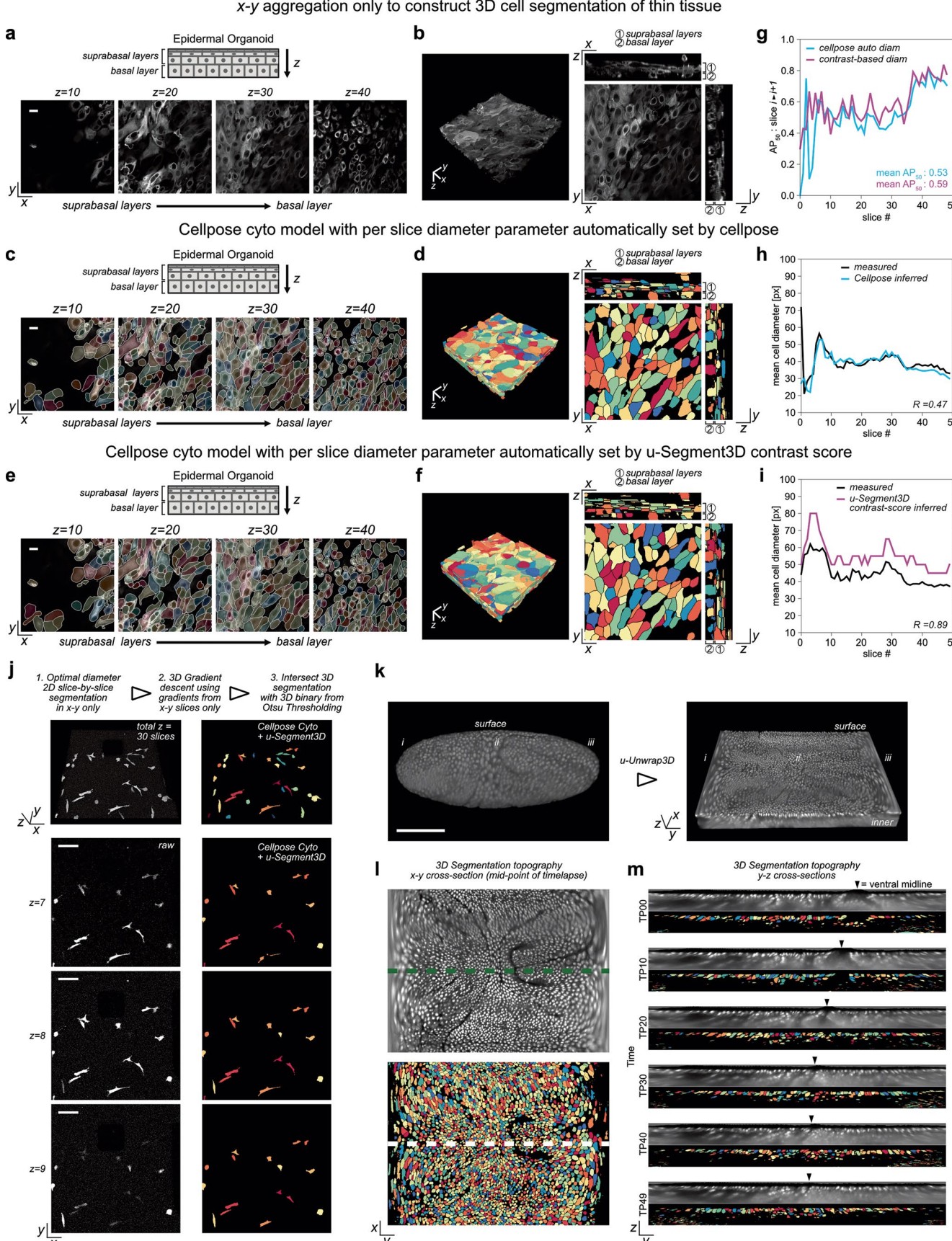

**Extended Data Fig. 8 | See next page for caption.**

**Extended Data Fig. 8 | Segmentation using only *x-y* 2D stacks. a**) Four equi-sampled x-y image slices from top to bottom, capturing the suprabasal to basal layer transition within an epidermal organoid culture. Scalebar: 20 μm. **b**) 3D rendering with axial interpolation to isotropic voxel resolution (left) and mid-section orthoslices (right). **c**) Cellpose 2D segmentations with diameter determined automatically per-slice by Cellpose. Cells are individually colored with white boundaries and overlaid onto the same *x-y* images in **a**). **d**) 3D rendering of the consensus segmentation of the *x-y* 2D stack in c) (left) and corresponding *x-y* slice at z = 20, mid-section *x-z*, *y-z* orthoslices (right). **e,f**) Same as c,d) respectively, with diameter determined automatically per-slice by contrast score. **g**) $AP_{50}$ between 2D segmentations in consecutive z-slices for increasing z slice number (#) using diameter auto-determined by Cellpose (cyan) or u-Segment3D's contrast score (magenta). **h**) Mean cell diameter predicted by Cellpose (cyan) and measured from the output segmentation masks (black) for each *x-y* slice. **i**) Mean cell diameter inferred by peak position of contrast score (magenta) and measured from the output segmentation masks (black)

for each *x-y* slice. *R* denotes Pearson's *R* in panels **g**)-**i**). **j**) Workflow to segment MDA231 human breast carcinoma cells by aggregating *x-y* slice segmentations predicted by Cellpose 2D 'cyto' models with optimal diameter determined by contrast score. 3D rendering of raw image volume and segmented cells (top) and in consecutive 2D *x-y* slices (bottom). Scalebar: 100 μm. Images in panel **j** adapted from the first timepoint of the '01' video in the train split of the Fluo-C3DL-MDA231 dataset in ref. 67. **k**) 3D rendering of raw image volume (left) and its surface proximal tissue section projected into a thin 3D volume using the u-Unwrap3D[39] framework. Scalebar: 100 μm. **l**) Consensus segmentation of individual cells in the unwrapped volume at timepoint (TP) 25 (top) using Cellpose 2D 'cyto' models with diameter optimized by contrast score (bottom). **m**) Snapshots of the mid *y-z* cross-section (position indicated by green dashed line in l) of the raw (top) and segmented (bottom) unwrapped volumes at 6 timepoints. Black arrowheads indicate the position of the ventral midline towards which cells converge from the two sides. Images in panels **k-m** adapted from the 'Cell01' video in the test split of the Fluo-N3DL-DRO dataset in ref. 67.

**a**

Workflow for nuclei and cancer cell segmentation in micrometastases

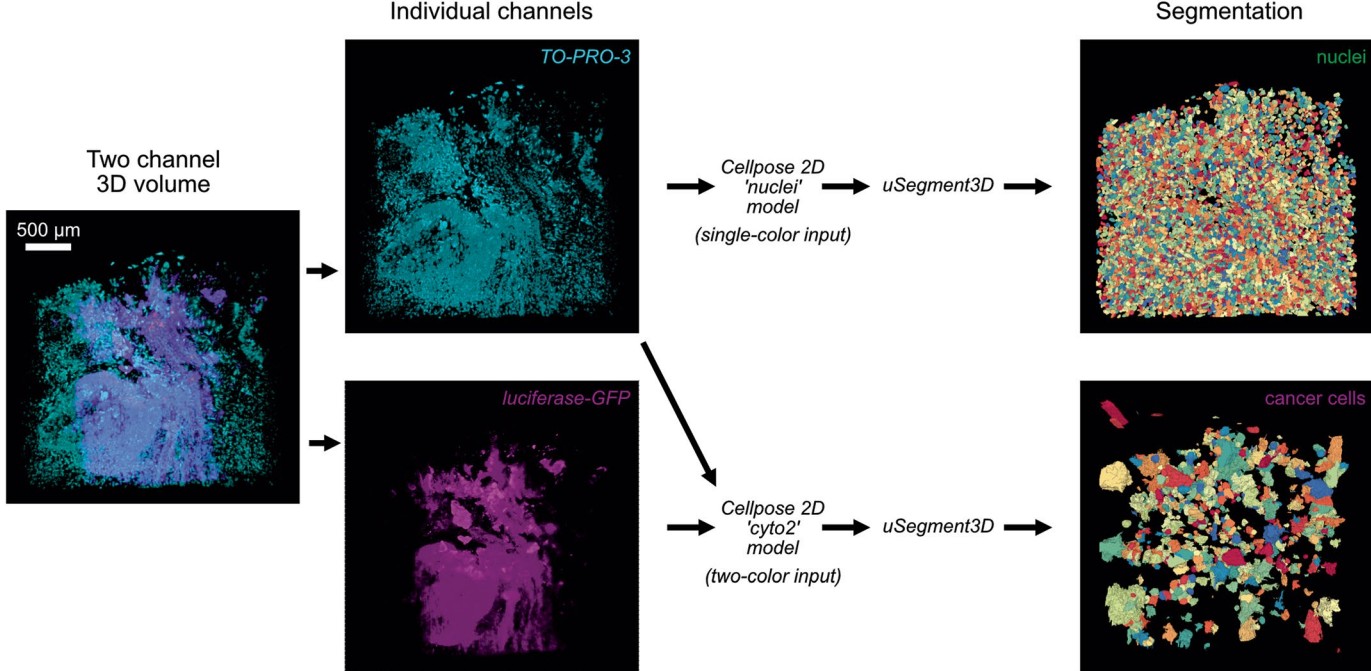

**b**

Filtering cancer cell segmentation after measuring image intensity statistics

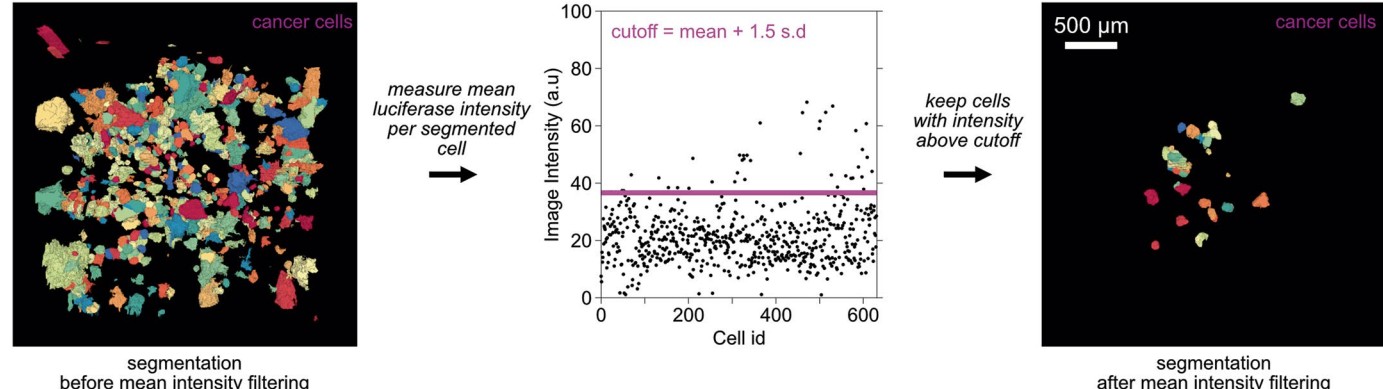

**Extended Data Fig. 9 | u-Segment3D generated consensus 3D segmentations can be filtered with image statistics to improve specificity for weakly labeled cells. a**) Workflow and 3D rendering of the resulting segmentation using the nuclei-only stained channel (TO-PRO-3, green) to extract lung nuclei (top branch); using both nuclei (green) and luciferase-GFP (magenta) channel to segment YUMM1.7 melanoma cells in micrometastatic colonies (bottom branch). **b**) Procedure to leverage the luciferase-GFP intensity to filter cells with a mean luciferase-GFP intensity above a cutoff (magenta) of mean + 1.5 standard deviation across all cells. All scalebars: 500 μm.

# Reporting Summary

## Statistics

For all statistical analyses, confirm that the following items are present in the figure legend, table legend, main text, or Methods section.

| n/a | Confirmed | |
|---|---|---|
| ☐ | ☒ | The exact sample size (*n*) for each experimental group/condition, given as a discrete number and unit of measurement |
| ☐ | ☒ | A statement on whether measurements were taken from distinct samples or whether the same sample was measured repeatedly |
| ☒ | ☐ | The statistical test(s) used AND whether they are one- or two-sided *Only common tests should be described solely by name; describe more complex techniques in the Methods section.* |
| ☒ | ☐ | A description of all covariates tested |
| ☒ | ☐ | A description of any assumptions or corrections, such as tests of normality and adjustment for multiple comparisons |
| ☐ | ☒ | A full description of the statistical parameters including central tendency (e.g. means) or other basic estimates (e.g. regression coefficient) AND variation (e.g. standard deviation) or associated estimates of uncertainty (e.g. confidence intervals) |
| ☒ | ☐ | For null hypothesis testing, the test statistic (e.g. *F*, *t*, *r*) with confidence intervals, effect sizes, degrees of freedom and *P* value noted *Give P values as exact values whenever suitable.* |
| ☒ | ☐ | For Bayesian analysis, information on the choice of priors and Markov chain Monte Carlo settings |
| ☒ | ☐ | For hierarchical and complex designs, identification of the appropriate level for tests and full reporting of outcomes |
| ☐ | ☒ | Estimates of effect sizes (e.g. Cohen's *d*, Pearson's *r*), indicating how they were calculated |

*Our web collection on statistics for biologists contains articles on many of the points above.*

## Software and code

Policy information about availability of computer code

| | |
|---|---|
| Data collection | Original microscopy (single cells(ruffles)) was acquired with a home-built light-sheet fluorescence microscope controlled by the software developed by Coleman technologies. It uses a 64-bit version of LabView 2016 equipped with the LabView Run-Time Engine, Vision Development Module and Vision Run-Time Module (National Instruments); navigate (https://github.com/TheDeanLab/navigate) developed by the Dean Lab (for cleared tissue); Zeiss software (for T cell co-culture); custom multi-scale light-sheet microscope with axially-swept light-sheet microscopy and controlled by custom Python software (https://github.com/DaetwylerStephan/self_driving_multiscale_control) (for Zebrafish vasculature) |
| Data analysis | u-Segment3D is available at https://github.com/DanuserLab/u-segment3D. Unwrapping of drosophila surface was performed with u-Unwrap3D available at https://github.com/DanuserLab/u-unwrap3D. Experiments in the paper except for Fig. 3, and its associated Extended Data Fig. 6,7 were performed prior to publication of Cellpose3 and uses Cellpose version 2.3.dev7+g03e02bc. The same conclusion does however apply to Cellpose 3 (the latest Cellpose version), which would be the version now installed by the u-Segment3D package. Fig. 3, and its associated Extended Data Fig. 6,7 used Cellpose3 (version 3.0.8). Omnipose used version 0.3.5.dev10+ge22262b. EmbedSeg used version 0.2.5. StarDist used version 0.9.1. PlantSeg used version 1.6.0. CellStitch used version 1.0.0. 3DCellComposer used version 1.2.1. |

For manuscripts utilizing custom algorithms or software that are central to the research but not yet described in published literature, software must be made available to editors and reviewers. We strongly encourage code deposition in a community repository (e.g. GitHub). See the Nature Portfolio guidelines for submitting code & software for further information.

## Data

Policy information about <u>availability of data</u>

All manuscripts must include a <u>data availability statement</u>. This statement should provide the following information, where applicable:

- Accession codes, unique identifiers, or web links for publicly available datasets
- A description of any restrictions on data availability
- For clinical datasets or third party data, please ensure that the statement adheres to our <u>policy</u>

> All data used in this study are publicly available. The already published segmentation datasets used for benchmarking and example demonstration are available from their sources as documented in Suppl. Table 2 and in the Dataset section of Methods. The original microscopy data generated for this paper are made available in a Zenodo repository, (https://doi.org/10.5281/zenodo.15692302)

## Human research participants

Policy information about <u>studies involving human research participants and Sex and Gender in Research.</u>

| | |
|---|---|
| Reporting on sex and gender | Samples were used for the purposes of demonstrating cell segmentation and not for obtaining biological insight. We did not consider sex and gender. |
| Population characteristics | Samples were used for the purposes of demonstrating cell segmentation and not for obtaining biological insight. No population characteristics or other covariates were considered. |
| Recruitment | Samples were used to demonstrate cell segmentation and not for obtaining biological insight. No specific recruitment criteria were used. T cells were obtained from blood leukocyte cones purchased from NHS Blood and Transplant, John Radcliffe Hospital, Oxford, UK. Blood cones were used under the ethical guidelines of the NHS Blood and Transplant. The Non-Clinical Issue division of National Health Service approved the use of blood leukocyte cones at the University of Oxford (REC 11/H0711/11). |
| Ethics oversight | T cells were obtained from blood leukocyte cones purchased from NHS Blood and Transplant, John Radcliffe Hospital, Oxford, UK. Blood cones were used under the ethical guidelines of the NHS Blood and Transplant. The Non-Clinical Issue division of National Health Service approved the use of blood leukocyte cones at the University of Oxford (REC 11/H0711/11). |

Note that full information on the approval of the study protocol must also be provided in the manuscript.

# Field-specific reporting

Please select the one below that is the best fit for your research. If you are not sure, read the appropriate sections before making your selection.

☒ Life sciences    ☐ Behavioural & social sciences    ☐ Ecological, evolutionary & environmental sciences

For a reference copy of the document with all sections, see nature.com/documents/nr-reporting-summary-flat.pdf

# Life sciences study design

All studies must disclose on these points even when the disclosure is negative.

| | |
|---|---|
| Sample size | For benchmarking u-Segment3D we used all available data in the public datasets. Additional datasets were selected to demonstrate further unique u-Segment3D features, not to derive biological conclusions or evaluate effect size. We use a pretrained Cellpose model and the rest of the calculations in u-Segment3D is deterministic therefore they are all n=1 demonstrations. |
| Data exclusions | We used all applicable data from public datasets for benchmarking experiments but had to exclude one image from DeepVesselNet which did not have both image and reference segmentation. |
| Replication | u-Segment3D is comprehensively benchmarked on >70,000 cells from 11 public datasets comprising diverse cell morphologies: convex, concave, branching and networks and from different cell densities: single cells, cell aggregates and tissue. All attempts at replication were successful. Demonstration of u-Segment3D on additional datasets used a pretrained Cellpose model and u-Segment3D algorithms are all deterministic, therefore results are by definition all replicable, given the same installed Cellpose version. |
| Randomization | Per standard segmentation benchmarking, we used the designated train/test/val splits provided by the public datasets. In Fig. 3 we generated train/test/val splits for datasets that did not have these splits, by random sampling. We follow the random sampling procedure detailed by EmbedSeg paper and used the same splits for all other algorithms to be fair. Additional experiments only seek to demonstrate features of u-Segment3D therefore no randomization was required (all computations are deterministics), n=1 were used per example. |
| Blinding | Blinding is not applicable for segmentation where all dataset is used, and there are no confounding covariates that would bias the result. Per standard segmentation benchmarking, we used the designated train/test/val splits provided by the public datasets. In Fig. 3 we generated train/test/val splits for datasets that did not have these splits, by random sampling. We follow the random sampling procedure detailed by |

EmbedSeg paper and used the same splits for all other algorithms to be fair. Additional experiments only seek to demonstrate features of u-Segment3D therefore no randomization was required (all computations are deterministics), n=1 were used per example.

# Reporting for specific materials, systems and methods

We require information from authors about some types of materials, experimental systems and methods used in many studies. Here, indicate whether each material, system or method listed is relevant to your study. If you are not sure if a list item applies to your research, read the appropriate section before selecting a response.

## Materials & experimental systems

| n/a | Involved in the study |
|---|---|
| ☐ | ☒ Antibodies |
| ☐ | ☒ Eukaryotic cell lines |
| ☒ | ☐ Palaeontology and archaeology |
| ☐ | ☒ Animals and other organisms |
| ☒ | ☐ Clinical data |
| ☒ | ☐ Dual use research of concern |

## Methods

| n/a | Involved in the study |
|---|---|
| ☒ | ☐ ChIP-seq |
| ☒ | ☐ Flow cytometry |
| ☒ | ☐ MRI-based neuroimaging |

## Antibodies

| | |
|---|---|
| Antibodies used | Primary antibody: Anti-Green Fluorescent Protein (GFP) chicken polyclonal primary antibody, Supplier Name: Aves Labs, Catalog Number: # GFP-1020<br><br>Secondary antibody: Alexa Fluor 488 AffiniPure F(ab')$_2$ Fragment Donkey Anti-Chicken IgY (IgG) (H+L), polyclonal, Supplier Name: Fisher Scientific, Catalog Number: # NC0456003 |
| Validation | Anti-Green Fluorescent Protein (GFP) chicken polyclonal primary antibody were analyzed by western blot analysis (1:5000 dilution) and immunohistochemistry (1:500 dilution) using transgenic mice expressing the GFP gene product. Western blots were performed using BiokHen (Aves Labs) as the blocking reagent, and HRP-labeled goat anti-chicken antibodies (Aves Labs, Cat. #H-1004) as the detection reagent. Immunohistochemistry used tetramethyl rhodamine-labeled anti-chicken IgY. |

## Eukaryotic cell lines

Policy information about cell lines and Sex and Gender in Research

| | |
|---|---|
| Cell line source(s) | MV3 cells were gifted by Peter Friedl (MD Anderson Cancer Center, Houston TX), whose lab established the line (DOI:10.1002/ijc.2910480116). We are unaware of a commercial source for this cell line.<br><br>Ker-CT cells (ATCC CRL-4048) were gifted by Dr. Jerry Shay (UT Southwestern Medical Center).<br><br>Blasted human CD8+ T cells, were produced by activating naïve T cells isolated from PBMCs.<br><br>COR-L23 cells were from Millipore Sigma<br><br>YUMM 1.7 cells were from ATCC. |
| Authentication | MV3 cells were authenticated using GenePrint 10 System from Promega.<br><br>Ker-CT cells were not authenticated.<br><br>Naïve T cells were checked using flow cytometry e.g. (CD3+CD45R0-CCR7+)<br><br>COR-L23 cells were not authenticated.<br><br>YUMM 1.7 cells were not authenticated. |
| Mycoplasma contamination | Cell lines were periodically tested and negative by PCR for mycoplasma contamination |
| Commonly misidentified lines (See ICLAC register) | No misidentified cell lines were used in this study |

## Animals and other research organisms

Policy information about studies involving animals; ARRIVE guidelines recommended for reporting animal research, and Sex and Gender in Research

| | |
|---|---|
| Laboratory animals | Danio Rerio (Zebrafish, strains: Tg(kdrl:Hsa.HRAS-mcherry) in a casper background. |

| Laboratory animals | Age: 3-5 dpf, therefore the sex of the organism was not yet determined. |
| --- | --- |
| | Mouse strains: NSG (https://www.jax.org/strain/005557, doi: 10.1038/s41586-020-2623-z) |
| | Age: Cancer cell injection around 6 weeks. Tumors take about a month to grow. |
| | Housing Conditions: Mice were housed at the Animal Resource Center at the University of Texas Southwestern Medical Center in Association for Assessment and Accreditation of Laboratory Animal Care International - accredited, specific pathogen-free animal care facilities under a 12 hours: 12 hours light:dark cycle with a temperature of 18c to 24c and humidity of 35% to 60%. |
| Wild animals | No wild animals were used. |
| Reporting on sex | The imaging experiments were conducted using both male and female animals. The purpose of the paper is to achieve cell segmentation irrespective of gender. |
| Field-collected samples | No field-collected samples were used. |
| Ethics oversight | All zebrafish husbandry and experiments described here have been approved and conducted under the oversight of the Institutional Animal Care and Use Committee (IACUC) at UT Southwestern under protocol number 101805. |
| | All mouse experiments complied with all relevant ethical regulations and were performed according to protocols approved by the Institutional Animal Care and Use Committee (IACUC) at the University of Texas Southwestern Medical Center (protocol 2016-101360). |

Note that full information on the approval of the study protocol must also be provided in the manuscript.

