## [Peer Review File · Nature Methods]

Universal consensus 3D segmentation of cells from 2D segmented stacks

Corresponding Author: Dr Gaudenz Danuser

Version 0:

Decision Letter:

31st Oct 2024

Dear Gaudenz,

My apologies regarding the delay in reviewing your paper. We were waiting on a third ref who never delivered.

Your Article, "Universal consensus 3D segmentation of cells from 2D segmented stacks", has now been seen by two reviewers. As you will see from their comments below, although the reviewers find your work of considerable potential interest, they have raised a number of concerns. We are interested in the possibility of publishing your paper in Nature Methods, but would like to consider your response to these concerns before we reach a final decision on publication.

We therefore invite you to revise your manuscript to address these concerns. During your revision, we ask that you focus on (1) adding the requested benchmarking, (2) adding more quantitative metrics of performance, (3) better describing how to tune hyperparameters for optimal performance, and (4) improving installation of the software.

Link Redacted

We hope to receive your revised paper within 2-3 months. If you cannot send it within this time, please let us know. In this event, we will still be happy to reconsider your paper at a later date so long as nothing similar has been accepted for publication at Nature Methods or published elsewhere.

OPEN SCIENCE REQUIREMENTS

REPORTING SUMMARY AND EDITORIAL POLICY CHECKLISTS

DATA AVAILABILITY

All novel DNA and RNA sequencing data, protein sequences, genetic polymorphisms, linked genotype and phenotype data, gene expression data, macromolecular structures, and proteomics data must be deposited in a publicly accessible database, and accession codes and associated hyperlinks must be provided in the "Data Availability" section.

CODE AVAILABILITY

Please include a "Code Availability" subsection in the Online Methods which details how your custom code is made available. Only in rare cases (where code is not central to the main conclusions of the paper) is the statement "available upon request" allowed (and reasons should be specified).

MATERIALS AVAILABILITY

ORCID

Nature Methods is committed to improving transparency in authorship. As part of our efforts in this direction, we are now requesting that all authors identified as 'corresponding author' on published papers create and link their Open Researcher and Contributor Identifier (ORCID) with their account on the Manuscript Tracking System (MTS), prior to acceptance. This applies to primary research papers only. ORCID helps the scientific community achieve unambiguous attribution of all scholarly contributions. You can create and link your ORCID from the home page of the MTS by clicking on 'Modify my Springer Nature account'. For more information please visit <http://www.springernature.com/orcid>.

Sincerely,
Rita

Rita Strack, Ph.D.
Senior Editor
Nature Methods

Reviewers' Comments:

Reviewer #2 (Remarks to the Author):

In this article, the authors present u-Segment3D, a theory/toolbox for exploiting 2D cell segmentations to generate 3D segmentations without data retraining. Nowadays, there are 2D deep learning methods that are very efficient for cell segmentations. Achieving the same level of performance in 3D is a challenge, mainly due to computational bottlenecks and the limited availability of reliable annotated data. This is why I believe that the development of this type of tool is essential for the community of biologists. The results obtained with u-Segment3D are impressive, however, the clarity of the paper should be improved. My comments are listed below:

Comments:

- In some experiments, for instance in Extended Data Figure 12, you perform nuclei segmentations. I think that for these datasets, 1) a comparison to StarDist3D should be performed; 2) use StarDist2D instead of/in comparison to cellpose2D for the 2D segmentation methods.

- A reference to StarDist3D is missing, even if it is only for nuclei and star-convex shapes, as you demonstrate the performance of u-Segment3D also on nuclei.

- Among methods reconstructing 3D volume from 2D segmentations, you only compare to Cellpose3D and Omnipose3D. I think that comparisons with CellStitch and 3DCellComposer should also be performed. If not please explain why.

- Comparisons to 3D cell segmentation methods should be performed (e.g., StarDist3D as mentioned in a previous comment, or any pertinent non supervised or classical methods). It will give an idea of the accuracy we may lose/win in IOU and F1 score.

- When comparing with other methods, you only use the metric AP. You should also use F1 to have an idea of the over- or under-segmentation.

- Line 45 (and other place in the manuscript), "compatible with any 2D cell segmentation method". This phrase is too vague and maybe not exactly right. It is not clear for me the input(s) u-Segment3D can have. Is it only the result of instance segmentation (i.e., one ID by object)? Or it can be a binary image and your toolbox do the erosion + connected component? Does it have to be a pixel-based segmentation? For instance, with parametric active contours or surfaces you do not have an image at the end but a continuous representation of your objects. Does your toolbox can transform these results in a binary image + ID? I think this should be clearer in the manuscript, the kind of inputs and the "any" is too vague and general, I think.

- The manuscript is not easy to follow making the method hard to understand. For instance, in Section "A formal framework for 2D-to-3D segmentation" there is a mix between general concepts and steps of your methods, making hard to understand what you exactly do at the end. Fig. 1 (a-c) corresponds to concepts while Fig. 1 (d-f) are steps of your toolbox. This is confusing.

- Notations are not always defined or not when they appear the first time. For instance, F (line 190), L (line 207), line 194: in x_i^t , index i for slice and t for iteration? Figure 1 : the three-bar sign
- In Section « a Formal framework for 2D-to-3D segmentation” references to the Section Methods are missing to well understand the notations and steps.
- Section “Smoothing of reconstructed ... segmentation”, how a user can intuitively choose the value of sigma according to the structure/shape?
- Please, unifies the notations. For instance, id or ID, n-D or n D...
- Some terms are too vague. For instance, combination (Fig. 1 and in the text) could mean union or intersection.
- It seems that you have a lot of hyperparameters. For users is it intuitive to set them? Are there default values?
- In Supplementary Movie 4 we do not see the initial foreground points.
- In your method, to what exactly refers the word “consensus”? Is it because you unify two approaches (stitching and gradient aggregation), i.e., consensus between two methods? Consensus between the 2D segmentations in 3 different planes? Or ...
- Can you also explain the “universal” in the title?

Reviewer #2 (Remarks on code availability):

I tried the provided code for nuclei segmentation. The installation was quite easy. However, the setting of the hyperparameters was not intuitive. The best results we got was satisfactory but there was under-segmentation. In dense area, a lot of nuclei had the same label. Maybe, we did not found the best appropriate hyperparameters.

Reviewer #3 (Remarks to the Author):

Summary

Zhou et al., introduce a framework for generating 3D instance segmentations from 2D instance segmentations. In doing so, they propose a solution for a challenge in developing accurate 3D instance segmentation algorithms for bioimaging: producing densely-labeled 3D ground truth is difficult and time consuming. To overcome this challenge, the authors developed a framework for creating 3D gradients from the 2D segmentations that can be used cluster segmented pixels into instances. They demonstrate the method on a variety of datasets with challenging morphologies. Additionally, they show the method can scale to large images.

Major comments

- The manuscript is generally well written. I have highlighted a few places below in the "minor comments" that I think could use clarification. Overall, I think the explanations of the method and benchmarking are clear.
- The design of the algorithms is well thought out and supported by experiments and theory.
- One of the claims of the paper is that this method is a competitive alternative to directly-trained 3D segmentation algorithms. However, most of the benchmarking was done against Cellpose, which as the authors say, performs the initial segmentations in 2D. To support this claim, I think it would be helpful if the authors benchmarked against algorithms that directly segment in 3D (e.g., PlantSeg for the plant images).
- Related to the point above, I think it would be helpful to have a bit more context for the AP values in Figure 3 in order to evaluate the performance of u-Segment3D. For example, in Fig. 3c, does increasing the AP to ~0.40 make the segmentation "usable"? How would a different 3D segmentation algorithm that is more designed for that type of sample perform (e.g., PlantSeg)?
- In order for this toolbox to be incorporated into segmentation algorithms, the developers need to be able to add u-Segment3D as a dependency. Based on the linked repository, it doesn't appear that u-Segment3D is available via any of the standard Python package sources (e.g., PyPI, conda-forge). I encourage the authors should make their work available via PyPI and/or conda-forge so that others can build upon their work.

Minor comments

- This sentence is unclear to me: "As users, we also observed fragmentation artifacts around 3D cell centroids in the generated segmentation, incompatible with its theory and inconsistent with simply stitching the equivalent Cellpose 2D cell masks" (Page 3, line 122)
- It would be nice if the authors expanded on how their parallelized implementation of gradient descent works in the methods section. In particular, how are points propagated across tile boundaries (e.g., if a point's attractor is in a different tile)?
- Some of the labels are overlapping with other elements in the figures, making them difficult to read/understand (e.g., Fig 5b(i-

iv)).

Reviewer #3 (Remarks on code availability):

Overall, the software looks well written. I appreciate that the authors have documented many of the functions with docstrings. I ran into some installation issues (described below), but was able to install it in the end. I think it would be nice if the authors could make the installation process more smooth by both making the package available on PyPI and/or conda-forge (see my comment above) and by improving the installation instructions.

- I ran into some issues using the installation instructions in the README, but I in the end, successfully installed u-Segment3D on Ubuntu 22.04 with Python 3.9. Initially using the first set of installation instructions in the README, PyICU failed to install via PIP. Additionally, torch was not able to use CUDA. I was able to install PyICA via conda-forge and torch + CUDA using the suggested conda installation instructions on pytorch.org (conda install pytorch torchvision torchaudio pytorch-cuda=11.8 -c pytorch -c nvidia).

- I tried a couple of the examples and they worked, but there were some minor issues. I first ran the tutorials/multi_cells/segment_Tcells_3D.py script and it executed successfully. Initially, I didn't realize that the matplotlib plots that are displayed are blocking as long as the window is open so I thought the script was just taking a long time. It might be nice to either display them in a non-blocking way or add a note in the script instructions. I also ran the anisotropic/segment_single-cell_tracking_challenge_2D_stacks_with_cellpose_auto-diameter.py script and it seemed to produce segmentations, but it did raise the exception below.

...

```
Exception ignored in: <function Variable.__del__ at 0x7f0dbf4c0ca0>
Traceback (most recent call last):
File "/local0/miniconda/envs/u_Segment3D_env/lib/python3.9/tkinter/__init__.py", line 363, in __del__
if self._tk.getboolean(self._tk.call("info", "exists", self._name)):
RuntimeError: main thread is not in main loop
...
```

Version 1:

Decision Letter:

Our ref: NMETH-A57367A

30th Apr 2025

Dear Gaudenz,

My apologies for our delay in the second round of review for your paper, but I am writing with good news.

Thank you for submitting your revised manuscript "Universal consensus 3D segmentation of cells from 2D segmented stacks" (NMETH-A57367A). It has now been seen by the original referees and their comments are below. The reviewers find that the paper has improved in revision, and therefore we'll be happy in principle to publish it in Nature Methods, pending minor revisions to satisfy the referees' final requests and to comply with our editorial and formatting guidelines.

TRANSPARENT PEER REVIEW

ORCID

Sincerely,
Rita

Rita Strack, Ph.D.
Senior Editor
Nature Methods

Version 2:

Decision Letter:

7th Sep 2025

Dear Gaudenz,

I am pleased to inform you that your Article, "Universal consensus 3D segmentation of cells from 2D segmented stacks", has now been accepted for publication in Nature Methods. The received and accepted dates will be July 31, 2024 and September 7, 2025. This note is intended to let you know what to expect from us over the next month or so, and to let you know where to address any further questions.

Over the next few weeks, your paper will be copyedited to ensure that it conforms to Nature Methods style. Once your paper is typeset, you will receive an email with a link to choose the appropriate publishing options for your paper and our Author Services team will be in touch regarding any additional information that may be required. It is extremely important that you let us know now whether you will be difficult to contact over the next month. If this is the case, we ask that you send us the contact information (email, phone and fax) of someone who will be able to check the proofs and deal with any last-minute problems.

Authors may need to take specific actions to achieve compliance with funder and institutional open access mandates.

If your research is supported by a funder that requires immediate open access (e.g. according to [Plan S principles](https://www.springernature.com/gp/open-science/plan-s-compliance) or the [NIH public access policy](https://www.springernature.com/gp/open-science/us-federal-agency-compliance)) then you should select the gold OA route, and we will direct you to the compliant route where possible. Because authors warrant under our subscription licensing terms that they haven't committed to licensing any version of their article under a licence inconsistent with the terms of our agreement – including the applicable embargo period – publication under the subscription model isn't suitable for authors whose funders require no embargo.

If you are active on Twitter/X or Bluesky, please e-mail me your and your coauthors' handles so that we may tag you when the paper is published.

Best regards,
Rita

Rita Strack, Ph.D.
Senior Editor
Nature Methods

Visit the Springer Nature Editorial and Publishing website at http://editorial-jobs.springernature.com?utm_source=ejP_NMeth_email&utm_medium=ejP_NMeth_email&utm_campaign=ejp_Nmeth for more information about our career opportunities. If you have any questions please click [here](mailto:editorial.publishing.jobs@springernature.com).

Dallas, February 28th, 2025

Dear Rita and Reviewers,

Thank you for considering our article for publication in Nature Methods.

We are happy that both reviewers found merit in our work and have conducted the following set of revisions to address all issues raised. All changes in the revised manuscript is m by blue text.

Summary of main revisions

(1) adding the requested benchmarking,

(a) Reviewer 2 asked for additional comparisons to unsupervised 2D-to-3D segmentation methods, specifically cellstitch and cellcomposer.

We have added the results of applying cellstitch and cellcomposer to datasets with ideal orthoslice segmentations, in comparison to u-Segment3D (Fig. 2, Extended Data Fig. 8, 9). These experiments demonstrate the in-principle best performance these methods can be expected to achieve. We found irrespective of average precision or F1 metric, u-Segment3D performs the best overall, with close to optimal 3D reconstruction across all 11 datasets. In contrast, Cellstitch and cellcomposer struggle significantly as morphologies become more branched and network-like, suggesting strong sensitivity to discrete errors from using three orthoviews. Indeed, with the vasculature networks in DeepVesselNet both methods were unable to stitch 2D segmentation into contiguous networks, scoring 0 in AP and F1.

Moreover we compared CellStitch and CellComposer to 3D translate pretrained Cellpose 2D predicted segmentations on the three public datasets: Ovules, LRP and DeepVesselNet in Fig. 3. We find that whilst both methods demonstrated good reconstruction from ideal orthoslices, their performance dramatically drops in practice, e.g. for Ovules, from $AP_{50} = 0.8$ and 1.0 using ideal orthoslices to 0.25 with Cellpose 2D segmentations. Indeed, the resultant 3D segmentations were worse than a classical distance transform 3D watershed baseline. This demonstrates strong sensitivity of both methods to the quality of 2D segmentation. Moreover, when comparing the 2D segmentation performance of 2D slices of the 3D stitched segmentation, we found not only are the 2D orthoslices worse than the original 2D segmentation, but both methods demonstrated

strong x-y view bias, that is the 3D segmentation was primarily driven by x-y segmentations. Both x-z and y-z segmentations were significantly worse, suggesting these methods do not appropriately leverage complementary information across orthoviews. Only u-Segment3D constructed 3D segmentations that outperform the 3D watershed baseline, translate 2D segmentations to 3D without bias in any orthoview, with 2D slices that improved upon the native 2D segmentation, and generalizes to branched networks morphologies.

(b) Both reviewers requested comparison to native 3D segmentation methods.

We have added to our results with pretrained Cellpose 2D models (Fig. 3, Extended Data Fig. 13) new 3D baseline methods for comparison; a classical unsupervised 3D watershed as a lower performance method, and a native trained 3D segmentation method, as an upper performance method. For Ovules and LRP, we used the original neural network published with the dataset, PlantSeg with pretrained weights as the ‘trained’ 3D method. For DeepVesselNet, which has minimal background, we used classical 3D binary Otsu threshold segmentation as both the classical and trained 3D baseline. For all other datasets, all from the EmbedSeg paper, we trained EmbedSeg3D models as the trained 3D baseline.

We have further added a new figure, Fig. 4, where we trained native 3D segmentation models and their counterpart 2D segmentation models. We then specifically compared the AP and F1 performance of i) classical unsupervised 3D watershed, ii) native 3D segmentation with StarDist3D, EmbedSeg3D and PlantSeg3D (Ovules and LRP only), and iii) consensus 3D segmentation with 2D segmentation (StarDist2D, EmbedSeg2D, Cellpose2D and PlantSeg2D (Ovules and LRP only)) using u-Segment3D. Across all datasets, the 3D AP₅₀ of u-Segment3D is comparable to or exceeds the 2D AP₅₀ of the 2D model with strong Pearson’s R of 0.79, affirming u-Segment3D faithfully translates the 2D model performance (Fig. 4d) and benefits from the orthoview integration. In the majority of cases, particularly for densely packed and complex morphology cells, the AP₅₀ of 3D consensus u-Segment3D segmentation exceeds the AP₅₀ of 3D native segmentations (Fig. 4e). A notable advantage of 2D segmentation, which we had previously underappreciated, was reduction of the 3D shape complexity – a complex, concave 3D shape can have simpler, more convex 2D cross-section shapes in 2D orthoslices. This was most strikingly evidenced with StarDist3D for mouse skull nuclei where u-Segment3D (AP₅₀=0.80) significantly outperformed StarDist3D (AP₅₀=0.05). We find the main advantage of native 3D segmentation is to better capture small objects underrepresented in 2D slices. The Platynereis nuclei dataset was the only one where u-Segment3D underperformed native 3D across all 2D models. In this case, retraining a 3D EmbedSeg model with u-Segment3D consensus segmentation as ground-truth improved the performance and recovered the missing small nuclei (Fig. 4f).

Finally, we additionally demonstrate the flexibility and consistency of u-Segment3D by showing 2D-to-3D segmentation using one, two and all three orthoviews on LRP (Fig. 4g). As expected, 3D segmentation performance increased with the number of orthoviews integrated. More interestingly, because this dataset is aligned with the orthogonal axes, the performance of two views nearly matches that of three views. Notably, neither cellstitch, cellcomposer nor Cellpose allow less than three orthoviews.

(2) adding more quantitative metrics of performance,

We have added F1 curve as a complementary measure of performance to AP as suggested by both reviewers. Writing both metrics in terms of true positives (TP), false positives (FP) and false negatives (FN), $F1 = \frac{2TP}{2TP+FP+FN}$ and $AP = \frac{TP}{TP+FP+FN}$. F1 is therefore correlated to AP that upweights the contribution of TP. Consequently, we add F1 curves for the idealized orthoslice experiment (Extended Data Fig. 9) and for the native 3D segmentation comparison (Fig. 4, Extended Data Fig. 14) and performance of trained 2D segmentation models (Extended Data Fig. 15). In all cases, F1 scores lead to the same interpretation as AP, but with higher values.

(3) better describing how to tune hyperparameters for optimal performance,

We have added a comprehensive parameter table (new Suppl. Table 1). We detail all tunable parameters with reference to the variable names used in the GitHub code. We highlight which parameters require tuning or monitoring for each new dataset, which are context-dependent – to be adjusted for a dataset based on knowledge of the acquisition and cell morphological properties, and which parameters have little effect most of the time, and thus are left default. We explain all parameters and the process they govern, and we give suggestions on when and how to adjust.

(4) improving installation of the software.

We have made u-Segment3D installable via PyPI to allow more direct installation as suggested by reviewers. Moreover, we reviewed and removed dependencies that previously required special handling, namely cuCIM and pyicu. We also use pyproject.toml to automatically configure installation for the three major operating systems: Windows, Linux and MacOS. Lastly, we provide instructions to install u-Segment3D using the GitHub link without need to first clone the repository.

Reviewer #2 (Remarks to the Author):

In this article, the authors present u-Segment3D, a theory/toolbox for exploiting 2D cell segmentations to generate 3D segmentations without data retraining. Nowadays, there are 2D deep learning methods that are very efficient for cell segmentations. Achieving the same level of performance in 3D is a challenge, mainly due to computational bottlenecks and the limited availability of reliable annotated data. This is why I believe that the development of this type of tool is essential for the community of biologists. The results obtained with u-Segment3D are impressive, however, the clarity of the paper should be improved. My comments are listed below:

We certainly appreciate the reviewer's recognition of the importance of 2D to 3D segmentation translation. Even more so are we pleased by the reviewer's recognition that this requires proper theory to overcome computational hurdles.

Comments:

- In some experiments, for instance in Extended Data Figure 12, you perform nuclei segmentations. I think that for these datasets, 1) a comparison to StarDist3D should be performed; 2) use StarDist2D instead of/in comparison to cellpose2D for the 2D segmentation methods.

We have added a new Fig. 4 and Extended Data Fig. 14 that explicitly compares the 3D segmentation performance on individual datasets including nuclei of i) a classical distance transform 3D watershed, ii) native trained 3D segmentation models: EmbedSeg3D, StarDist3D and PlantSeg3D, with iii) u-Segment3D applied to trained 2D segmentation models: EmbedSeg2D, StarDist2D, Cellpose2D and PlantSeg2D. These additional experiments highlight the universal compatibility of applying u-Segment3D's indirect method with any 2D segmentation method that can produce instance cell masks. Fig. 4 demonstrates comparable and most of the time superior performance of consensus 3D segmentation by u-Segment3D over native 3D segmentation, particularly for densely packed cells with complex morphologies. We were particularly impressed by the performance boost of StarDist2D with u-Segment3D ($AP_{50}=0.80$) over native StarDist3D ($AP_{50}=0.05$) on the Mouse skull nuclei dataset, suggesting significant advantage in segmenting complex 3D shapes via its simpler and more convex-shaped 2D slices.

- A reference to StarDist3D is missing, even if it is only for nuclei and star-convex shapes, as you demonstrate the performance of u-Segment3D also on nuclei.

The original version already referenced StarDist3D in the context of nuclei in the introduction (Weigert et al., (2020), “*Star-convex polyhedra for 3D object detection and segmentation in microscopy*”) c.f. lines 87-88, “Consequently, both classical⁴³⁻⁴⁶ and deep-learning based⁴⁷⁻⁴⁹ 3D segmentation method development focus primarily on nuclei, ...”

The revised version also references StarDist2D and StarDist3D in the section corresponding to our new Fig. 4. benchmarking with native 3D segmentation, “We evaluated EmbedSeg3D and 2D¹⁵ as representative of the state-of-the-art instance embedding approach; StarDist3D⁴⁷ and 2D² as representative of the shape-prior approach; ... ” (lines 626-627).

- Among methods reconstructing 3D volume from 2D segmentations, you only compare to Cellpose3D and Omnipose3D. I think that comparisons with CellStitch and 3DCellComposer should also be performed. If not please explain why.

We had not considered CellStitch and 3DCellComposer as comparators previously, because they can be formulated as limited instances of u-Segment3D. We have now performed like-for-like explicit benchmark comparison with both methods for translating ideal orthogonal slices (Fig. 2, Extended Data Fig. 8,9) and for pretrained Cellpose2D segmentations (Fig. 3). As expected, u-Segment3D is the overall best-performing method, with optimal 3D reconstruction across all 11 datasets, significantly outperforming on branched (VesselMNIST3D, zebrafish macrophages) and network (DeepVesselNet) shapes. Cellstitch always exhibits a performance gap. Meanwhile, Cellcomposer performs ideally on several datasets (Ovules, LRP, Arabidopsis) but is inconsistent and unpredictable, exhibiting significant performance gap on ‘simpler’ datasets (Mouse organoids, C. Elegans). The performance gap is clear with Cellpose 2D segmentations. Only u-Segment3D demonstrates good performance across all 3 datasets, comparable to native 3D segmentation. Both Cellstitch and Cellcomposer segmentations suffer detrimentally, on Ovules even performing worse than a classical 3D watershed. We further show u-Segment3D is the only method that leverages complementary information across orthoviews. The stitched 3D segmentation from CellStitch and Cellcomposer are dominated by the x-y segmentations (Fig. 3).

- Comparisons to 3D cell segmentation methods should be performed (e.g., StarDist3D as mentioned in a previous comment, or any pertinent non supervised or classical methods). It will give an idea of the accuracy we may lose/win in IOU and F1 score.

We have added native 3D cell segmentation methods: a classical unsupervised distance transform 3D watershed segmentation, and an appropriate native trained 3D segmentation to our results with pretrained Cellpose2D in Fig. 3 to aid interpretation.

We also added a new Fig. 4 and Extended Data Fig. 14 explicitly comparing 3D segmentation performance with AP and F1 curve on individual datasets between i) classical distance transform 3D watershed, ii) native trained 3D segmentation models: EmbedSeg3D, StarDist3D and PlantSeg3D, and iii) u-Segment3D applied to trained 2D segmentation models: EmbedSeg2D, StarDist2D, Cellpose2D and PlantSeg2D.

- When comparing with other methods, you only use the metric AP. You should also use F1 to have an idea of the over- or under-segmentation.

Per the reviewer's suggestion, we have added F1 curve as a complementary measure of performance to AP (average precision). Mathematically, we expect both measures to show similar interpretation. Writing F1 and AP in terms of true positives (TP), false positives (FP) and false negatives (FN), $F1 = \frac{2TP}{2TP+FP+FN}$ and $AP = \frac{TP}{TP+FP+FN}$. Both take into account the effects of over- and under-segmentation via FP and FN, but the F1 score upweights the TP. We chose in the original version to use AP as the primary measure, because it gives equal weighting to TP, FP and FN. In the revised version, we have added F1 curves for the idealized orthoslice experiment (Extended Data Fig. 9), the native 3D segmentation comparison (Fig. 4, Extended Data Fig. 14) and the performance comparison of trained 2D segmentation models (Extended Data Fig. 15). In all cases, F1 and AP are correlated. Thus F1 yields the same interpretation as AP, but has higher value.

- Line 45 (and other place in the manuscript), “compatible with any 2D cell segmentation method”. This phrase is too vague and maybe not exactly right. It is not clear for me the input(s) u-Segment3D can have. Is it only the result of instance segmentation (i.e., one ID by object)? Or it can be a binary image and your toolbox do the erosion + connected component? Does it have to be a pixel-based segmentation? For instance, with parametric active contours or surfaces you do not have an image at the end but a continuous representation of your objects. Does your toolbox can transform these results in a binary image + ID? I think this should be clearer in the manuscript, the kind of inputs and the “any” is too vague and general, I think.

This is very valuable feedback. We have revised the text to clarify that u-Segment3D takes any pixel-based instance segmentation input. For example, StarDist predicts convex shapes parameterized by rays, but as shown in Fig. 4, it is compatible with u-Segment3D because StarDist

has the option to convert its predictions to pixel instance masks. In general, there are numerous tools users could use to inter-convert contour-based and pixel-based representations.

We highlight the main textual changes:

In abstract: “Here we develop a theory and toolbox, u-Segment3D for 2D-to-3D segmentation, compatible with any 2D method generating pixel-based instance cell masks.”

In our outline of the general 2D-to-3D algorithm in the section, “*A formal framework for 2D-to-3D segmentation*” we explicitly write “1. Generate 2D pixel-based instance segmentations independently in orthogonal x-y, x-z, y-z views.”

u-Segment3D only translates supplied 2D instance segmentation masks. Therefore, it will perform instance 3D segmentation only if the supplied 2D segmentations are of individual instances specified either as (i) probability+gradients or (ii) labelled image (one ID per object)). If 2D binary segmentations were specified, then the binary mask in each 2D slice is treated as being one ‘cell’, irrespective of spatial contiguity. Connected component analysis should be applied to uniquely label individual contiguous regions for correct 3D translation. We now explicitly mention in discussion how to treat binary and non-image-based representations: “These theoretical insights led us to develop a general toolbox, u-Segment3D, to robustly implement 2D-to-3D segmentation universally for any 2D method that generates pixel-wise instance segmentation masks. Other representations should be first converted to labeled images to use u-Segment3D. For example, using connected component analysis to label individual contiguous regions in 2D binary masks, and rasterization to convert polygonal contour segmentations to pixel-wise masks.” (lines 804-808).

- The manuscript is not easy to follow making the method hard to understand. For instance, in Section “A formal framework for 2D-to-3D segmentation” there is a mix between general concepts and steps of your methods, making hard to understand what you exactly do at the end. Fig. 1 (a-c) corresponds to concepts while Fig. 1 (d-f) are steps of your toolbox. This is confusing.

We appreciate this valuable feedback. Segmentation is a crowded arena with many models but not much theoretical and conceptual justification. We feel a highlight of our manuscript is establishing formalism to 2D-to-3D segmentation. As shown by our empirical results, concept was much needed to avoid pitfalls of existing methods.

We structured the text to first introduce high-level concepts and then describe our implementation of it. This was done to emphasize to the reader, u-Segment3D is a particular realization of a general theory, and to not bog the reader down in the beginning with specific implementation details. Fig 1(a-c) illustrates the general rationale, of translating 2D

segmentations into 3D in an eroded space. Fig. 1(d-f) converts this concept to mathematical computations that can be implemented in code.

To aid understanding, we have revised the panel titles in Fig. 1 with more description of the goal:

- In Fig. 1c: “nD objects can be decomposed into spatially contiguous objects in n-1D slices”
- In Fig. 1d: “nD objects can be reconstructed from spatially contiguous objects in n-1D slices from n orthoviews by n-1D erosion and nD spatial contiguity

We have revised the text of the entire section to incorporated more explanatory sentences of the rationale, and to highlight the distinction between general concept and the transition to practical implementation (in particular lines 183-234). We hope the reviewer will be satisfied with this revised approach to our writing strategy.

- Notations are not always defined or not when they appear the first time. For instance, F (line 190), L (line 207), line 194: in x_i^t , index i for slice and t for iteration? Figure 1 : the three-bar sign

Thanks for the very helpful suggestion. We have now defined all notation in the description of the 2D-to-3D algorithm in the main text, section “*A formal framework for 2D-to-3D segmentation*” and in Fig. 1.

F denotes a consensus averaging function. L denotes the label image representation of instance cell segmentations. x_i^t denotes the x -coordinate of the i th foreground voxel at iteration t . The three-bar sign denotes equivalent representation in Fig. 1.

- In Section « a Formal framework for 2D-to-3D segmentation” references to the Section Methods are missing to well understand the notations and steps.

We have added missing references to Methods.

- Section “Smoothing of reconstructed ... segmentation”, how a user can intuitively choose the value of σ according to the structure/shape?

We have constructed a new Suppl. Table 1 with parameter names directly referencing those found in the code. The table explains the rationale, highlights critical parameters required to be adjusted per dataset, and provides tuning advice.

The section is intended to draw attention to the importance of implementation. In actuality smoothing is most critical for 1D-to-2D segmentation. The 1D normalized gradients, which are either unit length up- (y-direction) or down- (x-direction), is the x- and y- components of the 'reconstructed' 2D gradients. Without any 2D smoothing, the reconstructed 2D gradients only captures vertical and horizontal directions, and not the full 360° directionality necessary to represent 2D shapes. In higher dimensions, however, e.g. for 2D-to-3D segmentation, components of the higher-dimensional gradients is constructed by averaging the gradients from at least two lower-dimensional orthoviews. For example, in 2D-to-3D segmentation, the x-component of the 3D gradients averages the x-component of 2D gradients from x-y and x-z orthoviews. Consequently, even without 3D smoothing, the reconstructed x-component has 3D context. In practice, there must still be some smoothing of the reconstructed 3D gradients for gradient descent as demonstrated, however the level of smoothing is fairly uncritical. Generally, we set $\sigma=1$ for our 2D-to-3D segmentation, and adjust the gradient decay only if there is need to impute missing segmentations across slices (see Suppl. Table 1.)

We have added extra text to emphasize this, c.f. lines 354-355: "Due to the sensitivity to cell size and shape, tuning σ is impractical". To avoid unnecessary confusion, we revised the ensuing section: "*Translating gradient smoothing and suppressed gradient descent to the case of 2D-to-3D segmentation*" to reinforce the message, lines 379-380: "However we can keep $\sigma = 1$ and increase $\tau = 0.5$ to still recover perfect construction of the branched cell.", and the concluding sentences of this section.

- Please, unifies the notations. For instance, id or ID, n-D or n D...

We have checked and unified our notation using id, nD, x-y, x-z, and y-z.

- Some terms are too vague. For instance, combination (Fig. 1 and in the text) could mean union or intersection.

In Fig. 1, we use 'combine' to describe any consensus operation to merge two independent binaries into one. We have now clarified this: "The binary generated by an x-direction scan must be combined by a consensus operation e.g. computing a pixel-wise average (and re-binarizing), or pixel-wise intersection with the equivalent binary produced from an orthogonal y-direction scan to reconstruct a single foreground binary representing fully separated 2D cells." (lines 175-178).

- It seems that you have a lot of hyperparameters. For users is it intuitive to set them? Are there default values?

In our python package, we provide default parameters implemented in a single parameters.py module (<https://github.com/DanuserLab/u-segment3D/blob/master/segment3D/parameters.py>) that can be instantiated as a dictionary, and used in functions that implement each major process (<https://github.com/DanuserLab/u-segment3D/blob/master/segment3D/usegment3d.py>): i) preprocessing, ii) cellpose2D segmentation, iii) 2D-to-3D translation, and postprocessing by iv) size filtering and gradient consistency, v) label diffusion and vi) guided filtering. Whilst there are many parameters, they operate in sets corresponding to intuitive and distinct image processing operations. In practice only a few parameter sets require adjustment. We address this with Suppl. Table 1, where we explain the parameters with the actual variable names from the code, explain the individual image processing operations and highlight which one may need adjustment.

We have also highlighted in-text that when using the indirect method with input instance segmentation masks as in Fig 4 and Methods, only the 2D-to-3D translation parameters matter with only 4 main parameters to tune: i) choice of distance transform, ii) total number of gradient descent iterations, iii) amount of gradient decay and iv) minimum cell size. i) and iii) are adjusted to account for branching morphology, ii) and iii) to account for over- and under- segmentation and iv) to remove spurious segmentations by size.

- In Supplementary Movie 4 we do not see the initial foreground points.

We checked the movie and agree the foreground points appear too light. We have regenerated Supplementary Movie 4 so that foreground points are darker.

- In your method, to what exactly refers the word “consensus”? Is it because you unify two approaches (stitching and gradient aggregation), i.e., consensus between two methods? Consensus between the 2D segmentations in 3 different planes? Or ...

We have added additional clarification of ‘consensus’ in the introduction text c.f. lines 105-108: “To address the shortcomings of directly training 3D segmentation models, we revisit the idea of leveraging 2D cell segmentations from orthoviews without data retraining to generate a “consensus” 3D segmentation – a 3D translation of 2D segmentations in orthogonal slices, i.e. orthoslices, which leverages complementary information between orthoslices, and preserves or even improves the accuracy of the 2D inputs.”

We use ‘consensus’ to describe the consistent integration of segmentations in partially information-redundant 2D slices into one 3D segmentation. A key property is that the 2D orthoslices of the constructed 3D segmentation are consistent or should even improve upon the original input 2D segmentations. This property of u-Segment3D is demonstrated in column (iv) of

Fig. 3b-d, and Fig. 4d. Surprisingly, we found that this consensus property is not realized by alternative methods in practice. Both Cellstitch and cellcomposer do not show consensus behavior and Cellpose lacks this as highlighted by our work (Extended Data Fig. 4e,f). The integration can be performed by using any one orthoview e.g. x-y view, any combination of two views, e.g. x-y and x-z view or with all three orthoviews.

- Can you also explain the “universal” in the title?

We have added clarification in the Introduction: “To derive a formal framework for 2D-to-3D segmentation unifying stitching and gradient tracing, we revisited the problem from first principles. We find that 2D-to-3D translation can be formulated generally as an optimization problem, whereby we reconstruct the 3D gradient vectors of the distance transform representation of each cell’s 3D medial-axis skeleton. Notably, 3D cells can then be reconstructed using gradient descent and spatial connected component analysis. Together these two principles allow the implementation of a universal consensus 3D segmentation from 2D segmentations in one, two or all three orthoview stacks (e.g., in x-y, x-z and y-z). Here universality refers to the independence of the framework to cell morphology and which 2D segmentation method is employed to generate pixel-based instance segmentation masks.”

We want a term similar to the current popular usage of ‘foundational’ and ‘generalist’ to distinguish segmentation models trained on multiple datasets as opposed to ‘specialist’ models that are trained on a single dataset. We use ‘universal’ to describe how u-Segment3D (i) is applicable to diverse morphologies, with the appropriate parameter settings (e.g. by setting a gradient decay in the gradient descent to account for significant branching in shapes, see new parameter table (Supplementary Table 1)), (ii) exhibits no bias to any particular orthoview (Fig. 3), and (iii) encapsulates and generalizes existing stitching and gradient-aggregation approaches for 2D-to-3D segmentation. For example, we show 2D-to-3D segmentation is not limited to Cellpose’s heat diffusion distance transform (Extended Data Fig. 2). More broadly, any medial-axis skeleton-based distance transform can be used to perform aggregation. Indeed, we find better alternatives than the heat diffusion distance transform depending on the morphology. As we have shown, whilst all stitching or gradient-based 2D-to-3D segmentation can be applied, they do not demonstrate faithful translation in all cases and therefore are not universal. If there is concern about this notion of universality, we are willing to consider other formulations of the title. We have no intention to overstate.

Reviewer #2 (Remarks on code availability):

I tried the provided code for nuclei segmentation. The installation was quite easy. However, the setting of the hyperparameters was not intuitive. The best results we got was satisfactory but there was under-segmentation. In dense area, a lot of nuclei had the same label. Maybe, we did not found the best appropriate hyperparameters.

We appreciate the reviewer's effort. We have added Suppl. Table 1, where each parameter is referenced with the variable name from the code and relevant image processing is explained with advice on tuning. As quantified in Fig. 4d, u-Segment3D generates 3D segmentations matching, if not improving, the performance of the 2D model. Thus, the quality of the 2D segmentation is fully translated to the 3D space, which is primary objective of this implementation. Users can be confident that if their 2D model generates good 2D segmentations, they can use the indirect method and resolve 3D clumped cells by setting `gradient_decay = 0` and increase the iteration number in gradient descent (see parameters table guide in Suppl. Table 1). Indeed, we recommend the indirect method, skipping steps 1 (preprocessing) and 2 (cellpose), whenever users already have good 2D instance segmentation masks. This greatly simplifies the running process with only four key parameters that are very intuitive to adjust, if necessary at all: i) distance transform choice, ii) total number of gradient descent iterations, iii) amount of gradient decay and iv) minimum cell size filter. Please note, if the clumped cells are not resolved by the 2D segmentation in orthoslices, then they would also not be resolved in 3D, as we found with the Platynereis nuclei dataset (Fig. 4e).

Reviewer #3 (Remarks to the Author):

Summary

Zhou et al., introduce a framework for generating 3D instance segmentations from 2D instance segmentations. In doing so, they propose a solution for a challenge in developing accurate 3D instance segmentation algorithms for bioimaging: producing densely-labeled 3D ground truth is difficult and time consuming. To overcome this challenge, the authors developed a framework for creating 3D gradients from the 2D segmentations that can be used cluster segmented pixels into instances. They demonstrate the method on a variety of datasets with challenging morphologies. Additionally, they show the method can scale to large images.

Major comments

- The manuscript is generally well written. I have highlighted a few places below in the "minor comments" that I think could use clarification. Overall, I think the explanations of the method and benchmarking are clear.

- The design of the algorithms is well thought out and supported by experiments and theory.

We are pleased to get this feedback, thanks.

- One of the claims of the paper is that this method is a competitive alternative to directly-trained 3D segmentation algorithms. However, most of the benchmarking was done against Cellpose, which as the authors say, performs the initial segmentations in 2D. To support this claim, I think it would be helpful if the authors benchmarked against algorithms that directly segment in 3D (e.g., PlantSeg for the plant images).

We agree. In our original submission, we did include a comparison to plants Omnipose3D, a model trained in 3D natively for lateral root primordial (Extended Data Fig. 12). But this was for one dataset.

We have now comprehensively assessed native 3D segmentation compared to consensus 3D segmentation with u-Segment3D. Fig 4. and Extended Data Fig. 14 compares i) classical unsupervised distance transform 3D watershed, ii) training native 3D segmentation models:

PlantSeg3D (Ovules and LRP only), EmbedSeg3D, StarDist3D, and iii) training counterpart 2D segmentation models: PlantSeg2D (Ovules and LRP only), EmbedSeg2D, StarDist2D, Cellpose2D and applying u-Segment3D. We find u-Segment3D preserves and often even exceeds the 2D model AP_{50} performance after translating segmentations to 3D (Fig. 4d). u-Segment3D matches or even exceeds counterpart native 3D segmentation (Fig. 4e) in all datasets except one, particularly when cells are dense and have complex morphologies. A prerequisite is of course that a 3D cell is sufficiently large in 2D slices, so as to be detected and segmented by the 2D models. In the *Platynereis* nuclei dataset this requirement is not met. Accordingly, though u-Segment3D method faithfully translated the 2D performance, both 2D models (EmbedSeg2D and StarDist2D) performed worse than native 3D. To further support this conclusion, we retrained a 3D segmentation model (EmbedSeg3D) using the consensus u-Segment3D segmentation (with EmbedSeg2D) as a weak reference and was able to significantly improve 3D segmentation, from $AP_{50}=0.7$ to $AP_{50}=0.85$, much closer to native 3D ($AP_{50}=0.97$).

- Related to the point above, I think it would be helpful to have a bit more context for the AP values in Figure 3 in order to evaluate the performance of u-Segment3D. For example, in Fig. 3c, does increasing the AP to ~ 0.40 make the segmentation "usable"? How would a different 3D segmentation algorithm that is more designed for that type of sample perform (e.g., PlantSeg)?

We appreciate this comment. We have now added to Fig. 3, Extended Data Fig. 12, 13 two independent native 3D segmentation baselines. The first is a classical unsupervised distance transform 3D watershed (dtWS, black dotted lines). The second is a 3D segmentation trained specifically for the sample platform (black dashed lines). For Ovules and LRP, this is PlantSeg where we applied the author's specific pretrained 3D models. For EmbedSeg datasets, this is EmbedSeg3D which we trained with the same parameters and setting according to the Github examples. For DeepVesselNet, the signal-to-noise ratio is good enough to apply a binary 3D Otsu threshold followed by connected component labeling. We find our tuning of pretrained Cellpose gives overall good performance, but in *Arabidopsis*, it is only as effective as dtWS. This is due to the 2D performance not being good enough. In Fig. 4 we reapplied u-Segment3D with trained 2D models and formally verify the faithful translation of 2D model performance to 3D performance (Fig. 4d). With trained 2D models, the segmentation performance matches or even exceeds native 3D performance (Fig.4e).

- In order for this toolbox to be incorporated into segmentation algorithms, the developers need to be able to add u-Segment3D as a dependency. Based on the linked repository, it doesn't appear that u-Segment3D is available via any of the standard Python package sources (e.g., PyPI, conda-

forge). I encourage the authors should make their work available via PyPI and/or conda-forge so that others can build upon their work.

We have made u-Segment3D installable through PyPI. We have also removed dependencies that previously required special handling, namely cucim.

Minor comments

- This sentence is unclear to me: "As users, we also observed fragmentation artifacts around 3D cell centroids in the generated segmentation, incompatible with its theory and inconsistent with simply stitching the equivalent Cellpose 2D cell masks" (Page 3, line 122)

Thank you for the feedback. We have rephrased this sentence (lines 124-126, "Moreover, we and others⁷¹ have observed fragmentation of whole 3D cells into angular sectors, a behavior inconsistent with Cellpose's representation of whole cells as a 360⁰ angular field, and worse than just stitching the 2D cell masks⁷¹.")

The gradient field of Cellpose represents an object as a 360-degree angular field. Under gradient descent, all points move to the center. Therefore, if a segmented shape e.g. a rectangle should fragment, i.e. oversegmentation, the triangle should fragment into two smaller rectangles, but should not fragment into 'angular sectors'. Unfortunately, the angular fragmentation is what is happening empirically with Cellpose 3D as seen in Extended fig. 4e and Suppl. Movie 2 for the example of a single cell. This behavior is odd because it doesn't fit the expected algorithm behavior, and is certainly inconsistent with the predicted Cellpose 2D segmentations, and not present if 2D segmentations were simply stitched together. From our analysis, see Extended Fig. 3d, we think this is because of the adaptive density-based clustering Cellpose is using to parse individual cells. This method assumes gradient descent converges to the attractor, such that local hotspots of point density capture all points part of a cell. However, when this doesn't occur e.g. insufficient gradient descent iterations and insufficiently smooth gradients, the identified local point density hotspots do not correspond to full cells, but cell parts. Our method of identifying cell clusters after gradient descent based on connected component analysis does not rely on the local density of points thereby generating well-formed cells irrespective of the number of gradient descent iterations (see Extended Data Fig. 3, 4). Also see Extended Data Fig. 16, where the spuriously generated cells generated by u-Segment3D still represent coherent shapes and not angularly fractured.

- It would be nice if the authors expanded on how their parallelized implementation of gradient descent works in the methods section. In particular, how are points propagated across tile boundaries (e.g., if a point's attractor is in a different tile)?

We rewrote and expanded the Methods section to clarify our implementation. As input, the foreground and gradients have been computed globally, over the full image. These are tiled with subvolumes such that adjacent subvolumes spatially overlap. Each subvolume is treated independently whereby foreground points are locally advected under gradient descent within the tiles. The output is the final positions of points, but in global coordinates i.e. by adding the offset corresponding to the origin of the subvolume. Spatial clustering is performed in the original full image space, with an optimized connected component labeling algorithm. The tiling splits the attractor field, but points are advected towards the same global attractor. Given a spatial overlap such that adjacent subvolumes sufficiently capture the attractor (even if partially), we find after connected component cell clustering points all still partake in the same global structure. We suggest a spatial overlap that should be set to be at least the size of a cell. Of course, structures spanning multiple subvolumes will still be split and require stitching in postprocessing.

- Some of the labels are overlapping with other elements in the figures, making them difficult to read/understand (e.g., Fig 5b(i-iv)).

We have made the figure panels smaller in Fig. 5b-d, and grouped panel labeling, panel title and genus number annotation together for easier reading. We have further added a rectangle bounding box to highlight the baseline comparison to 3D Binary Otsu thresholding, and increased the spacing within Fig. 5b-d and that between Fig. 5e-g.

Reviewer #3 (Remarks on code availability):

Overall, the software looks well written. I appreciate that the authors have documented many of the functions with docstrings. I ran into some installation issues (described below), but was able to install it in the end. I think it would be nice if the authors could make the installation process more smooth by both making the package available on PyPI and/or conda-forge (see my comment above) and by improving the installation instructions.

We have now made u-Segment3D installable from PyPI, <https://pypi.org/project/u-Segment3D>. We have removed dependencies that required special handling, namely cucim. We have removed PyICU as a dependency. We have added instructions for each major operating system of Linux,

Windows and MacOS, to maintain and document problems found for each. We will keep these continuously maintained in the GitHub repository.

- I ran into some issues using the installation instructions in the README, but I in the end, successfully installed u-Segment3D on Ubuntu 22.04 with Python 3.9. Initially using the first set of installation instructions in the README, PyICU failed to install via PIP. Additionally, torch was not able to use CUDA. I was able to install PyICA via conda-forge and torch + CUDA using the suggested conda installation instructions on pytorch.org (conda install pytorch torchvision torchaudio pytorch-cuda=11.8 -c pytorch -c nvidia).

This level of feedback is much appreciated. Thanks for the tips. We have removed the PyICU dependency and made a note in the installation instructions to resolve if there are errors either by module loading a newer version gcc compiler or by installing a precompiled version from conda-forge. We tested the reviewer's solution for torch not able to use CUDA after installation, and have included it in our GitHub README.

- I tried a couple of the examples and they worked, but there were some minor issues. I first ran the tutorials/multi_cells/segment_Tcells_3D.py script and it executed successfully. Initially, I didn't realize that the matplotlib plots that are displayed are blocking as long as the window is open so I thought the script was just taking a long time. It might be nice to either display them in a non-blocking way or add a note in the script instructions. I also ran the anisotropic/segment_single-cell_tracking_challenge_2D_stacks_with_cellpose_auto-diameter.py script and it seemed to produce segmentations, but it did raise the exception below.

...

```
Exception ignored in: <function Variable.__del__ at 0x7f0dbf4c0ca0>
```

```
Traceback (most recent call last):
```

```
File "/local0/miniconda/envs/u_Segment3D_env/lib/python3.9/tkinter/__init__.py", line 363, in __del__
```

```
if self._tk.getboolean(self._tk.call("info", "exists", self._name)):
```

```
RuntimeError: main thread is not in main loop
```

...

Apologies, we primarily develop and run scripts in Spyder, where plots are generated and retained in a separate window. We have now added non-blocking to all the matplotlib plt.show(), plt.close('all') to prevent overfilling of the screen, and plt.savefig() to save the figures to the save

folder for assessment later. We've not been able to reproduce the specified error for anisotropic/segment_single-cell_tracking_challenge_2D_stacks_with_cellpose_auto-diameter.py script. However, in checking we discovered and corrected errors in the counterpart anisotropic/segment_single-cell_tracking_challenge_2D_stacks.py script. We would like to emphasize that the u-Segment3D code will be continuously maintained on Github and that we are proactive in addressing any raised issues. We hope such glitches, which are unavoidable, will vanish over the years as users report them back to us.

We thank you for your consideration and look forward to your evaluation of the revised manuscript.

Sincerely,

Felix Zhou, Ph.D.
Instructor, Lyda Hill Department of Bioinformatics

Gaudenz Danuser, Ph.D.
Professor and Chairman, Lyda Hill Department of Bioinformatics
Patrick E. Haggerty Distinguished Chair in Basic Biomedical Science
CPRIT Scholar in Cancer Research